# A gene regulatory network for neural induction

**Katherine E Trevers[1†§], Hui-Chun Lu[1†], Youwen Yang[1‡#], Alexandre P Thiery[2‡], Anna C Strobl[1], Claire Anderson[1], Božena Pálinkášová[1], Nidia MM de Oliveira[1], Irene M de Almeida[1], Mohsin AF Khan[1], Natalia Moncaut[1¶], Nicholas M Luscombe[3,4], Leslie Dale[1], Andrea Streit[2], Claudio D Stern[1]***

[1]Department of Cell and Developmental Biology, University College London, London, United Kingdom; [2]Centre for Craniofacial and Regenerative Biology, King's College London, London, United Kingdom; [3]The Francis Crick Institute, London, United Kingdom; [4]UCL Genetics Institute, Department of Genetics, Environment and Evolution, University College London, London, United Kingdom

**\*For correspondence:**
c.stern@ucl.ac.uk

[†]These authors contributed equally to this work
[‡]These authors also contributed equally to this work

**Present address:** [§]Research Department of Pathology, UCL Cancer Institute, London, United Kingdom; [#]School of Cardiovascular Medicine & Sciences, King's College London, London, United Kingdom; [¶]Cancer Research UK Manchester Institute, The University of Manchester, Alderley Park, United Kingdom

**Competing interest:** The authors declare that no competing interests exist.

**Abstract** During early vertebrate development, signals from a special region of the embryo, the organizer, can redirect the fate of non-neural ectoderm cells to form a complete, patterned nervous system. This is called neural induction and has generally been imagined as a single signalling event, causing a switch of fate. Here, we undertake a comprehensive analysis, in very fine time course, of the events following exposure of competent ectoderm of the chick to the organizer (the tip of the primitive streak, Hensen's node). Using transcriptomics and epigenomics we generate a gene regulatory network comprising 175 transcriptional regulators and 5614 predicted interactions between them, with fine temporal dynamics from initial exposure to the signals to expression of mature neural plate markers. Using in situ hybridization, single-cell RNA-sequencing, and reporter assays, we show that the gene regulatory hierarchy of responses to a grafted organizer closely resembles the events of normal neural plate development. The study is accompanied by an extensive resource, including information about conservation of the predicted enhancers in other vertebrates.

## Editor's evaluation

In this manuscript, Trevers and colleagues undergo a detailed genome-wide exploration of the mechanisms of neural induction in chick embryos. They describe the gene regulations governing the patterning of extra-embryonic ectoderm into neural ectoderm upon the graft of an early Hensen's node ectopically, an assay for neural induction and neural commitment. The data are assembled into a Gene Regulatory Network of 175 transcription factors and their projected interactions, based on a fine-scale temporal analysis. This study will be an important resource for the field of neural induction.

## Introduction

One of the most influential studies in developmental biology was the discovery, 100 years ago, that a small region of the vertebrate embryo, named the 'organizer', can induce ectodermal cells that do not normally contribute to the neural plate to form a complete, patterned nervous system (*Spemann, 1921*; *Spemann and Mangold, 1924*). In amphibians, where these experiments were initially conducted, the 'organizer' resides in the dorsal lip of the blastopore. A few years later, Waddington demonstrated that an equivalent region exists in birds (duck and chick) and mammals (rabbit) (*Waddington, 1933*; *Waddington and Schmidt, 1933*; *Waddington, 1934*; *Waddington, 1936*; *Waddington, 1937*): the tip of the primitive streak, a structure known as Hensen's node (*Hensen, 1876*). This interaction

between the organizer and the responding ectoderm, which causes the latter to acquire neural plate identity, has been termed 'neural induction' (*Spemann and Mangold, 1924*; *Nieuwkoop, 1952*; *Gallera, 1971b*; *Saxén, 1980*; *Gurdon, 1987*; *Storey et al., 1992*; *Stern, 2005*).

Neural induction has often been imagined as a single event, 'switching' the fate of the responding tissue from non-neural to neural. But it is clear from timed grafting and subsequent removal of organizer transplants that the responding ectoderm requires about 12 hr of exposure to the organizer to acquire neural identity in a stable way ('commitment') (*Gallera and Ivanov, 1964*; *Gallera, 1971b*; *Streit et al., 1998*). Recent work has also revealed that the expression of many genes changes over time after grafting an organizer, suggesting that the process has considerable complexity (*Streit et al., 1997*; *Streit et al., 1998*; *Streit and Stern, 1999*; *Streit et al., 2000*; *Sheng et al., 2003*; *Stern, 2005*; *Albazerchi and Stern, 2007*; *Papanayotou et al., 2008*; *Gibson et al., 2011*; *Pinho et al., 2011*; *Papanayotou et al., 2013*; *Trevers et al., 2018*). What happens during this 12 hr period? Is it possible to define distinct steps, and perhaps identify the molecular events that represent 'induction' and 'commitment'? Surprisingly, given the long time since the discovery of neural induction, these questions have hardly been addressed. To begin to answer them requires a precise approach to identify and model the key interactions between genes and transcriptional regulators inside the responding cells that accompany their responses to signals from the organizer over time.

Grafting an organizer to a region of ectoderm that does not normally contribute to neural tissue (but is competent to do so) provides the opportunity to study the progress of neural induction relative to 'time-zero', the moment when the organizer is first presented to the tissue. It also allows for the separation of neural inductive events from other processes that occur adjacent to the normal neural plate (e.g. mesendoderm formation and patterning), because sites that are remote enough from the normal neural plate and are competent to respond to an organizer graft can generate a complete neural tube without induction of mesoderm, and without recruiting cells from the host's neural plate (*Hornbruch et al., 1979*; *Dias and Schoenwolf, 1990*; *Storey et al., 1992*).

Here, we have taken advantage of these properties, together with recent major technological advances in transcriptomics and epigenetic analysis, to dissect the molecular events that accompany neural induction in the chick embryo in fine time course and to generate the first detailed gene regulatory network (GRN) for this process. The GRN comprises 175 transcriptional regulators and 5614 predicted interactions between them over the course of neural induction. We then compare the spatial and temporal properties of these changes with development of the normal neural plate of the embryo using in situ hybridization, single-cell RNA-sequencing (scRNAseq) and reporter assays. The latter also allow us to test the activity of some of the key gene regulatory elements. We present a comprehensive resource allowing visualization and querying of all of these interactions and regulatory elements on a genome-wide level. Together, our study offers a global view of the genetic hierarchy of transcriptional regulatory interactions during neural induction and normal neural plate development over time.

## Results

### Transcriptional profiling identifies responses to neural induction in time course

In the chick embryo, a graft of Hensen's node to the inner area opaca, an extraembryonic region of competent non-neural ectoderm (*Gallera and Ivanov, 1964*; *Dias and Schoenwolf, 1990*; *Storey et al., 1992*; *Streit et al., 1998*), induces the formation of a mature, patterned neural tube after about 15 hr of culture from the host ectoderm (*Figure 1A*). To characterize the events over this period, the induction of several neural markers was assessed. *OTX2* is first expressed in the pre-primitive-streak stage epiblast at EGKXII-XIII (*Albazerchi and Stern, 2007*; *Pinho et al., 2011*) and is induced by 7/7 grafts after 3 hr (*Figure 1B–F*). *SOX2* is first expressed at HH4+/5 (*Rex et al., 1997*; *Streit et al., 1997*; *Streit et al., 1998*; *Sheng et al., 2003*) and requires 9 hr for induction (8/8) (*Figure 1G–K*). *SOX1* starts to be expressed weakly in the neural plate around HH7-8 (*Stavridis et al., 2010*; *Uchikawa et al., 2011*) and is induced by 50% of grafts after 12 hr (*Figure 1L–P*). This time course confirms that a sequence of events occurs in response to a grafted node over 0–12 hr (*Figure 1—figure supplement 1A*), culminating in the acquisition of neural plate/tube identity (*Streit et al., 1997*; *Streit et al., 1998*; *Streit and Stern, 1999*; *Streit et al., 2000*; *Sheng et al., 2003*; *Stern, 2005*; *Albazerchi and Stern,*

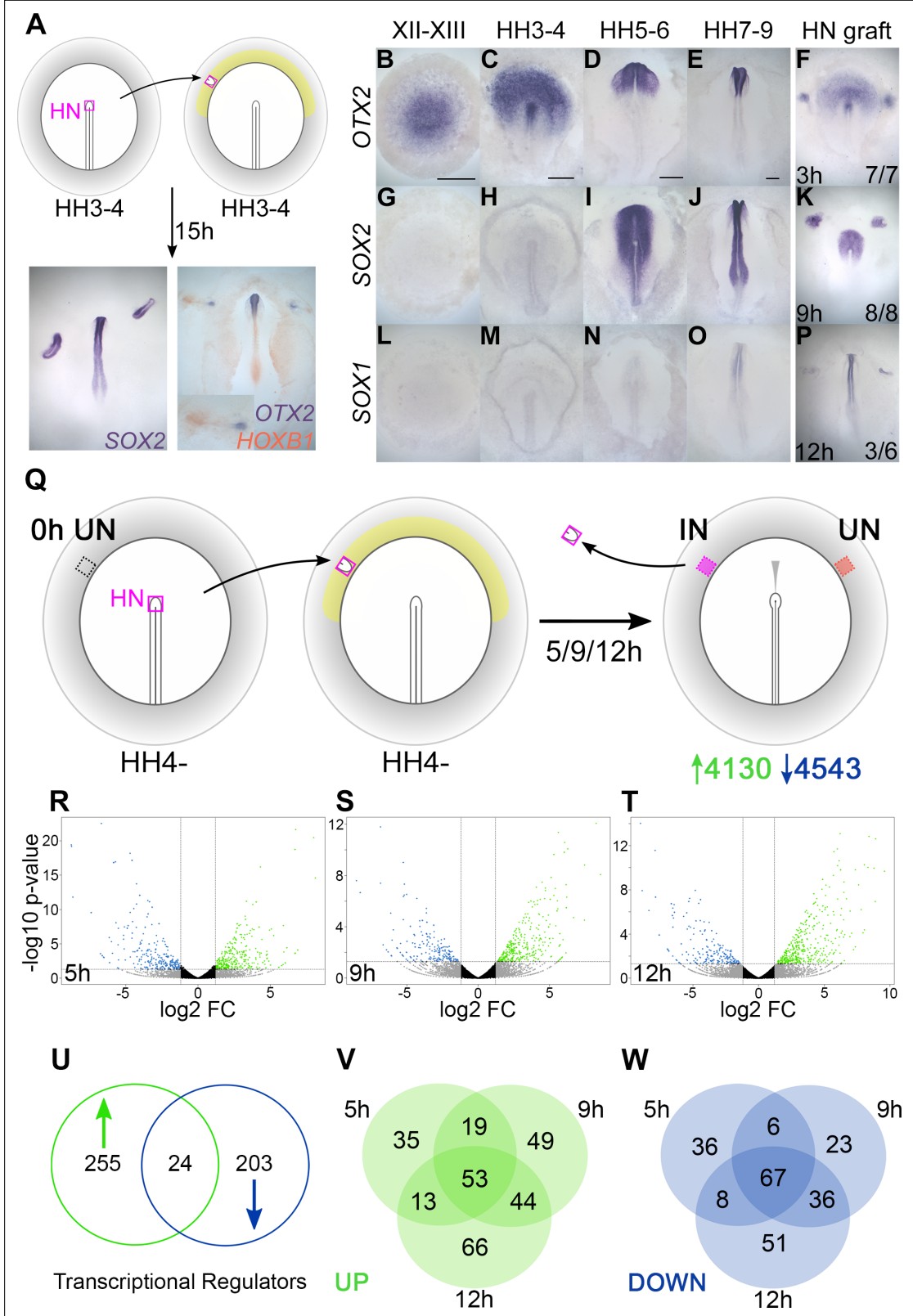

**Figure 1.** Transcriptional profiling identifies responses to neural induction in time course. (**A**) Hensen's node (HN) was grafted from HH3-4 donors to the inner area opaca (yellow) of hosts at the same stage. An ectopic neural tube expressing *SOX2*, *OTX2*, and *HOXB1* is induced after 15 hr of culture. (**B–F**) Expression of neural markers compared to their time course of induction by a node graft: *OTX2*; first expressed in pre-streak epiblast (EGKXII-XIII) and induced by grafts after 3 hr. (**G–K**) *SOX2*; first expressed in the neural plate at HH5-6 and induced after 9 hr. (**L–P**) *SOX1*; first expressed in the forming

*Figure 1 continued*

neural tube at HH7-8 and induced after 12 hr. (**Q**) Identifying transcriptional responses to a node graft. HN was grafted from HH4- donors to HH4- hosts. The HN graft was removed and underlying 'induced' (IN) and contralateral 'uninduced' (UN) ectoderm isolated after 5, 9, or 12 hr. Uninduced '0 hr' ectoderm from HH4- embryos was also dissected. Samples were analysed by RNA-sequencing (RNAseq). (**R–T**) Induced and corresponding uninduced tissues were compared at each time point to identify differentially expressed transcripts. Volcano plots show upregulated (green) and downregulated (blue) transcripts. (**U–W**) Venn diagrams of 482 genes encoding transcriptional regulators (**U**) that are upregulated (**V**) or downregulated (**W**) at different time points. Scale bars: B: 1mm; C: 250 μm; D: 250 μm; E: 250 μm. These scale bars apply to all other figures with embryos at equivalent stages throughout the paper.

The online version of this article includes the following source data and figure supplement(s) for figure 1:

**Source data 1.** RNAseq galGal3 analysis key and differentially expressed transcripts.

**Source data 2.** RNAseq galGal4 analysis key and differentially expressed transcripts.

**Source data 3.** Nanostring probe codeset.

**Figure supplement 1.** Further transcriptional responses to neural induction in time course.

*2007*; *Papanayotou et al., 2008*; *Gibson et al., 2011*; *Pinho et al., 2011*; *Papanayotou et al., 2013*; *Trevers et al., 2018*).

To uncover genes whose expression changes over this period, we performed RNAseq of the responding tissue at three time points following a graft of Hensen's node: 5 hr (to identify early 'pre-neural' responses), 9 hr (when *SOX2* expression defines neural specification), and 12 hr, as the host tissue starts to express *SOX1*. Hensen's node was grafted from HH4- chick donors to a region of competent epiblast (inner area opaca) of chick hosts at the same stage. Embryos were cultured for the desired period of time, after which the graft was removed and the underlying 'induced' tissue was collected. Control ('uninduced') ectoderm from the corresponding region on the contralateral side of the same embryos (*Figure 1Q*) was also collected, as well as competent area opaca at HH4-, representing a '0 hr' starting point for the time course. When 'induced' and 'uninduced' reads at each time point are compared by DESeq analyses, 8673 differentially expressed transcripts were identified (see volcano plots, *Figure 1R–T*). Of these, 4130 were upregulated (enriched in 'induced' tissue) and 4543 were downregulated (depleted in 'induced' tissue) relative to the 'uninduced' counterpart.

To construct a GRN we focused on 482 transcription factors and chromatin modifiers that are differentially expressed within these data. Of these 'transcriptional regulators', 255 are upregulated, 203 are downregulated, and 24 have more complex expression dynamics (*Figure 1U* and *Figure 1—figure supplement 1B*). Grouping transcripts based on the timing of their response (*Figure 1V–W*) reveals that 120 are differentially expressed throughout (53 upregulated and 67 downregulated), while others are associated with particular time points. Therefore, transcriptional responses to organizer grafts involve changes in the expression of many genes over a relatively short period.

Due to this complexity, we chose to increase the time resolution of the analysis. NanoString nCounter was used to quantify the gene expression changes of transcriptional regulators at six time points (1, 3, 5, 7, 9, and 12 hr after the node graft) in 'induced' tissue compared to 'uninduced' ectoderm. By consolidating the data from RNAseq and NanoString, a set of refined expression profiles was established for 213 transcriptional regulators – 156 that are enriched and 57 that are depleted in induced tissues (*Figure 3—source data 1*). These represent the core components of our GRN.

## Epigenetic changes identify chromatin elements associated with neural induction

We next sought to identify the regions of non-coding chromatin that govern these transcriptional responses to signals from the organizer. Histone modifications can regulate gene expression by altering chromatin structure. For example, H3K27ac is associated with actively transcribed genes and their enhancers, whereas transcriptionally inactive regions are often marked by H3K27me3 (*Tiwari et al., 2008*; *Heintzman et al., 2009*; *Creyghton et al., 2010*; *Kharchenko et al., 2011*; *Rada-Iglesias et al., 2011*; *Tolhuis et al., 2011*; *Zentner et al., 2011*; *Bonn et al., 2012*). Therefore, ChIPseq and ATACseq (*Buenrostro et al., 2013*; *Buenrostro et al., 2015*; *Corces et al., 2017*) were performed on induced and uninduced ectoderm following 5, 9, or 12 hr of a node graft, to detect histone and chromatin conformation changes during neural induction.

H3K27ac and H3K27me3 enriched chromatin (ChIPseq) were identified genome-wide by comparison to matched genomic input samples. Chromatin sites were then categorized according to their histone signatures across induced and corresponding uninduced tissues at each time point (*Figure 2—figure supplement 1A*). Sites that become acetylated and/or demethylated in induced tissues, compared to uninduced, were considered to undergo 'activation' (Indices 1–3) while those that become deacetylated and/or methylated undergo 'repression' (Indices 4–6). Chromatin marked by both H3K27ac (activation) and H3K27me3 (repression) marks in either tissue were described as 'poised' (Indices 7–12). The remaining indices (13–16) define sites that do not change marks between the two conditions.

Chromatin sites undergoing activation are enriched for H3K27ac marks in induced tissues at each time point (*Figure 2B–D*). Sites of repression are more varied: those belonging to this category at 5 hr lose acetylation, whereas several sites become methylated in induced tissues at 9 and 12 hr. Very few sites are 'poised' at any time. Overall, as neural induction progresses, the number of sites undergoing activation increases, along with a reduction in those that do not change state. In contrast, ATACseq suggests that there is little difference between induced and uninduced samples in terms of chromatin accessibility (*Figure 2—figure supplement 2*) at 5 hr, but the differences become more marked at later time points. At 12 hr, the ATACseq profile resembles that of the endogenous neural plate of the embryo (*Figure 2—figure supplement 2*).

Next, we focused on epigenetic changes associated with the 213 genes encoding transcriptional regulators for which refined expression profiles were established by NanoString analysis (see above). 'Constitutive' CTCF-bound sites (putative insulators) flanking these genes were obtained from chicken CTCF-ChIPseq data (*Khan et al., 2013*; *Kadota et al., 2017*). They were used to predict the boundaries within which we considered H3K27 marks as being associated with the gene of interest; these regions ('loci') are up to 500 kb in length and may contain several genes (*Figure 2—figure supplement 1B*). Within each locus, the H3K27ac or H3K27me3 enriched peaks were categorized as before (*Figure 2—figure supplement 1A*). The 213 selected loci contain a total of 6971 sites that change in response to a node graft (Indices 1–10; *Figure 2—figure supplement 1A*). We noticed that the flanking regions (proximal promoters) of transcriptional regulators whose expression increases are enriched for H3K27ac marks in induced tissue (*Figure 2—figure supplement 3*). In contrast, there is no difference between H3K27ac and H3K27me3 marks around the flanking regions of repressed transcriptional regulators at 5 and 9 hr – they only become enriched for methylation at 12 hr. Our analysis therefore identifies putative regulatory elements that are controlled epigenetically during the process.

## A GRN for neural induction

Having identified transcriptional responses to signals from the organizer and their accompanying chromatin changes, we combined them to construct a GRN to describe the time course of these events and to illustrate predicted interactions between transcriptional regulators. The open source platform BioTapestry (*Longabaugh et al., 2005*; *Longabaugh et al., 2009*; *Paquette et al., 2016*) (http://www.biotapestry.org) is used extensively to provide a visually intuitive representation of developmental networks (*Davidson, 1990*; *Arnone and Davidson, 1997*; *Davidson et al., 1998*; *Betancur et al., 2010*; *Simões-Costa and Bronner, 2015*; *Thiery et al., 2020*). Importantly, this software allows dynamic changes in the interactions between regulators and their target genes to be represented.

### A BioTapestry model for regulatory gene interactions

To integrate these complex time course data into a GRN that models the interactions during neural induction, we developed a custom computational pipeline (*Figure 3*). The 213 transcriptional regulators previously identified were selected as candidate components of the network because of their responses to a node graft in both RNAseq and NanoString data, and their associated chromatin changes in ChIPseq. Interactions were predicted by screening putative regulatory sites that belong to Indices 1–3 and 7 (activation) (*Figure 2—figure supplement 1C–E*) for the presence of putative binding sites for transcription factors that are differentially expressed in the dataset, and expressed at the appropriate time point to fit with a putative regulatory role (i.e. simultaneously or just before) for the target locus.

We then used the regulatory rules shown in *Figure 4A–B* to predict positive or negative regulatory events between regulators and their putative targets, overlaying the predicted transcription

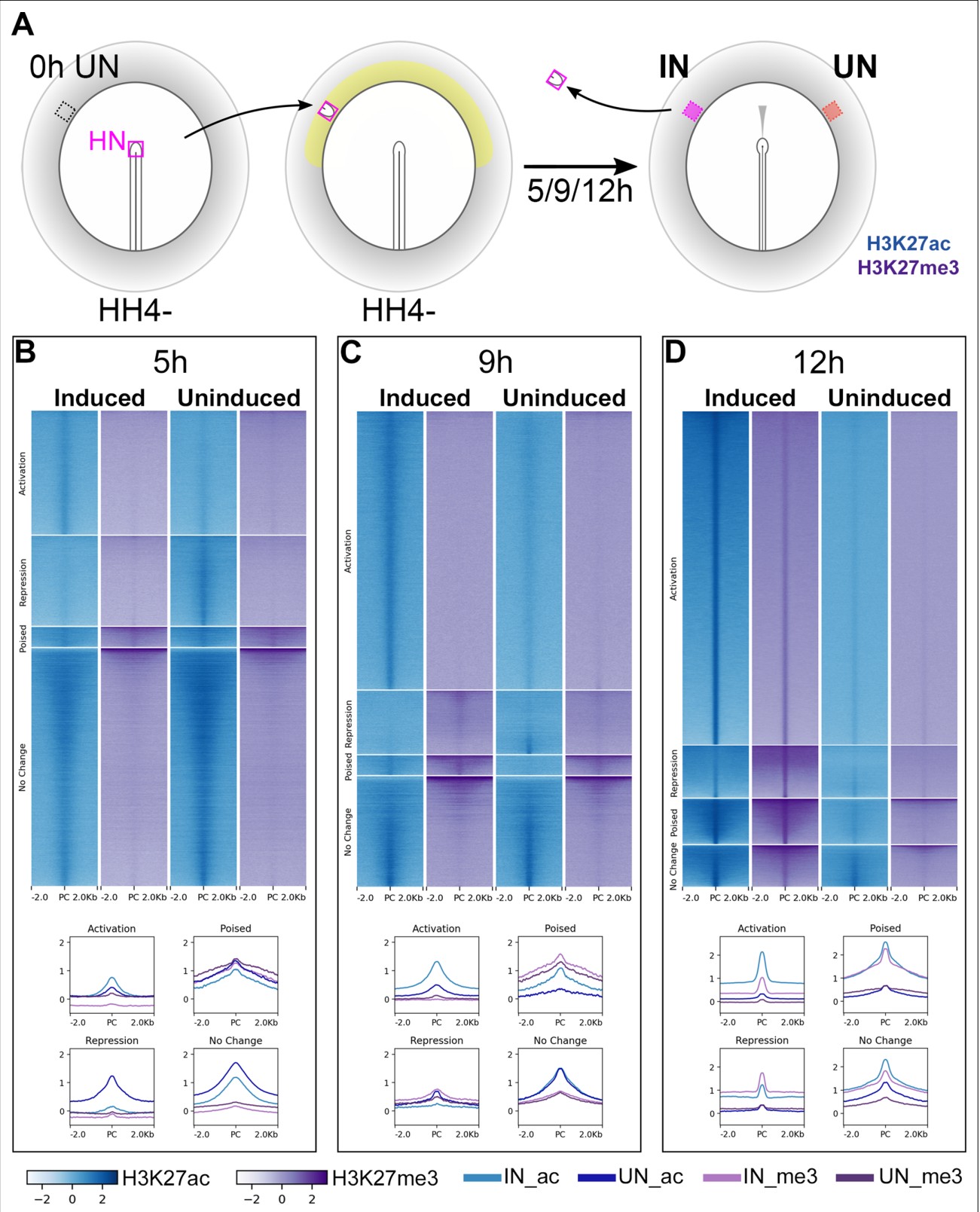

**Figure 2.** Epigenetic changes identify chromatin elements associated with neural induction. (**A**) Hensen's node (HN at HH4-) was grafted to hosts of the same stage. Embryos were cultured for 5, 9, or 12 hr before HN was removed and induced (IN) and contralateral uninduced (UN) ectoderm collected. ChIPseq was performed for H3K27ac and H3K27me3. (**B–D**) Putative regulatory sites were predicted according to the H3K27 profiles of IN and UN tissues at each time point (see *Figure 2—figure supplement 1*). They include sites undergoing 'activation' or 'repression', being 'poised', or showing

*Figure 2 continued on next page*

*Figure 2 continued*

no change. Heat maps illustrate the enrichment of H3K27ac (blue) and H3K27me3 (purple) in IN and UN tissues within ±2.0 kb from the peak centre (PC) for each annotated group. Graphs plot the distributions of H3K27ac and H3K27me3 enrichment for each group.

The online version of this article includes the following source data and figure supplement(s) for figure 2:

**Source data 1.** ChIPseq tissue dissociation details.

**Source data 2.** ATACseq indices.

**Figure supplement 1.** Further chromatin profiles during neural induction.

**Figure supplement 2.** ATACseq identifies open chromatin during neural induction.

**Figure supplement 3.** H3K27 histone marks flanking transcriptional regulators that are differentially expressed during neural induction.

factor binding profiles with the time course of expression from RNAseq and NanoString. At each time point, changes in expression of a putative regulator that could lead to the same change in a candidate target are represented as positive interactions. Negative, or inhibitory, interactions are predicted when changes in expression of the regulator could cause the opposite change in a downstream target. Genes lacking both regulatory inputs and outputs with other genes in the network were excluded.

The resulting GRN (*Figure 4C*), depicting interactions between 175 genes, was represented using BioTapestry (*Supplementary file 1* and *Supplementary file 2*). Most targets are predicted to receive multiple positive and/or negative inputs, which are initiated at, and can act across, multiple time points. For example, the non-neural gene *GATA2* is predicted to be repressed by *TFAP2C* after just 1 hr of a node graft, followed by increasing repression via *HIF1A*, *MEIS2*, and numerous other factors over the remaining time course. On the other hand, *BLIMP1* is predicted to be induced by *TFAP2C* after 1 hr, alongside *SNAI1*, *ETV1*, *ETV4*, *MGA*, and *SOX4* (*Supplementary file 2*). In total, 5614 interactions are predicted to occur between these components, highlighting the intricate and highly dynamic sequence of regulatory events that are triggered by signals from the organizer.

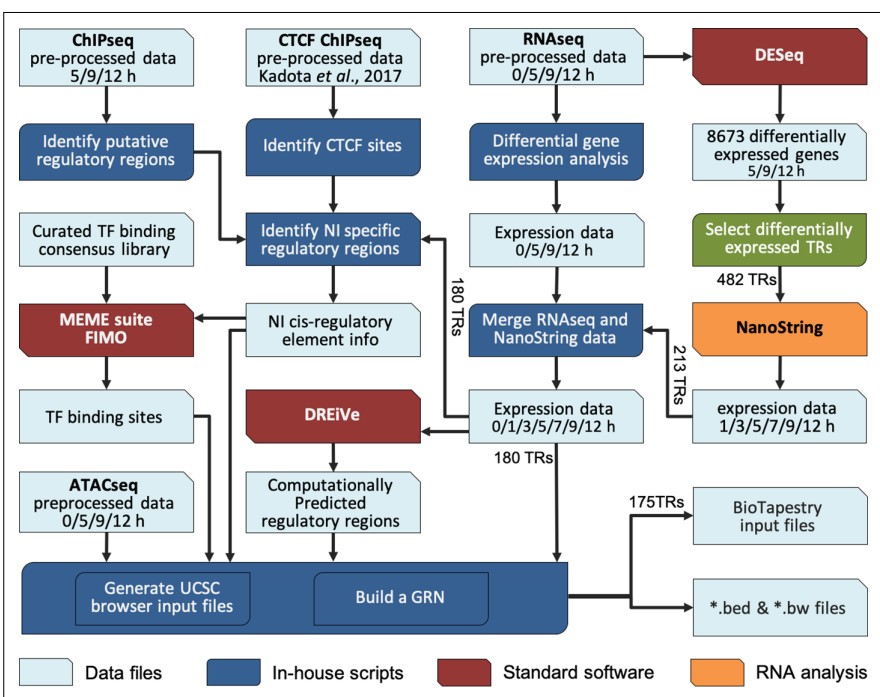

**Figure 3.** Constructing a BioTapestry model for regulatory gene interactions: computational workflow. Computational pipeline to combine transcriptomic and epigenetic time course data, to build a gene regulatory network (GRN) for neural induction. The output data are available to view in the UCSC browser.

The online version of this article includes the following source data for figure 3:

**Source data 1.** Integrated GRN gene expression profile.

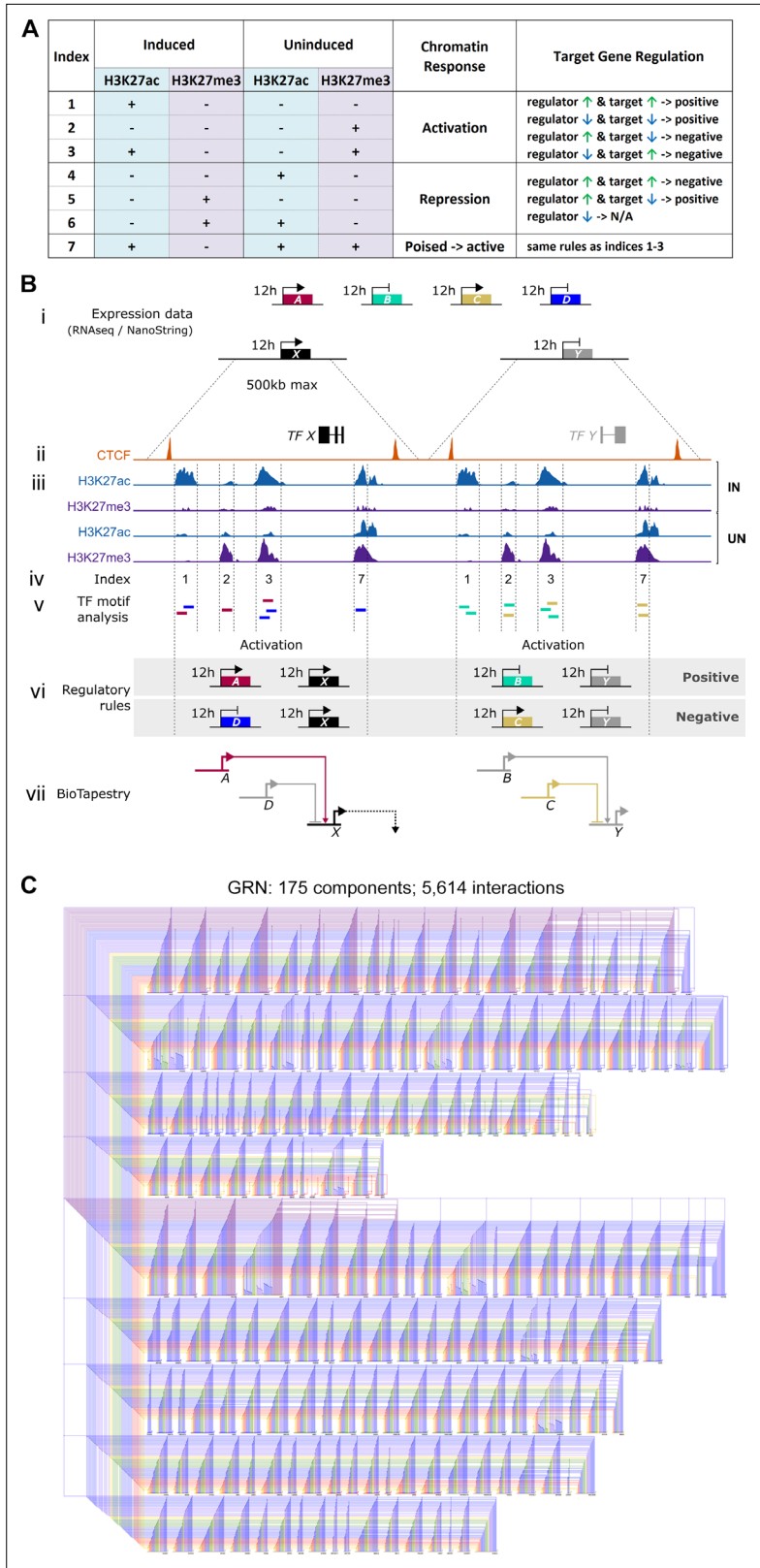

**Figure 4.** Constructing a BioTapestry model for regulatory gene interactions: the regulatory logic. (**A**) Network interactions were modelled using chromatin sites that belong to Indices 1–7 and the defined rules between regulators and their potential targets. At sites that become active in induced tissues (Indices 1, 2, 3, and 7): positive interactions are depicted when a putative input and its target are co-regulated and negative (inhibitory)

*Figure 4 continued on next page*

*Figure 4 continued*

interactions are modelled when an input and its target have opposing expression profiles. These rules are reversed at sites that undergo repression in induced tissues (Indices 4, 5, and 6) except that interactions are not predicted when the potential regulators of these repressed sites are not expressed. (**B**) Schematic depicting how expression and chromatin profiles were combined to predict interactions between gene regulatory network (GRN) components. (i) RNA-sequencing (RNAseq) and NanoString expression data provide a list of 213 transcriptional regulators that are upregulated (genes A, C, and X) or downregulated (genes B, D, and Y) after 0, 1, 3, 5, 7, 9, or 12 hr of a node graft. (ii) For these candidate GRN components, CTCF-ChIPseq data was used to predict neighbouring CTCF-bound domains of up to 500 kb. (iii) Within these, sites that are enriched for H3K27ac or H3K27me3 were identified by ChIPseq performed on induced and uninduced tissues at 5, 9, or 12 hr. (iv) Chromatin sites were categorized (according to *Figure 2—figure supplement 1A*) and those belonging to Indices 1, 2, 3, and 7 were selected to build a network. (v) These were screened for transcription factor binding motifs that correspond to other GRN components. This identifies genes A and D as potential regulators of target X and genes B and C as potential regulators of target Y. (vi) At each time point, the consequence of interactions was predicted according to the regulatory rules in **A**. Positive interactions are depicted when a putative input and its target are co-regulated. Negative (inhibitory) interactions are modelled when an input and its target have opposing expression profiles. (vii) Predicted interactions are modelled in BioTapestry using arrows for positive interactions and blunt ends for inhibitory interactions. Components are shown in colour at time points when they are expressed or shaded grey when they are not. (**C**) A GRN comprising 5614 predicted interactions between 175 components is visualized using BioTapestry. Each component is coloured by the time point when it first starts to express at 0 hr (purple), 1 hr (grey-blue), 3 hr (blue), 5 hr (green), 7 hr (yellow), 9 hr (orange), and 12 hr (red).

The online version of this article includes the following source data for figure 4:

**Source data 1.** List of GRN transcription factor binding sites within regulatory sites at 5h, 9h and 12h following a node graft.

## Incorporating individual regulatory sites and their dynamics into the network

BioTapestry networks usually represent each gene once, with multiple regulatory inputs, and each regulator as a single input into the target gene. However, it is now known that gene expression is controlled by multiple elements, each with characteristic spatial and temporal activity. In the past, such elements were identified within non-coding regions that are conserved across species by testing their ability to direct appropriate expression in vivo. This approach was used, for example, to identify many enhancers controlling *SOX2* expression (*Uchikawa et al., 2003*; *Okamoto et al., 2015*) – the spatiotemporal pattern of expression of most developmentally regulated genes is likely to be controlled by multiple elements. Our epigenetic analysis predicts that in almost all cases there are indeed multiple putative regulatory sites associated with genes that change expression, which can be in different chromatin states (*Figure 5A*). To illustrate the contributions of different chromatin elements, we generated a subnetwork for BRD8, which is located in a relatively gene sparse locus with only a few putative regulatory elements associated with it. The changes that occur as sites undergo epigenetic 'activation' or 'repression' (using Indices 1–7; *Figure 4A*) were modelled across the 12 hr time course. Since BioTapestry lacks a notation for discrete regulatory elements of individual genes, the subnetwork shows seven candidate *cis*-regulatory regions of *BRD8* (*BRD8_ site1-BRD8_site7* in *Figure 5B*) at 5, 9, and 12 hr represented as separate targets (*Figure 5B–D*). This depicts which specific elements contain binding sites for each regulating transcription factor. One element (*BRD8_site1* in *Figure 5B*) is initially acetylated; however, it does not seeem to receive inputs from other GRN components. This site is then methylated at 9 and 12 hr as a number of other elements become active, presumably to stabilize and maintain *BRD8* expression. In situ hybridization (*Figure 5E*) confirms that *BRD8* is not expressed in the pre-primitive-streak stage epiblast (EGKXII) or in the prospective neural plate at HH3-4. *BRD8* expression becomes enriched later in the neural plate and neural tube, as predicted by the network regulatory dynamics. This subnetwork illustrates the complexity of regulatory dynamics that underlie gene expression even at a relatively simple locus. Although it is not practical to represent the entire network including all the elements in their different states in this way using BioTapestry, all identified elements and their predicted activity are presented in the associated UCSC browser tracks (https://genome.ucsc.edu/s/stern_lab/Neural_ Induction_2021), together with the predicted binding sites relevant to the network contained within each site; a full list is also given in *Figure 4—source data 1*.

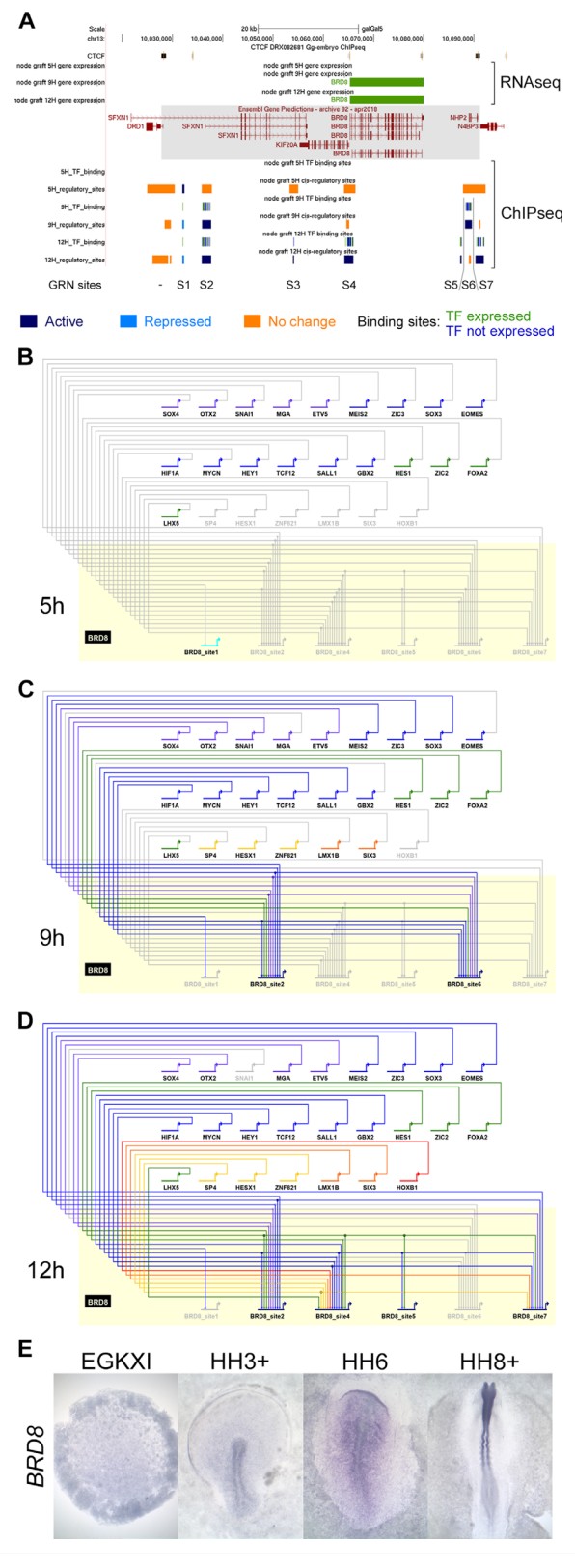

**Figure 5.** A subnetwork incorporating individual regulatory sites and their dynamics. (**A**) UCSC browser view of RNA-sequencing (RNAseq) and ChIPseq tracks associated with *BRD8*. *BRD8* is upregulated after 9 and 12 hr of neural induction (green bars). The *BRD8* regulatory locus (grey box; chr13:10028086–10091079) is defined by flanking CTCF-bound sites. Seven putative regulatory sites (S1-7) within this domain were predicted based on

*Figure 5 continued on next page*

*Figure 5 continued*

the ChIPseq H3K27 profiles. This includes sites that undergo activation (Indices 1–3, 7, coloured in dark blue), repression (Indices 4–6, coloured in cyan), or show no change (coloured in orange). Transcription factor binding sites by network components are shown; green for components that are expressed at the same time point and blue for those that are not. A BioTapestry subnetwork was generated from these predicted binding sites and expression profiles. Site 3, active at 12 hr, is not shown in the subnetwork as there is no TF that is expressed at 12 hr and predicted to bind to it. (**B**) *BRD8* regulation during neural induction: *BRD8* is initially not expressed; site 1 is active but is not predicted to be bound by other gene regulatory network (GRN) components. Each GRN component is coloured according to the time point when it first starts to express at 0 hr (purple), 1 hr (grey-blue), 3 hr (blue), 5 hr (green), 7 hr (yellow), 9 hr (orange), and 12 hr (red). (**C**) *BRD8* is upregulated after 9 hr; sites 2 and 6 undergo activation and could be bound by various transcription factors that are also expressed. Site 1 undergoes repression. (**D**) *BRD8* expression is maintained at 12 hr; regulators potentially bind to sites 2, 4, 5, and 7. Site 6 is no longer predicted to be active. (**E**) In situ hybridization detects BRD8 transcripts in the neural plate at HH6 and neural tube at HH8+, but not at earlier stages, EGKXI and HH3+.

## Prediction of core transcriptional regulators of the network

Genes with high 'outdegrees' and 'betweenness centralities' are often considered to be 'core genes' in a network. The 'outdegree' of a network component A is the number of outgoing interactions from component A to other components in the network (including component A itself), whereas the 'betweenness centrality' of a network component A is a measurement that captures the importance of component A based on the frequency at which interactions between pairs of other genes do so only by passing through component A (*White and Borgatti, 1994*). To predict key hubs in the network that may play more important roles during the process of neural induction, we calculated the 'outdegrees' and 'betweenness centralities' of network components across seven time points (0, 1, 3, 5, 7, 9, and 12 hr) (*Figure 6*). In the 0 hr network, *ESRRG*, *GRHL2,* and *TFAP2A* appear to be core genes in ectoderm cells prior to their exposure to a node graft (*Figure 6B*). Soon after grafting a node, these genes are immediately downregulated and repressed, with predicted repressive inputs from the core genes in the GRN. At the 1 hr time point following a node graft, *TFAP2C*, *SOX4*, and *BLIMP1* start to be expressed and are predicted to act as core genes, acting as repressors of the next layer of core genes in ectoderm cells (*Figure 6C–E*). Other predicted neural induction core genes, such as *SOX3*, *TCF12,* and *MYCN*, are highlighted as key transcriptional regulators acting throughout the process of neural induction at multiple time points (*Figure 6D–H*), and are predicted to bind to the enhancers of many genes in the GRN.

## Neural induction by a graft of Hensen's node recapitulates normal neural plate development

To what extent is the induction of an ectopic neural plate by an organizer graft comparable to the events of neural plate development in the embryo? To address this, we used three complementary approaches. First, we explored the spatial and temporal expression patterns of network components during normal embryo development. Then, we conducted scRNAseq analysis of stages of neural plate development concentrating on network components and their temporal hierarchical organization. Finally, we tested the activity of several of the putative regulatory elements in normal embryos to assess whether their activity in the neural plate in vivo resembles the hierarchy revealed by node grafts.

### Spatiotemporal expression of GRN components during normal neural plate development

A previous study identified genes whose expression differs between EGKXII-XIII (pre-primitive-streak) epiblast, neural plate at HH6-7, and non-neural ectoderm (*Trevers et al., 2018*). When these embryonic tissues (*Figure 7A*) are compared to neural induction, we find that the pre-streak epiblast is most similar to induced ectoderm after 5 hr of a node graft, while 9 and 12 hr time points are more closely related to mature neural plate (*Figure 7B*). This encouraged us to generate a detailed spatiotemporal map of transcripts during embryonic neural plate development. Genes that are differentially expressed in induced tissues compared to uninduced in RNAseq were selected. In situ hybridization was performed for 174 transcriptional regulators (including 123 that are represented in the GRN) at

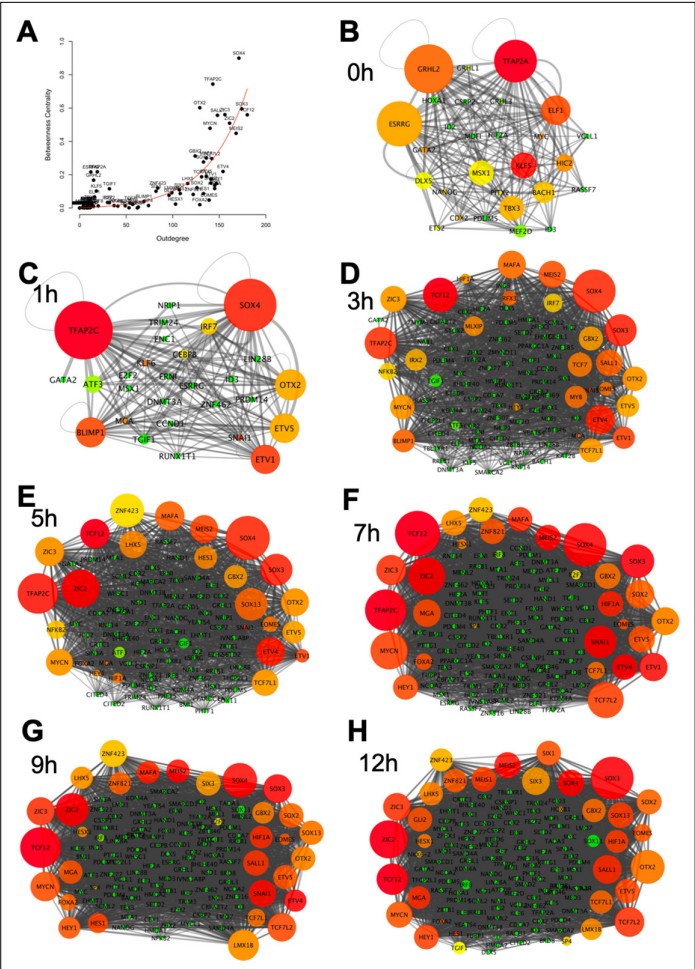

**Figure 6.** Identifying core regulatory factors in the neural induction gene regulatory network (GRN). (**A**) Correlation between the number of regulatory output (outdegree) and the degree of centrality of genes. (**B–H**) Network interactions at 0, 1, 3, 5, 7, 9, and 12 hr. The size of a node indicates the degree of the centrality of a gene. The degree of regulatory outputs is represented with the following colour scheme: red, orange, yellow, and green indicate high, medium, to low degree of regulatory outputs.

four stages: pre-streak (EGKXII-XIII), primitive streak (HH3-4), head process/early neural plate (HH5-6), and neural fold/tube (HH7-9).

There is a striking correspondence between neural induction by a node graft and normal neural development in terms of the spatiotemporal patterns we observed. In addition to confirming the expression of known neural markers (*ERNI*, *SOX3*, *SOX2*, and *SOX1*; *Figure 7C–F*), the predictions from node grafts uncover many novel responses such as *CITED4* (*ERNI*-like), *MAFA* (*SOX3*-like), *GLI2*, and *ZNF423* (*SOX2*-like) (*Figure 7G–J*). In total, 84/89 transcriptional regulators that are induced by a node graft were detected in neural tissues at some stage (*Figure 7* and *Figure 7—figure supplements 1–4*). Only 5 of the 89 genes (*HEY1*, *RFX3*, *STOX1*, *TAF1A,* and *ZIC2*) could not be detected by in situ hybridization – these are expressed at very low FPKM levels according to RNAseq. Most of the transcriptional regulators that are induced by a node graft (79/84) have not previously been associated with neural induction (*Figure 7—source data 1*). Likewise, a high proportion (65/85; 76%) of the transcripts that are downregulated after a node graft are depleted in neural tissues relative to non-neural territories. Among these are known non-neural markers such as *GATA2* and *MSX2* and the extraembryonic marker *HIF2A* (*Figure 7K, L and O*). Neural/non-neural border markers (such as *DLX5*, *MSX1*, and *TFAP2C*; *Figure 7M* and *Figure 7—figure supplements 5–8*) were also downregulated, supporting suggestions that neural induction involves the repression of an early border identity

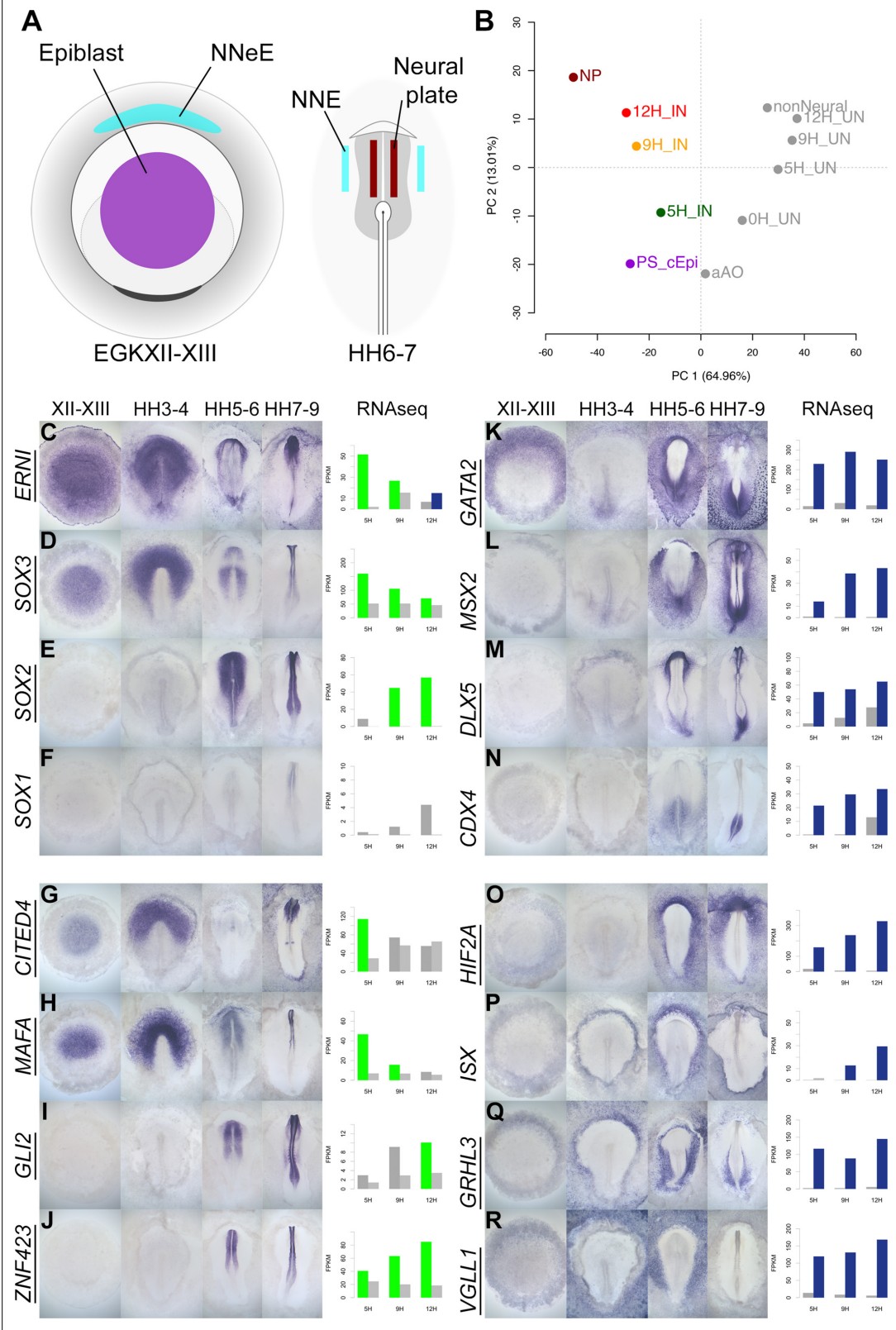

**Figure 7.** A spatiotemporal map of the normal embryo for transcripts differentially expressed during neural induction. (**A**) Central epiblast from pre-primitive-streak stage embryos (PS_cEpi) at EGKXII-XIII, neural plate (NP) at HH6-7, and corresponding non-neural extraembryonic ectoderm were dissected and processed by RNA-sequencing (RNAseq). (**B**) Principal component analysis comparing prospective neural and non-neural tissues from the normal embryo to induced (IN) and uninduced (UN) ectoderm 5, 9, or 12 hr after a node graft. (**C–R**) In situ hybridization of genes encoding

*Figure 7 continued on next page*

*Figure 7 continued*

transcriptional regulators at four stages of embryonic development (EGKXII-XIII, HH3-4, HH5-6, and HH7-9) compared to their RNAseq expression after 5, 9, and 12 hr of a node graft. Bar charts plotted RNAseq values (FPKM) in induced and uninduced tissues. Bars are shaded green when genes are upregulated in induced tissue (FPKM >10 in induced tissue and fold change (FC) >1.5 compared to uninduced). Bars are shaded blue when genes are downregulated (FPKM >10 in uninduced tissue and FC <0.5 compared to uninduced). When a gene is neither upregulated nor downregulated, the bars are coloured in dark grey in induced and light grey in uninduced. Genes represented within the neural induction GRN are underlined.

The online version of this article includes the following source data and figure supplement(s) for figure 7:

**Source data 1.** Sumary of changes for upregulated transcriptional regulators based on in situ expression patterns.

**Source data 2.** Summary of changes in downregulated transcriptional regulators based on in situ expression patterns.

**Source data 3.** Details of probes generated by digestion of cDNAs that were used for in situ hybridisation.

**Source data 4.** Details of probes generated by PCR that were used for in situ hybridisation.

**Source data 5.** PCR primer sequences.

**Figure supplement 1.** A spatiotemporal map in the normal embryo of transcripts that are differentially expressed during neural induction.

**Figure supplement 2.** A spatiotemporal map in the normal embryo of transcripts that are differentially expressed during neural induction.

**Figure supplement 3.** A spatiotemporal map in the normal embryo of transcripts that are differentially expressed during neural induction.

**Figure supplement 4.** A spatiotemporal map in the normal embryo of transcripts that are differentially expressed during neural induction.

**Figure supplement 5.** A spatiotemporal map in the normal embryo of transcripts that are differentially expressed during neural induction.

**Figure supplement 6.** A spatiotemporal map in the normal embryo of transcripts that are differentially expressed during neural induction.

**Figure supplement 7.** A spatiotemporal map in the normal embryo of transcripts that are differentially expressed during neural induction.

**Figure supplement 8.** A spatiotemporal map in the normal embryo of transcripts that are differentially expressed during neural induction.

(*Trevers et al., 2018*). We also uncover several other genes (such as *ISX*, *GRHL3*, and *VGLL1*) that are excluded from neural tissues (*Figure 7P–R*).

This approach reveals even subtle temporal changes in expression. For example, *CITED4* is sharply upregulated after 5 hr of a node graft but is later downregulated (*Figure 7G*). In the embryo, it is expressed in pre-streak epiblast and prospective neural plate at HH3-4, but CITED4 transcripts are then absent from the neural plate at HH5-6. *CDX4* (*Figure 7N*) and *HOXA2* (*Figure 7—figure supplement 6J*) are initially repressed by a node before their expression levels increase at 12 hr – just as they become enriched in the posterior neural tube from HH7-9. These observations confirm that transcriptional responses to a node graft closely follow the events that occur during development of the embryonic neural plate of the normal embryo.

## The GRN describes a temporal hierarchy occurring in single cells during neural development

To follow the expression of multiple genes during normal neural plate development and to compare this to the temporal hierarchy of our GRN, we assessed the expression of network components in individual cells at different stages of normal neural development. scRNAseq was performed on ectoderm dissected from chick embryos at HH4, HH6, HH8, and HH9+. These broad regions included prospective or mature neural plate and neural tube as well as non-neural ectoderm and neural plate border (*Figure 8A*).

The non-linear dimensionality reduction method UMAP was then used for unbiased cell clustering in low dimensionality space (*Figure 8B*). Cells collected from the two earliest developmental stages (HH4 and HH6) each form a distinct group, whereas cells from later stages (HH8 and HH9+) cluster primarily according to identifiable cell populations (*Figure 8C*). Cell identities were assigned to the 15 clusters using well-established markers (*Figure 8—figure supplement 1A*). This revealed groups corresponding to placodal (cluster 8), neural crest (clusters 10, 11), and neural tube identities (clusters 4, 6, 7, 9, 13, 14) within the HH8 and HH9+ cell populations. At HH6, neural plate cells expressing *LIN28B*, *SETD2,* and *SOX2* (cluster 0) are distinct from non-neural plate progenitors (cluster 3), which express *GATA2*, *DLX5*, and *TFAP2A*. At HH4, prospective neural cells express *MAFA*, *ING5*, and *LIN28B* (clusters 1, 2, and 5) whereas cluster 12 expresses the node marker ADMP.

To identify groups of genes that are co-expressed across the entire dataset, gene module analysis was performed using Antler (*Delile et al., 2019*). Cell state-specific modules were then selected by

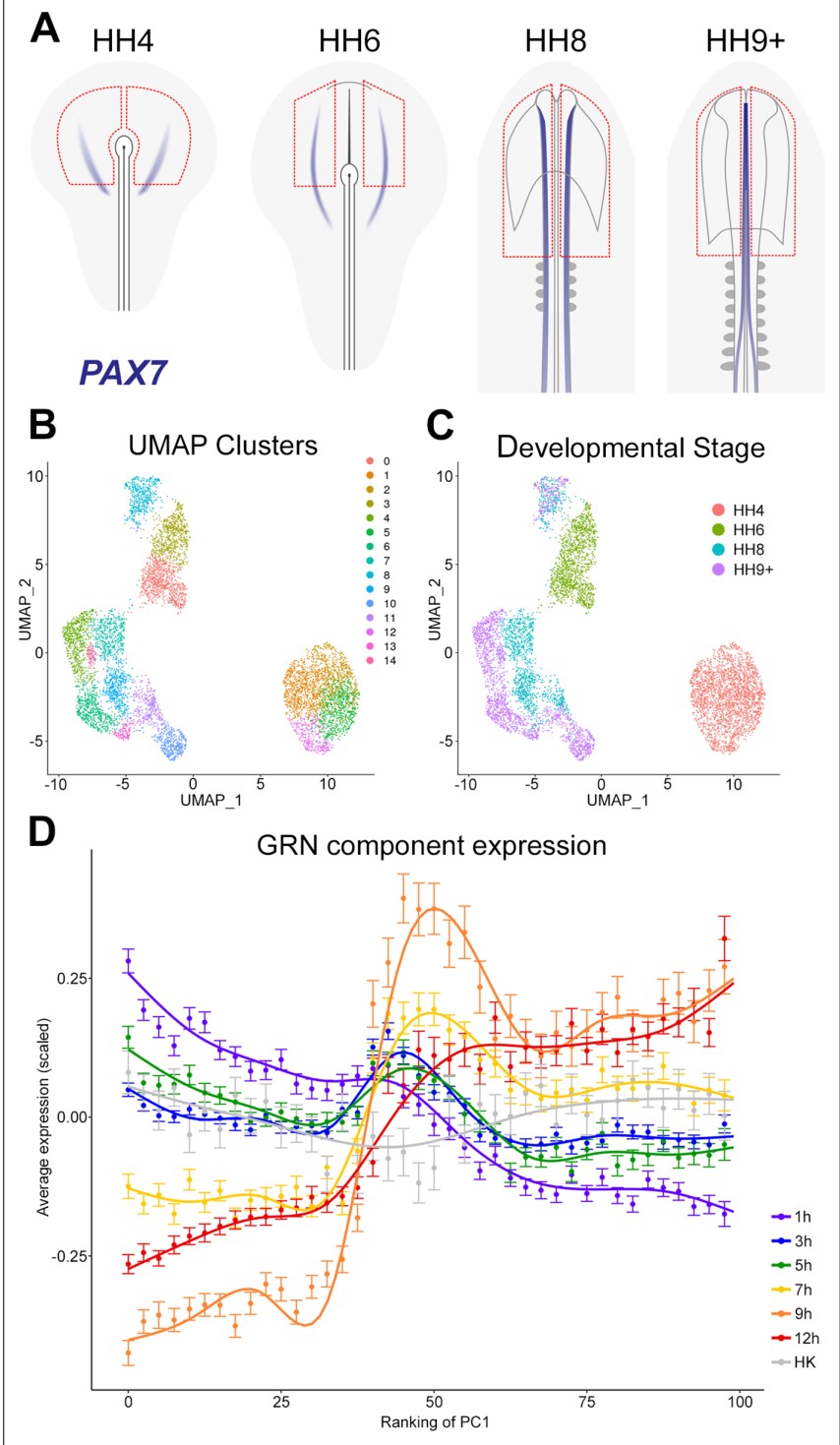

**Figure 8.** The gene regulatory network (GRN) describes a temporal hierarchy occurring in single cells during neural development. (**A**) Broad regions of prospective neural plate, neural plate border and non-neural ectoderm were dissected at HH4, HH6, HH8, and HH9+ and processed for single-cell RNA-sequencing (scRNAseq). *PAX7* expression marks the border between neural and non-neural tissues. (**B**) UMAP displaying unbiased clustering of cells into 15 clusters (0–14) representing distinct cell identities. (**C**) UMAP displaying the developmental stage from which the cells were collected; cells from HH4 and HH6 form independent clusters, whereas HH8 and HH9+ are more intermixed. (**D**) Temporal hierarchy of expression of GRN components in single cells during neural plate development. Prospective neural plate and neural tube cells (clusters 0, 1, 2, 5, 6, 7, 9, 13, and 14) were selected and ranked according to PC1 (see *Figure 8—figure supplement 1C*). The average gene expression (mean ± SE)

*Figure 8 continued on next page*

*Figure 8 continued*

is plotted for groups of components that are induced at each time point in the GRN, compared to housekeeping genes (HK). A general additive model was fitted to visualize the smoothed expression profile across the ranking of PC1 (using bins of 2.5).

The online version of this article includes the following figure supplement(s) for figure 8:

**Figure supplement 1.** Network components are co-expressed in single cells during neural development.

applying the criterion that at least 50% of the genes in a module must be differentially expressed (log fold change (FC) >0.25, FDR <0.001) in at least one cell cluster relative to the rest of the dataset. This defines 17 distinct modules (*Figure 8—figure supplement 1B*), which support our cell type classifications. Module-3 shows co-expression of *PAX2, WNT4,* and *NKX6-2* and is upregulated in prospective midbrain clusters 9 (at HH8) and 6 (at HH9+). Module-10 is defined by co-expression of *PAX7, SOX8, SOX10,* and *TFAP2B* and is associated with neural crest (clusters 10, 11). Module-25 has low levels of expression of *GATA2* and *ID3* (clusters 1, 2, 5, 12) at HH4, which are then further downregulated at later stages, except in non-neural cells at HH6 (cluster 3) and placodal cells at HH8 (cluster 8).

To explore the temporal hierarchy of GRN components, we assessed their expression in prospective neural plate and neural tube cell populations (clusters 0, 1, 2, 4, 5, 6, 7, 9, 13, 14). Within the 'neural' population, principal component (PC) analysis (PCA) revealed that PC1 inversely correlates with developmental stage (*Figure 8—figure supplement 1C*), therefore cells were ranked according to their position along PC1 to explore temporal expression dynamics. To visualize the expression of GRN components, genes were grouped according to the time point at which they are first induced by a node graft; their average (normalized and scaled) expression was then plotted (*Figure 8D*). This analysis reveals that components upregulated within 1–5 hr of a node graft are initially highly expressed in prospective neural cells and that their expression decreases progressively across PC1. GRN components that are induced after 7–12 hr are expressed at higher levels later, in cells of the maturing neural plate and neural tube. Therefore, GRN components identified from node graft experiments are expressed in a temporal hierarchy that closely replicates normal neural plate development.

## Responses to neural induction reveal enhancers that drive neural gene expression during normal development

Do the regulatory elements predicted by the GRN generated from Hensen's node grafts correspond to those that regulate normal development of the neural plate in the embryo? To define the putative enhancers more precisely, we combined our histone profiling (*Figure 2* and *Figure 2—figure supplement 1*) and ATACseq (*Figure 2—figure supplement 2*) to identify regions of open/active chromatin from induced and uninduced tissues after 5, 9, and 12 hr after a node graft. These regions were then compared to ChIPseq (for H3K27ac/me3) and ATACseq performed on pre-primitive-streak epiblast and mature neural plate at HH6-7.

We chose six putative enhancers for more detailed exploration. These were identified within the previously described CTCF loci using the overlapping combination of H3K27ac marks and ATACseq reads to predict enhancers that are likely to be active (*Figure 9—figure supplements 1–3*) based on the indices and expression profiles as described earlier. Each region was cloned into the pTK-EGFP vector (*Uchikawa et al., 2003*) to drive GFP expression via a minimal promoter; the reporter was then electroporated into the pre-primitive-streak stage (EGKX-XI) epiblast or a broad region including the prospective neural plate at HH3/3+ (*Figure 9A*) and embryos cultured for 5 hr or to the desired stages.

Each enhancer drives neural-specific GFP expression in a spatiotemporal pattern strikingly similar to the gene that lies in its proximity (*Figure 9B–O* and *Figure 9—figure supplements 1–3*). For example, SOX11_enh1 and SETD2_enh1 are enriched for H3K27ac in induced tissues after 5 hr (*Figure 9—figure supplement 1A–B*). Both are active in the epiblast at EGKXII-XIII and prospective neural plate at HH3-4, when *SOX11* and *SETD2* are expressed (*Figure 9—figure supplement 1* C-AY). SOX2_enh3 is 'poised' after 5 hr of neural induction, but acquires an active signature at 9 hr when it gains H3K27ac and ATAC reads (*Figure 9—figure supplement 2A*). As expected, this enhancer is not active in the prospective neural plate at HH4 (*Figure 9—figure supplement 2C–H*); GFP positive cells are detected later when *SOX2* is expressed in the neural plate proper at HH5-6 (*Figure 9—figure*

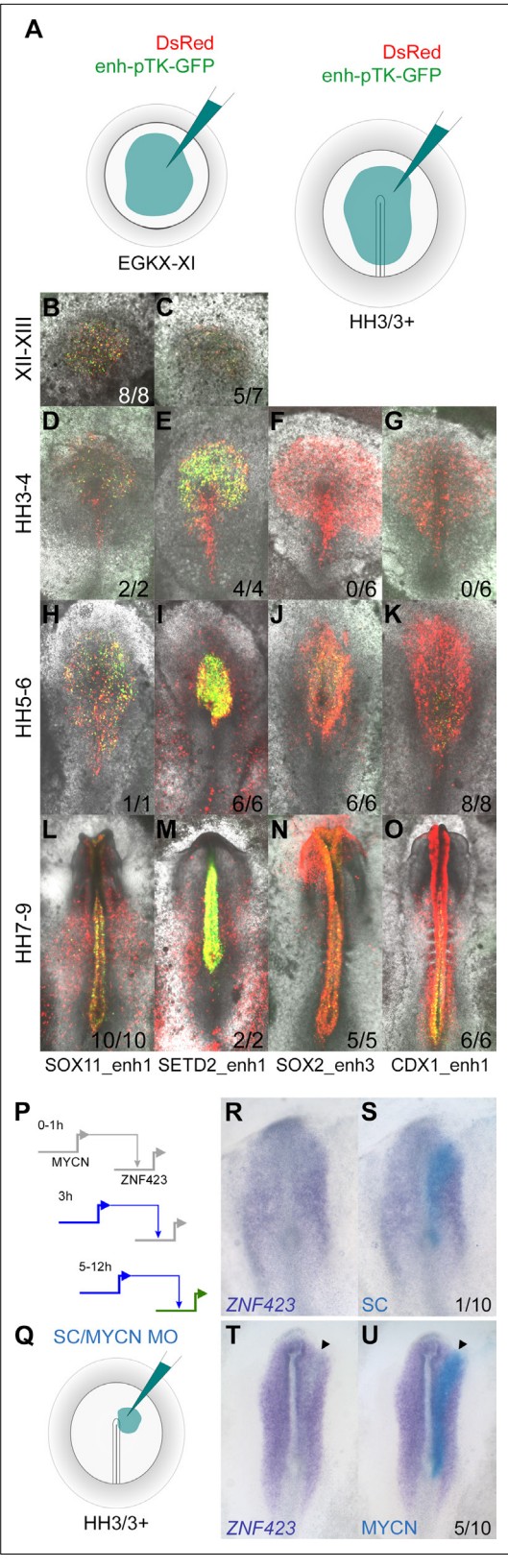

**Figure 9.** Enhancers identified by the neural induction assay have activity consistent with the expression of the gene during normal neural plate development. (**A**) Putative enhancers were cloned into the E-pTK-EGFP vector and co-electroporated into a broad region of the pre-streak stage (EGKX-XI) or HH3+ epiblast together with DsRed. (**B–C**) Activity of SOX11_enh1 and SETD2_enh1 at EGKXII-XIII; (**D–G**) activity of SOX11_enh1, SETD2_enh1,

*Figure 9 continued on next page*

*Figure 9 continued*

SOX2_enh3, and CDX1_enh1 at HH4; (**H–K**) HH5-6; (**L–O**) HH7-9. (**P–U**) Effect of knockdown of predicted network targets by electroporation of a morpholino (MO): a MYCN MO knocks down the expression of predicted target ZNF423, compared to a control MO.

The online version of this article includes the following source data and figure supplement(s) for figure 9:

**Source data 1.** Sequences of primers used for cloning enhancers.

**Figure supplement 1.** Detailed evidence of SOX11_enh1 and SETD2_enh1 activity.

**Figure supplement 2.** Detailed evidence of SOX2_enh3 and CDX1_enh1 activity.

**Figure supplement 3.** Detailed evidence of GLI2_enh1 and SIX3_enh2 activity.

---

*supplement 2I–N*). Likewise, GLI2_enh1 and SIX3_enh2 also become active after 9 hr; GLI2_enh1 is detected throughout the neural plate, whereas SIX3_enh2 is restricted to the forming forebrain at HH6 (*Figure 9—figure supplement 3A–Z*). These five enhancers remain active in the neural tube in correspondence with H3K27ac marks and ATAC reads that persist after 12 hr of a node graft.

CDX1_enh1 is also active in the caudal neural tube at HH7-9, when it also weakly labels some presomitic mesoderm cells. Its activity is detected at HH5-6, but only in a few cells of the caudal lateral epiblast adjacent to Hensen's node and not in the neural plate (*Figure 9—figure supplement 2U* -AL). This sparse activity overlaps closely with the discrete region from which neuro-mesodermal precursors (NMPs), which are said to express *CDX1* (*Gouti et al., 2017*), are thought to contribute to both neural and somite tissues (*Brown and Storey, 2000*; *Tzouanacou et al., 2009*). Since the *CDX1* gene is expressed earlier (HH3-4), but only in mesoderm and primitive streak, the activity of this enhancer is likely to be associated with NMPs.

Therefore, the domain of reporter activity for all six predicted enhancers corresponds to that of the normal expression of the associated gene. Furthermore, the stage at which the reporter first becomes active closely matches the stage at which the gene starts to be expressed during normal development.

Finally, to validate predictions from the network, we tested the interaction between MYCN (one of the core network genes; *Figure 6A*), which is upregulated from 3 hr, and its predicted target ZNF423, which is upregulated at 5 hr onwards (*Figure 9P*). Electroporation of a morpholino against MYCN into the prospective neural plate at HH3/3+ (*Figure 9Q*) knocks down the expression of ZNF423 (*Figure 9R–S*; 5/10 embryos), which is usually expressed later in the neural plate and neural tube (*Figure 7J*), compared to a control morpholino (*Figure 9T–U*; 1/10 embryos).

Taken together, these comparisons between node grafts and normal embryos strongly suggest that histone profiles that change in response to a node graft (upon which our network is based) can predict regulatory elements relevant to neural development. This establishes our GRN as a robust tool to describe the genetic hierarchy associated with the acquisition of neural identity.

## Browser resources

In addition to the BioTapestry GRN and other associated data described above, the results of this project, including data from node grafts in time course and from normal neural plate development, are made available on the UCSC browser at https://genome.ucsc.edu/s/stern_lab/Neural_Induction_ 2021. This incorporates RNAseq, ChIPseq, and ATACseq for all expressed and repressed genes (not limited to GRN components) and the annotated putative regulatory regions with predicted binding sites for GRN components. Finally, to help to predict potential conservation of the regulatory sites across species, we performed DREiVe analysis (*Sosinsky et al., 2007*; *Khan et al., 2013*; *Streit et al., 2013*) for domains within a 500 kb window around each of the genes represented in the GRN, using human (hg38), mouse (mm10), rat (rn6), golden eagle (aquChr2), zebrafish (danRer10), and chicken (galGal5). Blocks of sequence with conserved motifs are included as a track in the browser.

## Discussion

Although neural induction was first described a century ago, it has often been thought of almost as a discrete switch, changing the fate of cells as they respond to a signal or signals emanating from the 'organizer'. This study reveals a much greater complexity of responses to signals from the organizer, in very fine time course up to the time of appearance of mature neural plate markers. We show that

the hierarchy of responses to a graft of Hensen's node into a remote part of the embryo closely recapitulates the events of normal neural plate development, including not only the timing of gene expression in the developing neural plate, but also appropriate spatiotemporal activity of enhancers identified by the graft assay. Our GRN, comprising 175 transcriptional regulators and 5614 putative regulatory interactions between them, describes neural induction in unprecedented detail and allows us to predict the consequences of manipulating the activity of these components. This represents the most comprehensive example to date of a detailed network in a vertebrate embryo, modelled on both transcriptional and chromatin changes that take place in vivo, at high temporal resolution.

An important feature that distinguishes our approach from many others in the current literature is that we are able to monitor events in a known, precise time course relative to the time of exposure of the responding tissue to the organizer. This is only possible because of our experimental design: grafting the organizer into a region that does not normally give rise to the neural plate (but is able to do so in response to the graft) provides a clear 'time-zero', to act as reference. Thereafter, we have undertaken the time course at relatively short intervals, collecting transcriptional profiles at 0, 1, 3, 5, 7, 9, and 12 hr after the graft and epigenomic data at four of these time points (0, 5, 9, and 12 hr). It represents one of the most refined transcriptomic and epigenomic analyses of a developmental process studied in vivo and reveals the complexity of changes that can occur even over a relatively brief period. By performing our analysis on carefully dissected responding tissue (rather than a large portion or even a whole embryo), the resulting network should therefore be less subject to 'noise' from other events unrelated to the responses of competent cells to signals from the organizer. Since the fine temporal resolution reveals the 'direction' (e.g. increasing or decreasing) of expression changes at each time point, it is possible to identify concomitant changes in neighbouring non-coding chromatin and to predict their role in transcriptional activation or repression. Along with this, we can also observe changes in expression of transcription factors that have binding sites within those regions. These features make it possible to predict the GRN in a virtually unsupervised manner, using the workflow described.

## Is neural induction by a grafted organizer comparable to normal neural plate development?

Although a graft of Hensen's node is able to induce a neural plate, neural tube and their subsequent patterning into forebrain, midbrain, hindbrain, and spinal cord, as well as neural crest and placodes from the host epiblast of the area opaca, this is probably not how the process of central nervous system (CNS) development proceeds in the normal embryo. For a start, a large proportion of the cells that will form part of the neural plate (corresponding to the most cranial regions, forebrain, midbrain, and most of the hindbrain), never appear to be in close proximity to the node or anterior primitive streak. This implies that signals from tissues other than the node or streak are likely to play a role, especially at the earliest stages of neural development. We have shown that neural plate development is likely to start very early, before primitive streak formation, and probably under the initial influence of the hypoblast, involving FGF signals (*Streit et al., 2000*; *Trevers et al., 2018*). Therefore, although the node **can** induce an entire CNS including the earliest steps (as it also expresses FGF8), this is not what happens in normal development.

A second important issue arises by considering the formal definition of **induction**: '*an interaction between one (inducing) tissue and another (responding) tissue, as a result of which the responding tissue undergoes a change in its direction of differentiation*' (*Gurdon, 1987*). The 'change in the direction of differentiation', in more contemporary terms, might be described as a 'change of fate' or 'change in tissue identity'. By implication, to conclude that an inductive interaction takes place, we first need to define an inducing tissue (source of signals), the outcome of the induction (the final identity/fate resulting from the interaction) and importantly, a responding tissue whose normal fate, in the absence of induction, should **not** be the same as the outcome – otherwise there would be no 'change'. This means that it is not possible to determine whether the prospective neural plate of the normal embryo develops as a result of one or more inductive interactions, especially since we cannot determine the earliest stage at which the first inducing signals are presented to that responding tissue.

For these reasons, the time course initiated by a transplanted tissue that is able to induce a complete nervous system in a responding tissue whose fate is not to form a neural plate is an invaluable tool to study the acquisition of neural plate identity. Here, we have taken advantage of this approach by first

constructing a GRN with a fine time course in a competent responding tissue after such a transplant, defining the hierarchy and dynamics of expression of more than 150 transcription factors and other markers, along with changes in chromatin marks that accompany this. Then, we addressed the question of whether this hierarchy of events is in any way comparable to what occurs during normal neural development, using three complementary approaches:

First, using in situ hybridization for the majority of genes contained in the GRN, noting the time and location of their initial expression, we assessed whether markers upregulated by a node graft are expressed in the normal neural plate, and whether those repressed by a graft become excluded from the neural plate, and from what stage. This was then compared with the order of events after a node graft. We found a very strong correspondence: genes whose expression is induced by a node graft within 3 hr tend to be expressed in the pre-primitive-streak stage embryo (around stage EGK XIII), those induced at 5–7 hr in the central area pellucida anterior to and around the node at stages 4–6, and those induced at 9–12 hr are expressed in the elevating neural plate from stages 7–9, confirming previous observations made with just a few genes (*Figure 1* and *Figure 1—figure supplement 1*).

Second, we generated reporters for a subset of the predicted enhancers from the GRN, containing a minimal promoter (TK), the putative enhancer, and the coding sequence of GFP. This was co-electroporated into a wide region of the early embryo along with a vector encoding a fluorescent protein under the control of a strong ubiquitous promoter (DS-Red-Express) to mark the entire territory that had been electroporated. We then assessed the area within the electroporated territory (red from DS-Red) where the enhancer was active (green from GFP). In all six cases examined, the domain of expression corresponded to that of the normal expression of the associated gene, and the stage at which the reporter first became active closely matched the stage at which the gene starts to be expressed.

Finally, we compared gene expression of the neural plate/tube and the neural plate border in the normal embryo at single-cell level at four stages of development (primitive streak stage HH4, head fold stage HH6, four-somite stage HH8, and nine-somite stage HH9+; the time interval between each pair of these stages of normal development is approximately 2–3 hr). Dimensionality reduction and unsupervised clustering revealed clusters of cells representing neural plate and its different anterior-posterior parts, non-neural ectoderm, neural crest, and placodal cell states. Modelling transcriptional dynamics of components of the GRN across inferred pseudo-time revealed a very close correspondence between normal neural plate development and the sequence of transcriptional changes which take place following a node graft.

Taken together, these three approaches all strongly indicate that the sequence of events that characterize the response of area opaca epiblast cells to a graft of the organizer closely follows the dynamics of gene expression in the normal neural plate.

## The network – interpreting enhancer states

Our analysis reveals thousands of sites across the genome containing marks that change with the progression of neural induction. Each locus contains genes whose expression changes during this cascade, and their associated putative regulatory elements. As previously described for other systems such as embryonic stem (ES) cells in culture, regulatory elements may be decorated simultaneously by marks associated with activation (e.g. H3K27ac) and with repression (e.g. H3K27me3); these have been called 'poised' or 'bivalent' enhancers (*Cui et al., 2009*; *Heintzman et al., 2009*; *Xu et al., 2009*; *Creyghton et al., 2010*) and have been interpreted to indicate that the site is in transition from one state to the other, or perhaps able to go either way. Our fine time course reinforces this concept; in many cases we observe that cells pass from one state (e.g. H3K27me3 without acetylation) to the opposite, sometimes passing through a 'poised' state at the intermediate time point. This could suggest that these changes can occur sequentially in a cell. However, other interpretations are also possible. One possibility is that the apparent contradictory marks are due to different modifications of the two alleles of the locus in question, which could also be a transitional state. Another possibility is that this progression may reflect cells within a tissue that do not undergo the change synchronously, but some cells start earlier than their neighbours, the tissue gradually becoming more homogeneous. This sort of progression is sometimes seen by in situ hybridization or time-lapse movies of reporter activity (e.g. to visualize the activity of enhancer N2 associated with the SOX2 gene [Uchikawa et al., 2003; *Papanayotou et al., 2008*]). Likewise, in the present paper, SOX11 expression begins as a

mosaic (salt-and-pepper) in the early pre-streak epiblast before becoming more homogeneous, and the activity of the SOX11_enh1 (*Figure 9—figure supplement 1C–H*) shows a similar pattern at the same stage. Most likely these apparently 'poised' or 'bivalent' sites reflect a combination of these reasons.

Among the many regulatory elements predicted by our analysis are some that have been described previously (*Uchikawa et al., 2003*; *Uchikawa et al., 2011*; *Okamoto et al., 2015*; *Iida et al., 2020*). Their analysis was based on conserved stretches of non-coding sequence, followed by testing the activity of fragments in a vector containing a heterologous minimal promoter. In some cases, there are differences in the length of the sequence considered to act as an enhancer by both studies. For example, our SOX2_enh3 partially overlaps with, and its behaviour upon electroporation mimics the activity of, the previously identified *SOX2* D1 enhancer (*Okamoto et al., 2015*; *Iida et al., 2020*). Therefore, our ChIPseq analysis can verify known enhancers as well as identifying many new candidates.

Calculations of 'outdegrees' and 'betweenness centralities' of the components in the network across seven time points (0, 1, 3, 5, 7, 9, and 12 hr) reveals a relatively small subset of transcriptional regulators with high values in both parameters (*Figure 6*). These are defined as 'core genes' (or 'network hubs') in the network and may play more important roles during the process of neural induction. We tested whether one of these, MYCN, is required to regulate its predicted target, ZNF423 at the appropriate time point (*Figure 9R–S*) using morpholino electroporation. The knockdown reduced expression of the target gene in a significant proportion of embryos, confirming MYCN as a regulator of ZNF423 expression. However, it is worth considering that in the case of other enhancers such as the N2 element of SOX2, it appears that no single TF binding to the element is individually essential – rather, several TFs need to bind but the precise combination may not be so critical (*Uchikawa et al., 2003*; *Uchikawa et al., 2011*; *Okamoto et al., 2015*; *Iida et al., 2020*). This may be a general feature of such developmentally regulated elements, therefore testing whether a factor is 'essential' can give a misleading view of gene regulation in the biological context being studied.

## Can the network help to illuminate developmental concepts?

The GRN has great complexity over a relatively short period of time, with many transcription factors changing their expression and thousands of predicted interactions between them undergoing concomitant changes. We initially expected that a detailed view of the responses to signals from the organizer might somehow highlight specific changes, or states, that could be associated with some of the key concepts of developmental biology, such as 'induction' and 'commitment' (*Slack, 1991*; *Stern, 2004*) to a neural plate identity. These concepts have classically been defined based on specific embryological manipulations but it has hitherto not been possible to identify specific molecular signatures associated with these transitions. Could we use the present GRN to find objective molecular criteria that correspond to such changes in developmental properties?

Based on timed organizer grafts followed by their removal, a critical time was established at 12–13 hr after a graft (*Gallera and Ivanov, 1964*; *Gallera, 1965*; *Gallera, 1971b*; *Gallera, 1971a*), using similar conditions to those of our node grafts here. At this time point, removal of the node graft does not impair the subsequent development of the neural plate. This represents the endpoint of our GRN; at this time *SOX2* is robustly expressed, and *SOX1* is just starting to be expressed, in the responding tissue (*Pevny et al., 1998*). Although this functional assay does not test for 'irreversibility' of the cell fate transition, the finding that only after this point it can proceed autonomously without further signals from the organizer suggests that this time point might correspond to at least some degree of 'commitment' (or 'determination') to a neural plate identity.

Another earlier study has defined an initial state (named the 'common state') reflecting the earliest responses (1–3 hr) to a node graft into the area opaca (*Trevers et al., 2018*). A similar state is elicited by a graft of lateral head mesoderm (which induces placodes) and by a graft of the node (which induces the CNS), and the genes comprising this 'common state' have some similarity to those expressed by ES cells. In normal embryos, these genes are expressed in a broad region of the embryonic epiblast; moreover, a graft of the hypoblast into the area opaca also induces a similar set of genes. These observations suggested that the responses to inducing tissues start by a 'rewinding' (or reprogramming) cells to a state similar to that of the very early embryonic epiblast, from where cells can then be redirected to their new fate: neural plate, placode, neural crest, epidermis, and perhaps others. A similar state can also be elicited by exposure of area opaca to a source of FGF8 (a factor normally secreted

by both the hypoblast and the node, but lost by the node at the time when its inducing ability starts to decline; *Storey et al., 1992*; *Streit et al., 2000*).

Functional studies have established that competent ectoderm cannot respond to BMP inhibitors like Chordin unless it has previously been exposed to a graft of the organizer for at least 5 hr (*Streit et al., 1998*). This led to the identification of a few genes whose expression is associated with this transition (*Streit et al., 2000*; *Sheng et al., 2003*; *Gibson et al., 2011*; *Pinho et al., 2011*; *Papanayotou et al., 2013*), but the present experiments greatly expand and enrich this by placing genes in the context of a comprehensive network of gene interactions. Future experiments can take advantage of this information to determine the mechanisms that cause epiblast cells to become sensitive to BMP inhibition at this time point, and the upstream signals responsible.

To define possible 'key' steps, if they exist, we have attempted to identify 'core' factors that appear to represent important hubs in the GRN and their downstream targets and interconnections (*Figure 6*). This revealed a small number of genes that appear to occupy nodal, highly connected, positions in the network, and include, at successive time points, *ESRRG*, *GRHL2*, and *TFAP2A* in epiblast not exposed to a graft (0 hr) but then almost immediately downregulated, then *TFAP2C*, *SOX4,* and *BLIMP1* just 1 hr after a node graft. Indeed, *BLMP1* (*PRDX1*) has recently been shown to be required transiently at an early stage for neural plate, crest, and placode development, and its subsequent downregulation is also essential for further differentiation (*Prajapati et al., 2019*). Other later putative core genes include *SOX3*, *TCF12,* and *MYCN*. Future experiments could address whether these do indeed represent key steps accompanying neural induction. For the time being, we feel that functional definitions of the developmental concepts will remain key to defining the properties of developing tissues, especially given the complexity of the changes in gene expression accompanying neural induction.

## Does 'neural induction' occur in the normal embryo, and if so when and how?

The gene signature that defines the 'common state' is induced by a graft of Hensen's node after approximately 3 hr. The same genes are expressed in the epiblast of the normal embryo before primitive streak formation (stage EGK XIII), long before the node has formed; they can be induced by a graft of the hypoblast, as well as by the node. This suggests that in the normal embryo, it is the hypoblast rather than the node that initiates this process (*Foley et al., 2000*; *Streit et al., 2000*; *Trevers et al., 2018*). However, a node graft can mimic both these initial steps and those that follow. Therefore, although a node graft is able to generate a complete neural plate when grafted into a region of competent epiblast that normally does not give rise to this tissue, neural plate development in the normal embryo is likely to be directed by signals emanating from more than one tissue. A large part of the prospective CNS (forebrain, midbrain, and anterior hindbrain) is never close to the node but extends far anteriorly from the primitive streak. Indeed it has been suggested that the most anterior nervous system may be induced by signals other than the node, as well as earlier in development, and that the node and its derivatives may be involved mainly in the induction of the more caudal parts of the CNS (posterior hindbrain and spinal cord) (*Beddington, 1994*; *Thomas and Beddington, 1996*; *Foley et al., 2000*; *Streit et al., 2000*; *Metzis et al., 2018*). Armed with the GRN presented here, it will be particularly interesting in future to assess which sub-modules ('kernels') of the GRN are regulated by which secreted factors expressed in the node and whether and when similar factors are also normally expressed in other tissues that display partial or full neural induction activity.

## Neural induction in vivo and in vitro

A growing literature makes use of protocols to direct cultured mouse ES cells or human induced pluripotent stem cells into neurons, and calls this process 'neural induction'. After maintaining the stem cell population in a medium containing FGF to sustain proliferation and pluripotency, a common current protocol involves withdrawal of the FGF, followed by activation of the hedgehog pathway (*Gouti et al., 2014*; *Metzis et al., 2018*). Other protocols in the literature include BMP inhibition, manipulation of the Wnt pathway, and/or retinoid signalling, in various combinations. Almost all of these protocols need at least 5 days, and sometimes as long as 14 days, for the culture to generate neuronal cells. This is a much longer period than we see in vivo, which raises the question of to what extent they represent equivalent processes. Our network should help to compare them and to assess the degree of similarity as well as identify any differences. We have already shown that the initial

('common') state in response to induction (the first 3 hr or so in our assays) generates a state comparable to ES cells, suggesting that the remaining part of the differentiation protocols correspond to the period after this.

## Conclusion, and a resource to assess conservation among the vertebrates

In conclusion, our study provides the first GRN to represent the changes that accompany the process of neural induction after grafting an organizer to an ectopic site, and how it relates to normal neural plate development, in fine time course. We provide a comprehensive resource (deposited in the UCSC genome browser) that includes not only the network components (transcriptional regulators with predicted cross-regulatory interactions), but also a genome-wide view of all transcriptional changes, H3K27 acetylation/methylation, and chromatin conformation (ATACseq) during this process, from the time cells are initially exposed to neural inducing signals to the time at which definitive neural plate markers start to be expressed. We envisage that the GRN and the accompanying resource will be invaluable tools to study other important aspects of early neural development, such as the signalling inputs responsible for these changes, the molecular basis of the competence of cells to respond to neural inducing signals at different locations and stages, and aspects of neural patterning leading to regional diversification of the CNS. We also envisage that this resource will be useful to understand the process of neural induction not only in the chick but also in other vertebrate species, as well as in protocols for neural differentiation in ES cells in vitro. Indeed, we have noticed that out of 84 upregulated genes included in this network, the majority (74/84) are expressed in the prospective mouse neural plate (*Figure 7—source data 1*). These 84 genes include 69 not previously associated with neural induction in any species, of which the majority (56/69) are expressed in the future mouse neural plate. These observations suggest a high degree of evolutionary conservation. To aid in cross-species comparisons and possible studies on evolutionary changes that have occurred in neural induction, the browser resource also includes the results of computational prediction of conservation of the key GRN loci among several vertebrate species, using DREiVe, a computational tool designed to identify conserved sequence motifs and patterns rather than precise sequence conservation (*Sosinsky et al., 2007*; *Khan et al., 2013*; *Streit et al., 2013*).

## Materials and methods

### Chicken eggs and embryo culture

Fertile hens' eggs (Brown Bovan Gold; Henry Stewart & Co., UK), were incubated at 38°C in a humidified chamber to the desired stages. Embryos were staged according to *Hamburger and Hamilton, 1951*, in Arabic numerals or *Eyal-Giladi and Kochav, 1976*, in Roman numerals for pre-primitive streak (pre-streak) stages. Embryos were harvested and dissected in Pannett-Compton saline (*Pannett and Compton, 1924*) or 1× PBS, before being fixed in 4% paraformaldehyde (PFA) overnight at 4°C or prepared for ex ovo culture. All experiments were conducted on chicken embryos younger than 12 days of development and therefore were not regulated by the Animals (Scientific Procedures) Act 1986.

Chicken embryos were cultured ex ovo using a modified New culture method (*New, 1955*; *Stern and Ireland, 1981*) as previously described (*Voiculescu et al., 2008*). In brief, eggs were opened and the thin albumin was collected. Intact yolks were transferred to a dish containing Pannett-Compton saline. The vitelline membrane was cut at the equator while keeping the embryo central. The membrane was then slowly peeled away from the underlying yolk to keep the embryo attached. Membranes were placed on a watch glass with the embryo orientated ventral side up. A glass ring was positioned over the embryo and the edges of the membrane wrapped over the ring. This assembly was then lifted out of the dish and adjusted under a dissecting microscope while keeping the embryo submerged in a small pool of saline. The vitelline membrane was gently pulled taut around the glass ring before the excess was trimmed. Excess yolk was cleared from the membrane and embryo by gently pipetting a stream of saline and then replaced with fresh liquid, leaving the embryo on an optically clear membrane. At this point, embryos can be cultured as they are or with the addition of a node graft which were transferred to the ring and positioned while the embryo is submerged. To complete the culture, saline was removed from around and within the ring without disturbing the

embryo and grafts. The dry ring was then transferred to a 35 mm Petri dish containing a shallow pool of thin albumin. The edges of the ring were pressed down to prevent it from floating, leaving the embryo supported by a shallow bubble of albumin beneath the membrane. Completed cultures were incubated in a humidified chamber at 38°C for the desired length of time. After culture, embryos were fixed on the membrane with 4% PFA or submerged with ice-cold saline to allow tissue dissection.

## Neural induction assays

Neural induction assays were performed as previously described (*Stern, 2008*; *Streit and Stern, 2008*). For Hensen's node grafts, chick donors and hosts at Hamburger-Hamilton (HH) 3+/4- were used and New cultures incubated for 1, 3, 5, 7, 9, or 12 hr at 38°C in a humidified chamber. One or two nodes were grafted per embryo, placed contralaterally within the inner third of the area opaca, at or above the level of the host node. This region is competent to respond to neural inducing signals but only contributes to the extra-embryonic membranes and not the embryo proper (*Streit et al., 1997*). Nodes were grafted with their endodermal surface in contact with the host epiblast.

## RNAseq tissue collection and processing

For RNAseq, HH4- chick nodes were grafted to the area opaca of HH4- chick hosts. A single node was grafted to the left or right side per host, and embryos were cultured for 5, 9, or 12 hr. Node grafts were removed by submerging the embryo in saline while still attached to the membrane. Fine syringe needles (27 G or 30 G, BD Microlance) were used gently to lift the grafted tissue away from the underlying epiblast. Where grafts were firmly attached, 0.12% trypsin dissolved in saline was gently pipetted over the graft site to assist in removal. After dissection, trypsin activity was neutralized by treating embryos briefly with heat-inactivated goat serum (*Stern, 1993*), which was removed by washing with fresh saline. The induced epiblast directly beneath the graft, which appeared greyish and thickened, was dissected using syringe needles and mounted insect pins. Uninduced epiblast tissue from same position on the contralateral side was also dissected. Tissue samples per condition were pooled, frozen on dry ice, and stored at –80°C. At each time point (5, 9, or 12 hr) a total of 50 induced and contralateral uninduced pieces of tissue were collected. A further 50 pieces of uninduced tissue were collected from HH4- embryos that had not been cultured, representing a 0 hr control. Tissue samples for each condition were then pooled and lysed in 1 mL TRIzol (Invitrogen) for RNA extraction.

RNA sequencing was conducted by Edinburgh Genomics (formerly ARK Genomics; Roslin Institute, University of Edinburgh, UK). Total RNA was extracted and the quality was assessed using the Agilent 2100 Bioanalyzer – all samples registered an RIN value between 9.0 and 10.0. Libraries were constructed using the Illumina TruSeq mRNA library preparation kit. In brief, mRNAs were purified using oligo-dT conjugated magnetic beads before being chemically fragmented to on average 180–200 bp. Fragments were then transcribed using short random primers and reverse transcriptase to produce single-stranded cDNA, from which double-stranded cDNA was generated by DNA polymerase I and RNase-H. After synthesis, cDNA was blunt-ended and a single A-base added to the 3′ end. Sequencing adapters were ligated via a T-base overhang at their 3′ and these libraries were then purified to remove unincorporated adapters before being enriched by 10 cycles of PCR. Library quality was checked by electrophoresis and quantified by qPCR. The seven RNA libraries were sequenced over two lanes via 100-cycle, paired-end sequencing using the Illumina HiSeq 2000 system.

## Whole-mount in situ hybridization

Antisense riboprobes were generated from cDNA plasmid templates or Chick Expressed Sequence Tag (ChEST) clones (Source Bioscience). Templates were linearized by restriction digest (*Figure 7—source data 3*) or by PCR (*Figure 7—source data 4*) using M13 forward (5'-GTAAAACGACGGCCAG T-3') and M13 reverse (5'-GCGGATAACAATTTCACACAGG-3') primers (*Figure 7—source data 5*). The products were transcribed using SP6, T7, or T3 (Promega) and DIG-labelled nucleotides. DNA templates were digested using RNase-free DNase and the RNA probe was precipitated and resuspended in water. Working probes were diluted to 1 µg/mL in Hybridization (HYB) buffer and stored at –20°C.

Whole-mount in situ hybridization was performed as described previously (*Stern, 1998*; *Streit and Stern, 2001*), but omitting pre-adsorption of the anti-DIG-AP antibody (Roche 11633716001, used at 1:5000). Stained embryos were imaged from the dorsal perspective using an Olympus SZH10

Stereomicroscope with an Olympus DF PlanApo 1× objective and an Olympus NFK 3.3× LD 125 photo eyepiece. Images were captured using a QImaging Retiga 2000R Fast 1394 camera and QCapture Pro software as 24-bit, colour TIFF files at 300 dpi with dimensions of 1600×1200 pixels.

A spatiotemporal expression map was generated for genes that are differentially expressed after either 5, 9, or 12 hr of a node graft. These were selected from the *galGal3* and *galGal4* RNAseq analyses according to the criteria as described in '*Differential expression analysis*'.

## NanoString tissue collection and processing

Uninduced tissue or induced epiblast that had been exposed to node grafts was dissected as previously described. Four to eight pieces of tissue were collected in triplicate per condition. Tissues were dissected in ice-cold 1× PBS and collected on ice. All excess solution was removed using a fine needle and promptly processed by adding a total of 1 µL of lysis buffer from the RNAqueous-Micro Total RNA Isolation kit (Thermo Fisher). Tubes were immediately snap-frozen on dry ice and stored at –80°C.

NanoString experiments were run on the nCounter Analysis System using a custom codeset and following NanoString guidelines. The codeset consisted of 386 probes (*Figure 1—source data 3*) belonging to various categories. Transcriptional regulators that are upregulated (coloured red) or downregulated (coloured green) at 5, 9, or 12 hr were selected from RNAseq screening (details in '*Differential expression analysis*'). Other genes that respond to 5 hr exposure to a node graft (*Streit et al., 2000*; *Sheng et al., 2003*; *Gibson et al., 2011*; *Pinho et al., 2011*; *Papanayotou et al., 2013*) were also included. Probes were also designed against housekeeping genes (ACTB, GAPDH), markers of apoptosis and proliferation, transcriptional readouts of FGF, BMP, WNT, retinoic acid, Notch and Hedgehog signalling; epithelial, mesodermal, endodermal, neural plate border, pre-placodal, neural crest and Hensen's node markers, and standard positive and negative NanoString controls.

Tissue samples were processed using the NanoString master kit and following the manufacturer's instructions. Probes were hybridized to lysates overnight for 17 hr at 65°C before the reactions were transferred to the NanoString prep-station robot and processed using the 'high-sensitivity' program. Probe-target complexes were digitally counted from 600 fields of view on the NanoString Analyzer.

## Enhancer cloning and electroporation

Enhancers were identified based on changing active, inactive, or poised chromatin signatures over 5, 9, and 12 hr of neural induction. These were referenced against the 5, 9, and 12 hr ATACseq data to select putative enhancers with accessible chromatin. The primers used for cloning are detailed in *Figure 9—source data 1*.

Enhancers were cloned from chick gDNA as previously described (*Chen and Streit, 2015*). The correct sized fragments were amplified by PCR using PCRBio Ultramix (PCRBiosystems) and ligated into the pTK_JC_EGFP plasmid (*Chen and Streit, 2015*) using T4 ligase (NEB). Ligation products were transformed into DH5α *Escherichia coli* and grown on agar plates with Ampicillin selection. Clones were screened by colony PCR using PCRBio Ultramix and pTK_Fwd (5'-GTGCCAGAACAT TTCTCTATCG-3') and pTK_Rev (5'-GTCCAGCTCGACCAGGATG-3') primers. Positive clones were cultured overnight in 5 mL LB broth plus Ampicillin and the constructs were purified by mini-prep (Qiagen). Enhancer inserts were sequenced using Citrine_Fwd (5'-TGTCCCCAGTGCAAGTGC-3') and Citrine_Rev (5'-TAGAACTAAAGACATGCAAATATATTT-3') primers. Plasmids with sequence verified inserts were maxi-cultured in 200 mL, of which 1 mL used to make a 50% glycerol stock and stored at –80°C. Bulk plasmid DNA was extracted using the Endofree Plasmid Maxi-prep kit (Qiagen) following the manufacturer's instructions and resuspended in 100 µL of sterile water. Further purification was conducted to reduce residual salts which interfere with electroporation efficiency and embryo damage. Samples were spun for 15 min/15,000 rpm/4°C to pellet any particulates. The supernatant was removed to a fresh tube, to which 10 µL of 3 M NaOAc pH 5.2 and 250 µL of 100% EtOH was added to precipitate the plasmid DNA. This was pelleted by centrifugation at 15 min/15,000 rpm/4°C and the pellet was washed with 70% EtOH and spun again for 10 min/15,000 rpm/4°C. The supernatant was removed and the pellet allowed to air-dry completely, before being resuspended in 50 µL of sterile water and the concentration checked by nanodrop and adjusted to 5 mg/mL.

Enhancer plasmids were co-electroporated with pCMV-DsRed-Express-N1 (Clontech) which labels all electroporated cells. Tissues were electroporated using an anode electroporation chamber and wire cathode according to *Voiculescu et al., 2008*, with minor modifications. Electroporation mix

was made fresh by combining the following in water: enhancer_TK_EGFP plasmid (2 mg/mL), pCMV-DsRed-Express-N1 (1 mg/mL), 6% sucrose, 0.02% Brilliant Blue FCF. This was backloaded into a capillary needle with a fine tapered tip. The anode chamber was filled with 1× PBS and the electrodes positioned ≈2 cm apart.

Harvested embryos were positioned directly between the electrodes with the dorsal side (epiblast) upwards, closest to the cathode. Electroporation mix was applied using a mouth aspirator attached to the capillary needle, and by bringing the needle tip in close contact with the embryo. For pre-streak EGKX-XI embryos, electroporation mix was applied broadly across the central epiblast region and immediately electroporated using the following current settings: 3.5–4.0 V, 5 pulses, 50 ms duration and a 100 ms gap. At HH4- stages, electroporation mix was applied broadly across the prospective neural plate and primitive streak and immediately electroporated using 4.5–5 V, 5 pulses, 50 ms duration and 100 ms gap. After electroporation embryos were transferred to vitelline membranes, prepared for New culture and cultured to the desired stages (*Voiculescu et al., 2008*).

Enhancer activity was imaged using a Leica SPEinv microscope with a 5× objective and LAS-X software. Images were captured at 2048×2048 pixels, speed 400 ms, and 1× zoom with an averaging of 4. EGFP was excited using a 488 nm laser and detected with a 492–543 nm wavelength filter. DsRed was excited using a 532 nm laser and detected with a 547–653 nm wavelength filter. Files were saved as.lif format and composite images were stitched together in Fiji (*Schindelin et al., 2012*) using the Pairwise Stitching plugin (*Preibisch et al., 2009*).

## Morpholino electroporation

Fluorescein-labelled morpholinos at 500 µM (MYCN: TCTTGCTGATCATTCCCGGCATGGC, or Standard control: CCTCTTACCTCAGTTACAATTTATA) were co-electroporated with pCMV-DsRed-Express-N1 (1 mg/mL) as a carrier. Constructs were targeted to the lateral prospective neural plate of embryos at HH3/3+ and electroporated using the following conditions: 4.0 V, 5 pulses, 50 ms duration and 100 ms gap. Embryos were then cultured overnight for 12 hr to reach stages HH5-8.

In situ hybridization was performed (as described above) to reveal ZNF423 expression using an anti-DIG-AP antibody and NBT/BCIP. Embryos were then post-fixed overnight in 4% PFA. They were then washed extensively in PBST and incubated at 70°C for 1 hr. Further PBST washes were performed to remove residual traces of NBT-BCIP. Immunohistochemistry was then performed to detect fluorescein-labelled morpholinos using anti-FITC-AP (Roche 11426338910, used at 1:5000) and BCIP alone.

## ChIPseq tissue collection and processing

In brief, tissues were dissected in ice cold 1× PBS and transferred to low binding PCR tubes containing 50–100 µL of 1× Protease Inhibitor Cocktail in 1× PBS (cOmplete mini EDTA-free; Roche) on ice. Where necessary, 0.12% trypsin or 1× PBS + 200 mM EDTA was used to aid dissection. Treated tissues were rinsed with PBS to remove residual trypsin. Tissues were collected in batches every 30 min before tubes were spun at 100 × *g*/3 min/4°C to gently pellet the tissue. Tubes were snap-frozen in liquid nitrogen before storage at –80°C for up to 6 months. For pre-streak EGK XII-XIII embryos, the hypoblast was removed and central epiblast was collected from six embryos in triplicate. After removal of the underlying mesoderm and endoderm, the neural plate at HH6-7 was harvested from 20 embryos in triplicate. For 0, 5, 9, and 12 hr node induced and contralateral uninduced area opaca, 14–30 pieces of tissue were collected in triplicate. A further 60 pieces of 9 and 12 hr uninduced tissue were collected and processed separately after it was discovered that 30 pieces of uninduced tissue at these time points did not generate enough material for ChIP.

ChIP was performed by micrococcal nuclease (MNase) digestion to fragment the chromatin following a previous protocol (*Brind'Amour et al., 2015*), followed by immunoprecipitation according to the Low Cell ChIPseq kit (Active Motif, 53084) with some modifications. Corresponding induced and uninduced samples were processed side-by-side, each as three reactions: H3K27ac, H3K27me3, and input. In brief, 200 µL of protein G agarose beads were added to 2×1.5 mL low binding tubes and spun for 3 min at 1250 × *g* and 4°C. The supernatant was removed and the beads were washed with 690 µL of TE buffer at pH 8.0. These tubes were spun again for 3 min at 1250 × *g* and 4°C and the supernatant removed before bead blocking reactions were set up. To one tube, 200 µL of TE buffer pH 8.0, 20 µL of blocking reagent AM1 and 20 µL of BSA were added. This tube of 'pre-clearing' beads was incubated for 3–6 hr on a 4°C rotator. To the second tube, 180 µL of TE buffer pH 8.0, 20 µL

of blocking reagent AM1, 20 µL of BSA, and 20 µL of Blocker were added. This tube of 'IP' reaction beads was incubated overnight on a 4°C rotator.

While beads were blocking, tissues were prepared for nuclei isolation and MNase digestion. A pair of samples (induced and contralateral uninduced) were thawed on ice for 5–10 min and the tissues pooled into one tube per condition using low binding tips. Pooled samples were spun at 100 × $g$/3 min/4°C and the excess supernatant was removed without disturbing the tissues, to leave ≈5 µL remaining. To each tube, 20 µL of nuclei lysis buffer was added, containing 16 µL of nuclei isolation buffer (Sigma N3408), 2 µL 1× protease inhibitor (Roche cOmplete mini, EDTA-free) and 2 µL of 1% IGEPAL CA-630. Tubes were gently tapped to mix and incubated on ice for 5 min. Nuclei were dissociated gently by pipetting 15 or 30 times without making bubbles, then incubating samples on ice for 15 min before repeating these steps four to five times as listed in *Figure 2—source data 1*.

Chromatin was then fragmented using MNase. This was carried out in two steps to fragment the easily digestible chromatin and then re-digest the more compact chromatin in a second longer digestion. A 3× master mix containing 169.5 µL of sterile $H_2O$, 30 µL of 10× MNase buffer (NEB), 30 µL of 50% PEG 6000, and 6 µL of 100 mM DTT (made fresh) was prepared. In a separate tube, 3 µL of 10× MNase buffer (NEB M0247) was diluted in 25 µL of $H_2O$, before adding 2 µL of MNase, to give a 1:15 dilution of MNase enzyme. Of this, 4.5 µL of MNase 1:15 was added to the 3× master mix, to give a total volume of 240 µL. Then, 50 µL of 3× MNase master mix was added to each tube containing 25 µL of isolated nuclei. The mixture was pipetted 15 times to mix without making bubbles, while tubes were kept on ice. This mixture was transferred to a fresh 0.2 mL low binding PCR tube at room temperature and the digestion incubated at room temperature for 2.5 min. This tube was returned to ice and 6 µL of 200 mM EDTA added. Tubes were spun at 500 × $g$ and 4°C for 5 min to pellet larger undigested chromatin and the supernatant (containing smaller, readily digestible chromatin) was transferred to a fresh 1.5 mL low binding microfuge tube. A further 30 µL of the MNase master mix was added to the remaining chromatin pellets on ice and resuspended by pipetting 60 times, before tubes were incubated for 6 min at room temperature. To break up remaining nuclear debris, 4 µL of 200 mM EDTA and 4 µL of 1% TX-100 +1 % sodium deoxycholate were added and the mixture pipetted 30 times. All 38 µL of this second digestion was collated with the first supernatant and kept on ice (total volume ≈110 µL).

Once the 'pre-clearing' beads were ready (after 3–6 hr) they were spun for 3 min/1250 × $g$/4°C. The supernatant was removed and the beads washed with 665 µL of ChIP Buffer (from Active Motif kit) by inverting the tube several times. Beads were spun again for 3 min/1250 × $g$/4°C and supernatant removed again, before 232 µL of ChIP buffer was added. An even suspension of 100 µL of beads was added to each tube of MNase digested chromatin (≈110 µL). Then, 10 µL of proteinase inhibitor cocktail, 10 µL PMSF (from the Active Motif kit), and a further 290 µL of ChIP buffer were added, giving a total volume of 520 µL. Tubes were then placed on a rotator at 4°C for 3 hr to pull down chromatin fragments that bind non-specifically to the protein G agarose beads.

After 'pre-clearing', samples were spun at 3500 rpm and 4°C for 3 min and the supernatant was divided into three separate 0.5 mL low binding PCR tubes: 200 µL each into H3K27ac and H3K27me3 tubes and the rest into the input tube (≈60 µL). To the experimental tubes, 2 µL of H3K27ac (Active Motif 39133, lot# 31814008) or H3K27me3 (Active Motif 39155, lot# 31814017) antibody was added, and these were incubated overnight at 4°C. To the input, 80 µL of low EDTA TE buffer was added and stored overnight at 4°C.

The following day, ChIP reaction tubes were spun briefly at 4°C to collect the solution. 'IP' reaction beads were removed from the rotator and mixed to an even suspension by pipetting with a wide bore tip. Keeping the beads well mixed, 50 µL of bead slurry was added to each immunoprecipitation and the tubes returned to the 4°C rotator for 3–4 hr to pull down ChIPed chromatin. Next, samples were washed according to Section G of the Active Motif Low Cell ChIPseq manual and eluted with 100 µL of Elution Buffer AM4.

Chromatin was de-crosslinked by incubating ChIP and input samples at 65°C for 2 hr. Then, 125 µL of phenol:chloroform:isoamyl alcohol 25:24:1 and 64 µL of chloroform:isoamyl alcohol mixture was added and samples shaken vigorously for 15 s before centrifuging at 16,000 × $g$/15 min/4°C. The top aqueous layer was transferred to a fresh low binding tube. An additional 100 µL of nuclease free water was added to the original tube, which was shaken and centrifuged again. Then the aqueous top layer was removed and added to the first. To this, 2 µL of blue glycogen and 20 µL of 3 M NaOAc pH 5.2

were added and samples flicked to mix. Finally, 660 µL 100% ethanol was added, and tubes flicked to mix well before precipitating overnight at –20°C. The next day, the chromatin was centrifuged at 16,000 × *g*/30 min/4°C. Supernatant was removed and the pellet washed with 900 µL of cold 70% EtOH. The pellet was spun at 16,000 × *g*/30 min/4°C and washed with 70% EtOH again, a total of four times. Then, chromatin pellets were air-dried completely for 10–15 min and resuspended in 20 µL of TE low EDTA buffer overnight at 4°C, before being stored at –20°C until library preparation.

## ChIPseq library preparation

Libraries were prepared using the Next Gen DNA Library kit (Active Motif 53216) and Next Gen Indexing kit (Active Motif 53264) but using ProNex size-selective beads (Promega) to enrich for mono- and di-nucleosomes (≈200–500 bp). Each sample of ChIPed chromatin was thawed on ice, to which 20 µL of low EDTA TE Buffer was added. For each library, 'Repair 1' reaction mix was prepared according to the Active Motif manual by combining 13 µL of low EDTA TE, 6 µL of Buffer W1, and 1 µL of Enzyme W2. This was added to the 40 µL of chromatin and mixed by pipetting. Reactions were incubated for 10 min at 37°C in a thermocycler with the lid open. To clean up the reaction, 180 µL (3× volume) of evenly suspended ProNex beads were added and incubated for 10 min at room temperature before placing on a magnetic rack for 5 min. The supernatant was removed and beads were washed three times with 100 µL of wash buffer, for 30–60 s each. The final wash buffer was removed and the beads were air-dried for 5 min. Next, Repair II reactions were prepared by combining 30 µL of low EDTA TE, 5 µL of Buffer G1, 13 µL of Reagent G2, 1 µL of Enzyme G3, and 1 µL of Enzyme G4. This was added to dry beads and pipetted to mix. Tubes were incubated on a thermocycler for 20 min at 20°C with lid open. Next, 90 µL (1.8× volume) of PEG NaCl was added to each reaction and mixed by pipetting 15 times. Samples were incubated at room temperature for 10 min and then on the magnetic rack for 5 min and beads washed and air-dried as before. Next, Ligation I mix was prepared by combining 20 µL of low EDTA TE, 3 µL of Buffer Y1, and 2 µL of Enzyme Y3, which was added to each tube of dry beads and mixed by pipetting. To each sample, 5 µL of Y2 index was added to uniquely barcoded libraries. Reactions were mixed by pipetting 60× and incubated in a thermocycler for 15 min at 25°C with the lid open. To clean up, 39 µL (1.3×) of PEG NaCl was added to each tube and mixed by pipetting 15×. Samples were incubated for 10 min at room temperature and 5 min on a magnetic rack, before beads were washed and air-dried as before. Ligation II mix was prepared by combining 30 µL of low EDTA TE, 5 µL of Buffer B1 2 µL of Reagent B2-MID, 9 µL of Reagent B3, 1 µL of Enzyme B4, 2 µL of Enzyme B5, and 1 µL of Enzyme B6. This was added to the dried beads and mixed by pipetting 30×, before incubating in a thermocycler for 10 min at 40°C with the lid open. Then, 55 µL (1.1×) of PEG NaCl was added and mixed by pipetting 30×. Samples were incubated for 10 min at room temperature and 5 min on a magnetic rack before beads were washed and air-dried as before. DNA was eluted by adding 50 µL of elution buffer and pipetting 15×. Tubes were incubated tubes at room temperature for 5 min, and then on a magnetic rack for 5 min. The eluate was transferred to a clean tube and the beads washed with a further 10 µL of elution buffer. This was collected and combined with the first 50 µL of eluate.

DNAs were size-selected using 1.2×/0.4× ratios of ProNex beads to purify ≈200–500 bp fragments. First, 72 µL of ProNex beads were added to the eluate and mixed by pipetting 30×. Samples were incubated for 10 min at room temperature and for 5 min on a magnetic rack. The supernatant (containing fragments <500 bp) was transferred to a clean tube. Then, a further 24 µL of ProNex beads were added to this supernatant and mixed 30× by pipetting. This was incubated for 10 min at room temperature followed by 5 min on a magnetic rack. The supernatant was removed and beads (with DNA fragments >200 bp bound) were washed three times with 100 µL of wash buffer, each time soaking beads for 30–60 s. The final wash buffer was removed and beads air-dried for 5 min. DNA was eluted by adding 10 µL of low EDTA TE, pipetting 10× to mix, and incubating for 10 min at room temperature. After 5 min on a magnetic rack, the eluate was transferred to a fresh tube and the beads incubated for a further 10 min with 10 µL of TE. This was collected and combined with the first collection to give a total of 20 µL.

Libraries were amplified by adding amplification mix containing 10 µL of low EDTA TE, 5 µL of Reagent 1, 4 µL of Reagent 2, 10 µL of Buffer R3, and 1 µL of Enzyme R4 to each tube. This was mixed by pipetting and incubated on a thermocycler using the following conditions: Denature1 – 98°C for

30 s, Denature2 – 98°C for 10 s, Anneal – 60°C for 30 s, Extension – 68°C for 60 s, Repeat to Denature2 for total 13 cycles, Final extension – 68°C for 60 s, Soak – 4°C forever.

Libraries were purified by adding 75 µL (1.5× volume) of ProNex beads and pipetting 60×. Samples were incubated at room temperature for 10 min, and 5 min on a magnetic rack. The supernatant was discarded and beads washed three times with 100 µL of wash buffer, before being air-dried for 5 min. Libraries were finally eluted in 20 µL of TE and their quality checked by Tapestation. They were sequenced by 75 bp single end reads to a depth of ≈20 million reads per library using the NextSeq 500 system.

## ATACseq tissue collection and processing

OMNI-ATACseq was conducted according to *Corces et al., 2017*, with some adjustments, which was in turn based on *Buenrostro et al., 2013*; *Buenrostro et al., 2015*. In brief, 0, 5, 9, and 12 hr node induced and contralateral uninduced tissues were dissected in ice-cold 1× PBS and transferred to low binding PCR tubes containing 1× Protease Inhibitor Cocktail in 1× PBS (cOmplete mini EDTA-free; Roche) on ice. Where necessary, 0.12% trypsin was used to aid dissection, and treated tissues rinsed with 1× PBS to remove residual trypsin activity. Tissues were collected in batches every 30 min before tubes were spun at 100 × *g*/3 min/4°C to gently pellet tissues. Tubes were snap-frozen in liquid nitrogen before storage at –80°C for up to 4 months. At each time point, 14–22 pieces of tissue were collected in duplicate.

Once sufficient tissues were collected, they were thawed on ice and the nuclei isolated in 2 mL of 1× Homogenization Buffer Unstable (HBU) with a Dounce homogenizer on ice. HBU 1× contains 5 mM CaCl$_2$, 3 mM Mg(Ac)$_2$, 10 mM Tris pH 7.8, 320 mM sucrose, 0.1 mM EDTA, 0.1% NP-40, 0.02 mM PMSF, and 0.3 mM β-mercaptoethanol, dissolved in H$_2$O. Tissues were ground using 15 strokes with the loose pestle and 20 strokes with the tight pestle. Homogenate was then passed through a 30 µm nylon mesh filter into a fresh 2 mL low binding microfuge tube. Next, nuclei were gently pelleted by centrifugation for 10 min/500 × *g*/4°C and the supernatant removed. The pellet was resuspended in 200 µL of 1× HBU by gentle pipetting. Nuclei were counted by taking 6×10 µL homogenate samples, staining them with DAPI and viewing on a haemocytometer. After counting, nuclei were washed with 1 mL of ATAC-RSB + 0.1% Tween-20 and spun for 10 min/500 × *g*/4°C to pellet. The supernatant was carefully removed and the nuclei pellet was resuspended in 50 µL of transposition mix by pipetting. The volume of Tn5 transposase (Illumina, 20034197) added to each reaction was scaled to 0.1 µL per 1000 nuclei, so that ATAC samples with different numbers of nuclei were treated equivalently. Therefore, transposition mix contained: 25 µL of 2× TD Buffer, 0.5 µL of 1% Digitonin, 0.5 µL of 10% Tween-20, 16.5 µL of 1× PBS, X µL of Tn5 transposase (0.1 µL of Tn5 per 1000 nuclei), topped up with sterile H$_2$O up to 50 µL. Transposition reactions were incubated for 30 min at 37°C in a thermomixer with 1000 rpm shaking. Immediately after samples were purified using the Zymo DNA Clean and Concentrator-5 Kit, eluted using 21 µL of elution buffer and stored at –20°C until library preparation.

Libraries were prepared exactly as described (*Corces et al., 2017*), using the adapter sequences listed in *Figure 2—source data 2* (*Buenrostro et al., 2013*). After qPCR amplification, qPCR profiles were manually assessed to determine the additional number of cycles to amplify according to *Buenrostro et al., 2015*. Libraries were purified using the Zymo DNA Clean and Concentrator-5 Kit and eluted in 11 µL of H$_2$O. Concentrations and banding profiles were checked by Qubit and Tapestation. Sequencing (50 bp paired-end) was conducted by Oxford Genomics to ≈15–20 million reads –sufficient for open chromatin profiling of the chick genome.

ATACseq was also performed on pre-primitive-streak stage epiblast dissected from EGKXII-XIII embryos and on mature neural plate dissected from HH6-7 embryos (15–20 pieces of tissue per sample). Samples were collected in 1× Protease Inhibitor Cocktail in 1× PBS (cOmplete mini EDTA-free; Roche) and dissociated using a Dounce homogenizer. ATACseq was performed as described (*Buenrostro et al., 2013*; *Buenrostro et al., 2015*).

## Sample preparation for scRNAseq

Fertilized chicken eggs (Henry Stewart & Co. Ltd, Norfolk UK) were incubated at 38°C. The embryos were removed from the egg, staged, and pinned ventral side up on a resin plate in Tyrode's saline. Embryos were dissected with a G-31 syringe needle. The endoderm and mesoderm were removed with a small volume of dispase (10 mg/mL) before dissecting an ectodermal region spanning the

neural plate and neural plate border (*Figure 8A*). Tissue samples were pooled from multiple embryos of the same stage prior to dissociation (HH4=~55 embryos; HH6=~45 embryos; HH8=10 embryos; HH9+=7 embryos). For cell dissociation, samples were incubated in FACSmax cell dissociation solution (Amsbio, T200100) with 30 U/mL Papain (Sigma, P3125) at 37°C for 20 min. The tissue was pipetted 10 times every 5 min in order to facilitate mechanical dissociation. To stop enzymatic dissociation, an equal volume of resuspension solution (HBSS with 0.1% non-acetylated BSA (Invitrogen, 10743447)), 1% HEPES, 1× non-essential amino acids (Thermo Fisher Scientific, 11140050) and 10 µM rock inhibitor (StemCell Technologies, Y-27632) was added before passing cells through a 20 µm filter (Miltenyi Biotech, 130-101-812). Samples were then centrifuged at 100 rcf at 4°C for 5 min and resuspended in resuspension solution twice. Cells were then FAC sorted with 1 µL 0.1 mg/mL DAPI to remove dead cells and any remaining cell doublets. Immediately after FACS, samples were centrifuged and placed in 90% MeOH with 10% resuspension solution for storage.

Given the limited number of cells collected from individual dissection rounds and to obtain the required cell concentration for 10×, multiple rounds of dissection were carried out over successive days. Immediately prior to loading on the 10× Chromium controller, samples were pooled by stage and underwent three rounds of centrifugation at 1500 rcf at 4°C for 10 min followed by rehydration with DPBS with 0.5% non-acetylated BSA and 0.5 U/µL RNAse inhibitor (Roche 3335399001).

## Single-cell library preparation and sequencing

Cells were loaded on the 10× Chromium controller with the aim of capturing 3000–5000 cells per stage. cDNA synthesis and library preparation were carried out using the Chromium Single-Cell 3' Reagent v3 Kit (10× Genomics, Pleasanton, CA, USA) according to the manufacturer's protocol. This was carried out by the advanced sequencing facility at the Francis Crick Institute, London. Barcoded libraries were multiplexed and sequenced on an Illumina HiSeq 4000.

## RNAseq analysis

The initial RNAseq survey was conducted when galGal4 was the latest available chicken genome assembly. galGal4 and galGal3 compensated each other with the gene annotations (some genes were only annotated in one of the assemblies), therefore the RNAseq data was analysed with galGal3 and galGal4 as described below. Differentially expressed genes were then selected to represent on the NanoString probe codeset for further gene expression screening in triplicate with fine time course.

Raw sequencing data in FASTQ format underwent quality control analysis according to the pipeline published by *Blankenberg et al., 2010*. Files were converted to Sanger format using FASTQ *Groomer*. The quality scores of reads were calculated using FASTQ *Summary Statistics*. Quality scores were measured as phred = $-10 \log_{10}(p)$, where 'p' is the estimated probability of a base being called incorrectly. Reads with a phred score of equal to or greater than 20 (i.e. 99% probability of a base being correctly identified) were kept, while poor quality reads with average phred scores of less than 20 were filtered out using FASTQ *Quality Filter*. The remaining paired-end reads were trimmed at the 3' ends where the phred score drops below 20. All reads that passed quality control analysis were initially aligned to the chicken genome (assembly version Gallus_gallus-2.1, GenBank Assembly ID GCA_000002315.1) using TopHat (*Trapnell et al., 2009*). The raw data was re-analysed when the chicken genome was updated to assembly version gallus_Gallus-4.0, GenBank Assembly ID GCA_000002315.2 in 2013.

To build the GRN, RNAseq data was re-analysed using galGal5 to allow incorporation of the ChIPseq, ATACseq, and scRNAseq data on the same genome assembly. The node graft time course data together with previously published embryonic neural tissue RNAseq data (*Trevers et al., 2018*) were re-analysed using the *galGal5* genome assembly. Sequencing adapters and poor quality or N bases at 5' and 3' ends were trimmed from each sequence read using Trimmomatic-0.36 (*Bolger et al., 2014*). Unpaired reads and those that were less than 36 bases long after trimming were also removed. The remaining reads were aligned to the *galGal5* genome using TopHat2 (*Kim et al., 2013*), alignment rates were 67.4% ± 12.9%. A custom GTF file was generated by adding additional gene ERNI (NM_001080874) annotation, taken from RefSeq, to Gallus_gallus-5.0 Ensembl release 94 GTF file. Transcripts were counted and normalized based on the custom-edited GTF file using Cufflinks (*Trapnell et al., 2012*) program *cuffquant* and *cuffnorm,* respectively. The output table contains transcript FPKM (Fragments Per Kilobase of transcript per Million mapped reads) values across 11 samples

(node graft four time points and embryonic neural tissues with their controls). Sequence read tracks were computed using deepTools (*Ramírez et al., 2016*) *bamCoverage* (parameters *-bs 1 --scaleFactor 10^6/Library size --extendReads --samFlagInclude 66 –ignoreDuplicates --effectiveGenomeSize 1230258557*).

## Differential expression analysis

The initial RNAseq analysis of pooled samples was mainly used for selection of differentially expressed genes to design a probeset for NanoString expression analysis, for more detailed study of the timing of expression. Differentially expressed transcripts were identified by comparing uninduced and induced tissues at each time point (5, 9, and 12 hr). The 'easyRNASeq' (*Delhomme et al., 2012*) together with the Ensembl *galGal3* Gene Transfer File (GTF) were used for transcript counts from aligned reads. Differential expression analysis was then performed using two different methods: Cufflinks *cuffdiff* (*Trapnell et al., 2012*) and DESeq (*Anders and Huber, 2010*). Genes that are upregulated by a $\log_2$(FC of ≥1.2 or downregulated by ≤–1.2 in either Cufflinks or DESeq analyses, and were statistically significant with a p-value of ≤0.05 (i.e. coloured red or orange in *Figure 1—source data 1*) were selected). This identified 7745 transcripts that were differentially expressed across three time points (*Figure 1—source data 1*). Gene annotations were assigned to 4508 of these, corresponding to 2333 unique genes. Due to the incomplete nature of the *galGal3* chicken genome, 3237 transcripts were left unannotated.

This process was repeated using the Ensembl and UCSC *galGal4* GTF when they were released. For Ensembl transcripts, gene annotations and coordinates were obtained from Ensembl Biomart. For UCSC transcripts, gene annotations and coordinates were obtained from UCSC *galGal4* GTF file and annotation data from the UCSC table browser. Fully annotated transcripts from either Ensembl or UCSC lists were combined to provide the most comprehensive set of annotations. Genes that were upregulated by a $\log_2$(FC) of ≥1.2 or downregulated by ≤–1.2 in DESeq and were statistically significant using a p-value of ≤0.05 (i.e. coloured pale blue in *Figure 1—source data 2*) were selected. In situ hybridization was performed for transcriptional regulators that satisfied these criteria (*Figure 1—source data 2*). This reanalysis identified 8673 differentially expressed transcripts across the three time points. Gene annotations were assigned to 7184 of these, corresponding to 4145 unique genes, but 989 transcripts still remained unannotated. Volcano plots in *Figure 1R–T* were generated using the data generated from galGal4 assembly.

More stringent criteria were applied to select transcriptional regulators to represent on the NanoString for a more detailed study of the timing of expression. Transcriptional regulators that are upregulated at 5, 9, or 12 hr with $\log_2$(FC)≥1.2, p<0.05, and an induced base mean ≥45 were selected from DESeq analysis were selected. Transcriptional regulators that are downregulated at 5, 9, or 12 hr with $\log_2$(FC) ≤–1.2, p<0.05 and uninduced base mean ≥200 were also selected. A total of 213 transcriptional regulators were selected to be included in the NanoString probeset.

For the GRN pipeline, the expression data (*Figure 3—source data 1*) generated based on *galGal5* genome assembly was analysed in R-3.5.1 environment. FC was calculated from the levels of expression in the induced relative to the corresponding uninduced tissue, or neural relative to non-neural tissue samples. Genes with FPKM >10 in an induced or neural tissue and FC >1.5 are defined as upregulated, whereas genes with FPKM >10 in an uninduced/non-neural tissue and FC <0.5 are defined as downregulated. For the time point 0, a gene is defined as upregulated when 0 hr uninduced FPKM >10, FC >1.5 against 5 hr induced tissue, and 5 hr uninduced FPKM value is over 10 and larger than the value in 5 hr induced tissue. The FPKM expression levels of the candidate GRN members are provided in *Figure 3—source data 1*. This gene expression information was then incorporated with the results from the NanoString analysis, which were performed in triplicate. For details of how the RNAseq data were used to build a GRN, please see the section '*Pipeline for GRN*'.

## NanoString data analysis

Raw NanoString data were analysed in Microsoft Excel according to NanoString guidelines. Data were checked to ensure 600 fields of view were counted and that binding density values fell within the 0.05–2.25 range. Next, the geometric mean (geomean) of six positive control probes was calculated for each assay and averaged across the entire dataset. From these, a positive lane normalization factor (PLNF) was calculated by dividing the average geomean by the geomean for each assay. Then,

negative and endogenous probe counts were multiplied by their respective PLNF to normalize for differences in numbers of reporter and capture probes between assays. Next, the geomean of house-keeping (HK) genes ACTB and GAPDH was calculated for each assay and averaged across all assays. The average HK geomean was then divided by the HK geomean for each lane, to generate a lane-specific normalization factor (LSNF). Then, negative and endogenous probe counts were multiplied by the LSNF to normalize for cell number differences between assays. For each assay, the mean plus two times standard deviation was calculated from the eight normalized negative control probes. This value was subtracted from all normalized endogenous probe counts to remove background noise levels. Any transcript counts that became negative as a result were reset to zero. Next, the normalized expression counts were set to 1 when the counts are equal to zero. Average counts for each probe and standard deviation were calculated across triplicate assays. Finally, FC for each probe was calculated by dividing the induced average by uninduced average. Differential expression was defined as FC of ≥1.2 (upregulation) or ≤0.8 (downregulation). The statistical significance was calculated using a two-tailed Type 2 t-test with a p-value of 0.05. The result is provided in *Figure 3—source data 1*.

## ChIPseq analysis

Single-end sequence reads were quality checked and trimmed using Trimmomatic-0.36 with the same criteria as RNAseq analysis and aligned to the *galGal5* genome using bowtie2 (*Langmead and Salzberg, 2012*) with default parameters. The alignment rates were 92.3% ± 6.5%. PCR duplicates were removed using Piccard *MarkDuplicates* (parameter *REMOVE_DUPLICATES = TRUE*). Peak calling was computed using MACS2 (*Zhang et al., 2008*) *callpeak* (parameters *-f SAM -B --nomodel --broad --SPMR -g 1230258557 -q 0.01*) with genomic input as control. The MACS2 outputs were quantified and filtered for the peaks enriched with either H3K27ac or H3K27me3 by comparing to the corresponding inputs with cut-off p-value <1E-5, FC >1.2, and q-value >3. Signal tracks were computed using deepTools *bamCompare* (parameters *--scaleFactorsMethod SES --operation log2 -bs 1 --ignoreDuplicates*) and displayed using R package trackViewer (*Ou and Zhu, 2019*).

CTCF ChIPseq data (*Kadota et al., 2017*) were processed with the same pipeline. The resulting file (bed format) from peak calling was used as an input file to the pipeline for constructing our GRN.

## ATACseq analysis

Paired-end sequence reads were quality checked and trimmed using Trimmomatic-0.36 with the same criteria as RNAseq and aligned to the *galGal5* genome using bowtie2 (parameters *-X 2000 --sensitive-local*). The alignment rates were 92.36% ± 5.33%. PCR duplicates were removed using Piccard *MarkDuplicates* (parameter *REMOVE_DUPLICATES = TRUE*). Coverage tracks were generated using deepTools bamCoverage (parameters *-bs 1 --scaleFactor 10⁶/library size --extendReads --samFlagInclude 66 --ignoreDuplicates --effectiveGenomeSize 1230258557*).

## scRNAseq data analysis

scRNAseq alignment and downstream analysis was run using Nextflow (version 20.07.1) (*Di Tommaso et al., 2017*) and Docker for reproducibility. All of the required packages and respective versions are found within the Docker container alexthiery/10×_neural_tube:v1.1. The full analysis pipeline, including documentation, can be found at https://github.com/alexthiery/10x_neural_tube, (*Trevers et al., 2023a* copy archived at swh:1:rev:30b789553714ccc4ed436f37fdf0efb4e332c888).

In order to obtain the recommended 50k reads/cell, libraries were sequenced twice, with reads from both flow cells pooled during sequence alignment. scRNAseq reads were demultiplexed, aligned, and filtered using Cell Ranger (version 3.0.2, 10× Genomics). Reads were aligned to *galGal5*, using a custom-edited GTF annotation file as described in '*RNAseq analysis*' section. Prior to alignment, MT, W, and X chromosome genes in the g*alGal5* GTF file were prefixed accordingly for simple identification downstream. Coordinates for W chromosome genes in *galGal6* were also transferred to *galGal5* and prefixed in the GTF.

Quality control, filtering, dimensionality reduction, and clustering were carried out in Seurat (version 3.1.5) (*Stuart et al., 2019*). We first applied a modest filtering threshold, removing cells with greater than 15% UMI counts from mitochondrial genes and cells with fewer than 1k or greater than 6k unique genes. After filtering 8652 cells remain in the dataset (HH4=2745; HH6=1823; HH8=1750; HH9+=2334). Throughout subsequent clustering steps, we measure total UMI counts, total gene

counts, and percentage mitochondrial content for each cluster and remove any clear outlier clusters. Following initial clustering, cell clustering was dominated by the expression of a few W chromosome genes and we therefore removed W chromosome genes. We then identified and removed contaminating cell clusters, including putative mesoderm and primordial germ cells. Percentage mitochondrial content, cell cycle, and sex were all regressed out during scaling of the final dataset.

PCA was used as an initial dimensionality reduction step, with the top 15 PCs used for clustering. We then calculated k-nearest neighbours in order to embed a shared nearest neighbour (SNN) graph. The Louvain algorithm was used to partition the SNN graph according to the Seurat default parameters. At each clustering step, we visualized multiple resolutions using the R CRAN clustree package (*Zappia and Oshlack, 2018*) and determined the optimal resolution manually based on cluster stability.

An unbiased approach was used to identify modules of co-correlated genes using the Antler gene modules algorithm (https://github.com/juliendelile/Antler) (*Delile et al., 2019*). For this we kept genes which have a Spearman correlation greater than 0.3 with at least three other genes. Ward's hierarchical clustering is then carried out on the Spearman gene-gene dissimilarity matrix of the remaining genes. Iterative clustering by the Antler algorithm selects modules based on their consistency. In order to identify gene modules that explain the greatest amount of variation between our cell clusters, we filtered modules for which fewer than 50% of genes are differentially expressed (logFC >0.25, adjusted p-value <0.001) between at least one cell cluster and to the remaining dataset. Differential expression tests were carried out using Seurat FindAllMarkers function (Wilcoxon test).

Neural cell clusters were subset from the parent dataset using a candidate marker approach. PC1 inversely correlates with developmental stage, therefore cells were subsequently ranked according to their position along PC1. GRN components were grouped based on the timing of their initial expression following neural induction. Housekeeping genes (*GAPDH; SDHA; HPRT1; HBS1L; TUBB1*; and *YWHAZ*) were used as a control. A general additive model was used to estimate the changes in scaled expression across the ranking of PC1, with a smoothing spline fitted by REML.

## Pipeline for GRN
### Merging RNAseq and NanoString data
Two hundred and thirteen transcriptional regulators were selected from the RNAseq time course data (0, 5, 9, and 12 hr). The data from NanoString nCounter assay provided fine time course expressions (1, 3, 5, 7, 9, 12 hr after node graft) of the GRN components. It is not only a supportive dataset to the RNAseq expression data for the time points 5, 9, and 12 hr, but also provides gene expression levels for 1, 3, 7 hr for the GRN. The details for differential expression analysis are described in the '*Differential expression analysis*'. The integrated time course gene expression profile can be found in *Figure 3—source data 1*. There were 180 transcriptional regulators that have expressions in agreement in both RNAseq and NanoString screening and thus remained in the list as the candidate GRN components for the subsequent GRN analysis.

### Identifying CTCF sites
To define the neural induction regulatory loci, the genomic coordinates of the GRN candidate members were extracted from Gallus_gallus-5.0 Ensembl release 94 GTF file. The in-house script searches upstream and downstream of the candidate GRN components up to 500 kb for the peaks with highest *signalValue* from CTCF ChIPseq peak calling result file. The coordinates of regulatory loci are listed in *Figure 3—source data 1*.

### Identifying putative regulatory regions
Different types/indices of putative regulatory sites (*Figure 2—figure supplement 1A*) were obtained from the ChIPseq H3K27 peak calling result files using BedTools (*Quinlan and Hall, 2010*). The command *intersect* reports the overlapping peaks between the ChIPseq results, while *intersect* with parameter -v reports the non-overlapping peaks between a set of ChIPseq outputs. The command *merge* (parameters -d 100c 1,4 -o count,collapse) was used to merge and generate a unique peak set from a subset of results. 'Activation' regulatory sites, as an example, were merged from Indices

1–3. The coordinates of the regulatory loci in *Figure 3—source data 1* were then used as an input to the in-house script to extract the coordinates of regulatory sites of the GRN candidate members. FASTA format sequences of these regulatory sites were also extracted from *galGal5* genome assembly Ensembl release 94 for the transcription factor binding site screening.

## Putative TF binding sites

A transcription factor binding motif library of the GRN candidate members was extracted from JASPAR2020 CORE non-redundant database (*Fornes et al., 2020*). SALL1 and GRHL3 consensus binding motifs were obtained from *Karantzali et al., 2011*; *Klein et al., 2017*, and converted into meme motif format. A total of 91 transcription factor binding motifs were curated as an input library for running binding motif screening on the putative regulatory sites (FASTA format files) using MEME suite FIMO (*Grant et al., 2011*). The in-house script selects predicted biding sites from FIMO outputs with confidence score >10 and calculates the genomic coordinates of the binding sites. The resulting outputs are provided in *Figure 4—source data 1*.

## Building the GRN

The in-house script integrates gene expression profile and transcription factor binding profile within the putative regulatory sites that undergoing activation (Indices 1–3 and 7) to generate the neural induction GRN. A positive regulatory interaction is modelled when the regulator and the target gene are both up- or downregulated at one time point, whereas a negative regulation is predicted when the regulator and the target gene have opposing expression profiles (*Figure 4A–B*). Duplicated interactions derived from the bindings of a regulator to the multiple regulatory sites of a target gene were removed. The putative regulatory sites from 5h ChIPseq data were used for generating the GRN at time points 1, 3, and 5 hr. The sites from 9h ChIPseq were used for the GRN at time point 7 and 9 hr. The in-house script generates files, including Model Hierarchy CSV and Time Expression XML Data, for visualizing the GRN using BioTapestry (*Longabaugh et al., 2005*; *Longabaugh et al., 2009*; *Paquette et al., 2016*). The BED files were also generated for uploading to the UCSC browser for visualizing the GRN related data. Five genes, including BAZ1A, GATAD2B, CRIP2, PBX4, and SALL3, were excluded from the GRN, because they have neither input nor output interactions with other members of GRN. This gives the final GRN with 175 components in the network.

## Predicting regulatory regions from regions with conserved sequence motifs

DREiVe (*Sosinsky et al., 2007*; *Khan et al., 2013*; *Streit et al., 2013*) was used to search for conserved elements within 500 kb up- and downstream of the 175 GRN components. The default settings were used to compare between six species: human (*hg38*), mouse (*mm10*), rat (*rn6*), golden eagle (*aquChr2*), zebrafish (*danRer10*), and chicken (*galGal5*). Elements that are conserved across a minimum of four species were selected and displayed on the UCSC browser.

## Extracting a subnetwork

To generate a subnetwork of a selected target gene BRD8, the transcription factors, predicted to bind to the regulatory sites that undergoing either activation or repression (Indices 1–7), were extracted from the FIMO outputs. BedTools *merge* (parameters *-d 500c 1,4 -o count,collapse*) was used to check the overlaps of the regulatory sites across three time points (5, 9, and 12 hr). Each unique regulatory site was treated as one independent component in the network. The in-house script generates the same output files as the main GRN for visualizing using BioTapestry.

## Network properties

To identify the core transcription regulators in the GRN, the betweenness centrality (*C*B) of each gene is computed as follows *White and Borgatti, 1994*:

$$C\text{B}(i) = \sum \partial(i) / \partial$$

where $\partial$ is the total number of directed shortest paths from one component to another component in the network, and $\partial(i)$ is the number of these directed shortest paths that pass through $i$ for each time point. Cytoscape (*Shannon et al., 2003*) was then used for the network representation.

## Acknowledgements

This study was funded by grants from NIH (R01 MH 60156), MRC (G0400559), Wellcome Trust (063988) and BBSRC (BB/R003432/1 and BB/K007742/1) to CDS and BBSRC (BB/K006207/1) to AS. The work of NML is supported by the Francis Crick Institute which receives its core funding from Cancer Research UK (FC010110), the UK Medical Research Council (FC010110), and the Wellcome Trust (FC010110). The scRNAseq data analyses were performed using infrastructure from the Crick Scientific Computing science technology platform. NML is a Winton Group Leader in recognition of the Winton Charitable Foundation's support towards the establishment of the Francis Crick Institute.

## Additional information

### Funding

| Funder | Grant reference number | Author |
| --- | --- | --- |
| National Institute of Mental Health | R01 MH60156 | Claudio D Stern |
| Medical Research Council | G0400559 | Claudio D Stern |
| Wellcome Trust | 063988 | Claudio D Stern |
| Biotechnology and Biological Sciences Research Council | BB/R003432/1 | Claudio D Stern |
| Biotechnology and Biological Sciences Research Council | BB/K007742/1 | Claudio D Stern |
| Biotechnology and Biological Sciences Research Council | BB/K006207/1 | Andrea Streit |
| Francis Crick Institute | | Nicholas M Luscombe |
| Cancer Research UK | FC010110 | Nicholas M Luscombe |
| Medical Research Council | FC010110 | Nicholas M Luscombe |
| Wellcome Trust | FC010110 | Nicholas M Luscombe |

The funders had no role in study design, data collection and interpretation, or the decision to submit the work for publication. For the purpose of Open Access, the authors have applied a CC BY public copyright license to any Author Accepted Manuscript version arising from this submission.

### Author contributions

Katherine E Trevers, Resources, Validation, Investigation, Writing – original draft, Writing – review and editing; Hui-Chun Lu, Conceptualization, Resources, Data curation, Software, Formal analysis, Validation, Investigation, Visualization, Methodology, Writing – original draft, Writing – review and editing; Youwen Yang, Resources, Validation, Investigation, Methodology; Alexandre P Thiery, Resources, Formal analysis, Investigation, Writing – review and editing; Anna C Strobl, Božena Pálinkášová, Validation, Investigation; Claire Anderson, Irene M de Almeida, Natalia Moncaut, Investigation; Nidia MM de Oliveira, Investigation, Project administration; Mohsin AF Khan, Resources, Software, Formal analysis; Nicholas M Luscombe, Supervision, Methodology; Leslie Dale, Supervision; Andrea Streit, Supervision, Funding acquisition, Methodology, Writing – review and editing; Claudio D Stern, Conceptualization, Resources, Supervision, Funding acquisition, Writing – original draft, Project administration, Writing – review and editing

### Author ORCIDs

Andrea Streit http://orcid.org/0000-0001-7664-7917
Claudio D Stern http://orcid.org/0000-0002-9907-889X

Decision letter and Author response
Decision letter https://doi.org/10.7554/eLife.73189.sa1
Author response https://doi.org/10.7554/eLife.73189.sa2

## Additional files

### Supplementary files
• MDAR checklist

• Supplementary file 1. BioTapestry File (.BTN) of the full network - to open/view, please download the BioTapestry software from http://www.biotapestry.org.

• Supplementary file 2. Screen shots from the BioTapestry representation, showing the GRN at each successive time point.

### Data availability
Full scRNAseq software and pipelines deposited in https://github.com/alexthiery/10x_neural_tube (copy archived at *Trevers et al., 2023a*). Full software/scripts/pipelines for GRN construction deposited in https://github.com/grace-hc-lu/NI_network (copy archived at *Trevers et al., 2023b*), sequencing datasets in ArrayExpress under E-MTAB-10409, E-MTAB-10420, E-MTAB-10424, E-MTAB-10426, and E-MTAB-10408, expression patterns submitted to GEISHA (http://geisha.arizona.edu/geisha/). Code for DREiVe: https://github.com/grace-hc-lu/DREiVe (*Khan et al., 2023*).

The following datasets were generated:

| Author(s) | Year | Dataset title | Dataset URL | Database and Identifier |
| --- | --- | --- | --- | --- |
| Stern C | 2023 | A gene regulatory network for neural induction (RNA-seq of total RNA) | https://www.ebi.ac.uk/biostudies/arrayexpress/studies/E-MTAB-10409 | ArrayExpress, E-MTAB-10409 |
| Stern C | 2023 | A gene regulatory network for neural induction (ATAC-seq) | https://www.ebi.ac.uk/biostudies/arrayexpress/studies/E-MTAB-10420 | ArrayExpress, E-MTAB-10420 |
| Stern C | 2023 | A gene regulatory network for neural induction (ChIP-seq) | https://www.ebi.ac.uk/biostudies/arrayexpress/studies/E-MTAB-10424 | ArrayExpress, E-MTAB-10424 |
| Stern C | 2023 | A gene regulatory network for neural induction (ATAC-seq) | https://www.ebi.ac.uk/biostudies/arrayexpress/studies/E-MTAB-10426 | ArrayExpress, E-MTAB-10426 |
| Stern C | 2023 | A gene regulatory network for neural induction (RNA-seq of coding RNA from single cells) | https://www.ebi.ac.uk/biostudies/arrayexpress/studies/E-MTAB-10408 | ArrayExpress, E-MTAB-10408 |

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
