## [Editor Report]

In this manuscript, Trevers and colleagues undergo a detailed genome-wide exploration of the mechanisms of neural induction in chick embryos. They describe the gene regulations governing the patterning of extra-embryonic ectoderm into neural ectoderm upon the graft of an early Hensen's node ectopically, an assay for neural induction and neural commitment. The data are assembled into a Gene Regulatory Network of 175 transcription factors and their projected interactions, based on a fine-scale temporal analysis. This study will be an important resource for the field of neural induction.

---

## [Decision Letter]

**Decision letter after peer review:**

Thank you for submitting your article "A gene regulatory network for neural induction" for consideration by *eLife*. Your article has been reviewed by 4 peer reviewers, one of whom is a member of our Board of Reviewing Editors, and the evaluation has been overseen by Kathryn Cheah as the Senior Editor. The following individuals involved in review of your submission have agreed to reveal their identity: Richard M Harland (Reviewer #2); Leonardo Beccari (Reviewer #3).

Essential revisions:

1) All the reviewers agree that this works will constitute an outstanding resource for the community and for further experimental exploration. As such, the study would better fit the "resource" section of the journal. Please reformat the study accordingly is you wish to resubmit it here;

2) As a resource, they also all agree that the study should include a few specific examples to show how this extensive set of data advances our understanding of the molecular processes of neural induction, and consequently how it should be used. Possible examples have been suggested below by the reviewers;

3) Improvement in transcriptomic and epigenomic data analysis along the lines suggested by reviewer 3 are essential to consolidate and strengthen the study;

4) Deposition of the in situ hybridation data in a public repository;

5) Please discuss how node graft modulates the AO program (reviewer 4, point 1).

6) Please address the comments on pioneering factors, poised/bivalent promoters (reviewers 3 and 4).

7) Take home messages from both the synthetic network analysis and the chosen examples should be clarified.

*Reviewer #1 (Recommendations for the authors):*

In this manuscript, Trevers and colleagues undergo a genome-wide exploration of the molecular actors of neural induction in chick embryos. Specifically they describe the gene regulatory network (GRN) governing the patterning of extra-embryonic ectoderm into neural ectoderm, upon the graft of an early Hensen's node ectopically, an assay extensively used by these authors in the past to understand how neural induction and neural commitment were activated by node grafts: They have notably previously identified early responsive genes such as ERNI, mid-term responsive genes (e.g. *Sox2*), and late specification markers such as Sox1. It still remained unclear to which extent this model faithfully mimics the mechanisms of endogenous neural tissue induction at the midline of the embryo, a question explored here.

Here, the authors examine the states of the induced extra-embryonic ectoderm at three different time points after grafting (5h, 9h, 12h), using RNAseq, ATACseq and ChIPseq for histone active/repressive marks. They further select about 200 transcriptional regulators (transcription factors and chromatin modifiers) to conduct a Nanostring analysis at 6 time points and build a large GRN for neural induction, with its temporal dynamics, based on the putative binding sites for these regulators located around target genes.

This study generates an extensive set of data, integrated to a genome browser, allowing one to explore the refined details of the molecular events triggered by the graft of a node into the ectoderm. Moreover using the expression patterns for 174 transcriptional regulators in vivo, and single cell transcriptomes, the authors compare their GRN to the dynamics of gene (co)-expression in the neural plate and conclude that the temporal dynamics of the ectopic neural induction assay matches the sequence of gene expression during neural induction at the midline.

This study nicely confirms the previous model of neural induction in chick embryos. The new data could be used to predict and assay novel details of neural induction or neural commitment as suggested by the introduction. More than 5600 interactions were predicted but they remain to be validated: for example, the direct regulation of a few new GRN components by their predicted upstream modulators could have been tested. At this point, it is unclear how this study expands our knowledge of neural induction. Nonetheless, this dataset will constitute an important resource for future exploration of neural induction mechanisms.

The direct demonstration of the role of one or two new regulators would add significant validation to the predictions of this study. Among the 174 regulators studied, some probably present a particularly interesting expression patterns, or position in the GRN, suggesting a critical yet unknown function. Demonstrating the role of such a factor on the known elements of the GRN (ERNI, *Sox2*, Sox1 etc) and validating its regulation by the predicted upstream regulators in vivo, during neural induction at the midline, would demonstrate the discovery power of this new GRN.

*Reviewer #2 (Recommendations for the authors):*

The manuscript by Trevers, Lu et al. Is an impressive and useful dataset on the emergence of neural, placode, neural crest and non-neural ectoderm at the levels of RNA expression, and chromatin marks. The data have been assembled into hypothetical Gene regulatory networks based on timecourses of expression at closely spaced intervals. The most impressive part of the work is this fine scale temporal analysis of gene expression and regulation, which occurs over several hours in the chick epiblast, where the embryological assays that define competence and commitment can be correlated with gene expression, and with changes in chromatin marks. The work primarily emphasises the genes that change in response to node/organizer grafts, but the paper also studies the emergence of these transcripts in normal neural development, and shows that the node inductions closely mirror the changes in the normal ectoderm.

While the work is potentially very useful, the current manuscript does not digest the data to generate biological insights of particular interest to the reader. The Venn diagrams summarize the number of changes, but don't give much sense of how they fit into our current understanding of neural induction, and the Biotapestry diagrams are difficult to analyze by eye when they include so much detail. The ppt file provides more annotated versions of the Biotapestry, but again, it is difficult to extract biological interpretations from them.

Biologically, the paper does present some arguments that suppression of non-neural gene expression (by TFAP2C) and other data supports the contention that suppression of non-neural and neural border fates is important in neural induction, but there is little discussion of neural induction itself. On the positive side, there are several already known and some new cis-regulatory modules that have been identified, and some tested for function by electroporation, but this just makes the reader want to know what motifs or logic lie in these domains. For example, the ability of the node, but not BMP antagonists to induce neural tissue in the area opaca is consistent with the early expression domain of Sox3 as a neural competence defining gene, but it would also be useful to couch some of the discussion in terms of what other genes might lead to increased competence to respond to FGFs or absence of BMP signaling over the timecourse of analysis. Recent work for the Briscoe lab has suggested that FGF signaling is important in conditioning posterior fates prior to neural specification in mammalian stem cells, supporting data in *Xenopus* and fish that FGF signaling has a large role in defining a competence region for posterior neural induction. It would be interesting to relate the current work to these kinds of findings. What are the genes that predispose to anterior responses or posterior ones, and how separable are they in time? Also, what can be learned from the large dataset on commitment in neural induction and to what extent does it correlate with expression of rostrocaudal or mediolateral regional markers in the dataset?

So overall, this is a high priority paper for publication as a resource, but the biological insights that come from the work are surprisingly few, making the manuscript disappointing to read as a regular article. I am not going to follow the prescriptive requirements in the *eLife* instructions, but I would hope that the authors include more analysis and comparisons to the paper to bring out the strengths of the datasets.

Specific comments:

The clustering of single cell data is interesting, but it is hard to understand why the clusters were not labelled using the data (illegible) in figure S4, indeed why is S4 not in the main paper instead of the less informative data in Figure 2 or 3B. Why was BRD8 chosen for detailed presentation? Is it a neural specific gene? I note that it is not in Geisha, and its expression is not described here. The biotapestry representation falls down with this many linkages, and it would be good to have included a simplified diagram for the different sites/stages.

I had hoped for some representations of specific genes that are central to the likely steps in neural induction, (is BRD8 one of them?) along with a discussion of what feeds into them, and goes beyond the previous characterization of the *Sox2* regulatory motifs.

*Reviewer #3 (Recommendations for the authors):*

In this work, Trevers et al. combine classic Hensen node transplantation experiments and transcriptional and epigenetic profiling to characterize the gene regulatory network at work during anterior neural commitment. Besides, they combine the analysis of chromatin marks for active/ repressed enhancers together with transcription factor binding sites (TFBS) prediction to define the cis-regulatory code controlling the expression of the genes involved in this process. These next-generation-sequencing based approaches are further supported by extensive analysis of gene expression patterns by whole-mount in situ hybridization and enhancer transgenesis assays. The authors also use scRNAseq analysis to prove that the gene transcriptional dynamics characterized in their node graft model closely recapitulate those of normal anterior neuroectoderm induction. The most notable outcomes of this overall nicely conceived and experimentally well-performed study are the definition of a GRN composed of 175 transcriptional regulators and the characterization of 79 genes not previously associated with neural induction being differentially expressed during this process and will be of interest for the neurodevelopmental biology community. Although the transcriptional profiles associated with the neural induction process have been in part characterized in bulk or sc cell transcriptomic profiling of both mouse and chicken embryos, including recent work from the same authors [1]-[4], the resources generated in this study are intended to further dig into the molecular event occurring during anterior neuroectoderm induction. The main limitation of this study is that it remains largely descriptive without fully addressing the logic of the GRN governing the neural induction process, thus limiting the impact of the work. This aspect could perhaps benefit from a more detailed analysis of the data generated in this study.

The main experimental concerns about the study of Trevers et al. are related to the design and analysis of the transcriptional profiles of induced vs non induced ectoderm.

Suggestions/comments about the analysis of the GRN and gene cis-regulatory code are also presented below:

– In the Material and Methods section, the authors describe having analyzed the RNAseq data against different chicken genome assembly versions, ranging from galGal2.1 to galGal5. The source data file provided by the authors provides the differential expression analysis performed in the galGal3 and galGal4 assemblies. Thus, it is difficult to understand which analysis was used for the elaboration of the figures and the description of the results. Provided that no specific conclusions arise from the analysis of different genome assembly versions (in which case it should be clearly stated), I find the description of the different analyses unnecessary and even confusing. In my opinion, authors should describe only the one used for the elaboration of the figures and data interpretation in the manuscript, preferentially using the latest genome assembly and gene annotation versions.

– The RNAseq analysis presented by the authors has been performed using individual pooled samples of induced /non induced dissected ectoderm, which imposes identifying differentially expressed genes on the basis of p-value rather than on FDR. While I understand that the complexity of the experiment limits the possibility of performing a high number of biological replicates, the robustness of the data would greatly benefit from having at least a second biological replicate, particularly because this work aims at characterizing the gene regulatory network operating during neural induction based on the analysis of differential gene expression across the process. Besides, the authors report having used different criteria for genes in their galGal3/galGal4 -based differentially expressed analysis and to produce the galGal5 FPKM table of differentially expressed genes (log2(FC)>1.2 in galGal3/galGal4 based analysis and FC>1.5 in the latter case). The log2(FC) results in the source data table for the galGal3 and galGal4 assemblies also seem to be presented in a different manner which does not help to correlate the results. Authors should uniformize their analysis criteria or clearly state the rationale for the use of different parameters.

– While the authors report a roughly similar number of activated and repressed genes in induced vs non-induced ectoderm samples (based on RNAseq data) the number of activated and repressed regulatory elements strongly diverge between the two samples, with a large majority of elements activated (particularly at 9h and 12h post grafting) rather than repressed. This difference could be due to "technical" reasons (e.g: the presence of a fraction of cells in the samples where certain regulatory elements are repressed only in a subpopulation of cells may not allow to distinguish these H3K27me3+ elements from those that not at all active, resulting in the identification of only those region that have a nearly all vs nothing response). Alternatively, it could be due to the fact that gene activation relies on combinatorial enhancer activity while its repression may require the action of fewer regulatory elements. Could the authors comment on this?

– Authors define the regulatory elements associated with a specific gene based on the correlation in their activity and the gene transcriptional state and on the distribution of CTCF sites. Although it is known that CTCF sites contribute to defining the limits of topologically associated domains (TADs, in which the gene regulatory landscape should be located), they are also found within TADs. Furthermore, considering the orientation of CTCF sites is also an important indicator of whether two CTCF bound regions (or clusters thereof) may contribute to define a gene regulatory domain. The authors could try to improve the definition of the gene regulatory domains by using available chicken HiC data, in combination with CTCF coverage and site orientation. Since TADs are in large part independent from gene transcriptional activity different HiC datasets (eg [5]-[7)] could be combined.

– Combining the analysis of epigenetic states and transcriptional profiles of induced/ non-induced ectoderm with binding site prediction of the differentially expressed transcriptional regulators is an interesting approach for addressing the gene regulatory network at work during the neural induction process. However, in my opinion, the analysis presented in the manuscript and figure 3 falls shorts compared to valuable resources generated by the authors (although this is not my field of expertise). For example, would it be possible for the authors to describe better the structure of the network in terms of the distribution of connections (predicted regulatory interactions) per node (genes), perform an unbiased identification of kermel genes and overrepresented network motifs (e.g. feedback and feedforward loops), possibly highlighting some corresponding to known interaction as well as new gene-gene connections?

– The role of pioneer factors in the control of cell fate acquisition/ reprogramming is of transversal interest in the developmental biology and chromatin and transcription fields (eg [8]-[10]). The neural induction experimental paradigm used in this work and the datasets generated could point out transcription factor pioneering activities relevant for the neural induction process and provide interesting insights for the understanding of the logic of the neural induction GRN, for example, by analyzing TFBS overrepresented in regulatory elements that are active long before the transcriptional onset of their (putative) target genes.

– Despite having generated an important amount of ATACseq data, the authors present these results very briefly, not indicating the statistics of elements displaying differential accessibility across samples and explaining very concisely how these data were used to aid in the selection enhancers for the transgenesis assay. Chromatin accessibility profiles could also help in the analysis of differentially active/repressed/ poised elements presented in Figure 2 and Figure S2-S3 and in the identification of elements targeted by pioneering TFs. Could the authors expand on this analysis (provided that the quality of the data allows genome-wide analysis of changes in chromatin accessibility)?

– Authors define poised enhancers as those displaying either both H3K27ac/H3K27me3 marks in the same sample or lacking both of them. This definition remains delicate both because of the potential heterogeneous cell population in the dissected tissue (as authors discuss) and by the lack of H3K4me1 profiles. In this sense, crossing the H3K27ac/ H3K4me3 and ATACseq data could improve defining these poised elements (H3K27ac-, H3K27me3+ and high chromatin accessibility [11]).

– Authors claim having identified the identification of 79 transcriptional regulators not previously associated with neural induction and differentially expressed in the node graft model and during normal chicken neural induction. This finding would deserve further exploration/ discussion, for example, by analyzing the expression of these genes in mouse scRNAseq datasets or from RNAseq profiles in cell culture models of ES/IPS cell-Neural differentiation.

– Based on the description of the authors of the ChIPseq data analysis, authors base their analysis of active/ repressed elements in induced/non induced ectoderm samples on qualitative criteria; i.e, regions called by MACS2 peaks in one sample but not in the other. However, using a more quantitative approach (for example, MAnorm analysis of coverage levels) may also highlight elements that increase or decrease their activation/repression state in induced vs uninduced ectoderm samples as well as across time, which could contribute to answering the question raised in the public review. Besides, and related to that point, if the analysis is restrained to gene promoters, does the proportion of elements differentially enriched in H3K27ac/H3K27me3 reflect the transcript data? Are H3K27ac /H3K27me3 elements located at a different average distance from gene TSS?

– The new advances in RNAseq library preparation and column assisted RNA extraction from low cell number samples allow to produce reliable RNAseq data from considerably lower amounts of tissue, thus eventually reducing the technical challenge of increasing the number of replicates for the transcriptional profile analysis.

*Reviewer #4 (Recommendations for the authors):*

This study represents a milestone in the field of neural development, uncovering thousands of predicted regulatory interactions underpinning the process of neural induction and more generally, underlines the complexity of the process of cell fate acquisition. The work constitutes an important resource, presenting and validating useful data sets that capture transcriptomic changes (bulk RNA seq, scRNAseq, Nanostring and in situ hybridisation) leading up to expression of mature neural plate markers. Moreover, by combining ChIPSeq for key chromatin modifications and ATAC seq data the authors identify putative enhancer regions mediating neural induction and carefully validate six of these using mis-expression in the early chick embryo. Overall, the work opens a new frontier in the analysis of molecular basis of neural induction – the logic (beyond the hierarchies identified here) of the thousands of transcription regulating events will require extensive further experimentation.

Overall, the experiments are well designed, the data are clearly explained and for the most part well presented in the figures. The paper would be strengthened by inclusion of examples of conservation in mammals, of the key GRN loci identified in the chicken. This reviewer does not have the expertise to comment on detail of data analyses packages and approaches presented.

1) "Of these, 4130 were upregulated (enriched in "induced" tissue) and 4543 were downregulated (depleted in "induced" tissue) relative to the "uninduced" counterpart."

It is an important finding that Organiser induced gene expression changes involve down regulation as well as up regulation of genes in the AO epiblast – can the authors provide an indication of how much down regulation is due to initial starting cell state and may not reflect the endogenous neural induction process?

Related to this point (page7 para 2), it is important to have established the relationship between AO epiblast response to the node and sequence of gene expression patterns during normal/ endogenous neural plate formation. The authors note the similarities, but were there also differences that would allow better discrimination of regulatory steps that reflect the AO epiblast cell state? These points should also be considered/discussed in the text.

2) H3K27Me3 is not always a marker of transcriptionally silenced genes. While I note the authors comment that " this is often the case" more caution in subsequent interpretation may be required, along with discussion of further chromatin modifications which could corroborate gene silencing.

3) Can the authors explain their choice to use the term poised as opposed to bivalent, when referring to gene loci associated with both H3K27Me3 and H3K27ac modifications. Bivalency is usually inferred when H3K27Me3 and H3K4Me3 chromatin modifications are present, was any such analysis also carried out to confirm the "poised" state.

4) "In situ hybridization was performed for genes encoding 174 transcriptional regulators (including 123 that are represented in the GRN) at 4 stages: pre-streak (EGKXII-XIII), primitive streak (HH3-4), head process/early neural plate (HH5-6) and neural fold/tube (HH7-9)".

Are these ISH data deposited or accessible somewhere? An appropriate place for this resource that is familiar and accessible to the scientific community is Giesha http://geisha.arizona.edu/geisha/ This should be included in the data availability section.

5) Page 7 para 3 (Figure 6)

The "ectoderm" dissected and analysed here should be more accurately defined. Figure 6A schematic shows that the authors excluded the ventral midline at HH4 and HH6, but were perhaps unable to do this at later stages? This sets in context their finding that "Cells collected from the two earliest developmental stages (HH4 and HH6) each form a distinct group, whereas cells from later stages (HH8 and HH9+) are clustered primarily according to cell type (Figure 6C)". The authors should comment on inclusion of the ventral midline in these data sets. It is also curious that cluster 12 includes a node marker ADMP, as explanted tissues did not include the node in Figure 6A.

6) Page 9, the first sentence of the Discussion

"This study unfolds the full complexity of the responses to signals from the "organizer" in very fine time course up to the time of appearance of mature neural plate markers, such as SOX1."

By what criteria do the authors consider that their study uncovers the "full complexity" of responses to the signals from the organiser. I think they have uncovered the complexity.

7) page 10 para 3

Analysis of signalling pathway targets likely to be informative – should make clear that the GRN here only includes transcription factors (TF) and that changing TF expression is also very likely to be regulated by exposure to changing signalling as development proceeds – presented by neighbouring non neural tissues – rather than due simply to a hierarchal cascade of TF regulation from early stages.

8) The last sentence of the Discussion is almost identical to the last sentence of the Results section.

9) Somehow at the end of the paper we are still left wanting an overview of ways forward with these data sets – in particular are there any take home messages / obvious next experiments which would elucidate the regulatory logic of the transcriptional dynamics /hierarchies uncovered here? Is it clear, for example, that many of the TFs characteristic of specific time points are targets of the same signalling pathway? Can they include examples of conservation in mammals (using the DREiVe computational tool) of key GRN loci identified in the chick?

---

## [Author Response]

Essential revisions:1) All the reviewers agree that this works will constitute an outstanding resource for the community and for further experimental exploration. As such, the study would better fit the "resource" section of the journal. Please reformat the study accordingly is you wish to resubmit it here;

We have followed this suggestion and now resubmit the revised paper as a “resource”.

2) As a resource, they also all agree that the study should include a few specific examples to show how this extensive set of data advances our understanding of the molecular processes of neural induction, and consequently how it should be used. Possible examples have been suggested below by the reviewers;

As a resource, we feel that the main usefulness of this paper is to provide a platform that will allow others to make predictions and to test hypotheses as well as to validate the network in various ways. Along with the new revisions the resource is even more powerful than before. Nevertheless, we have indeed provided validation of several of the predictions in 5 complementary ways:

(a) for 6 of the newly predicted regulatory elements (associated with SOX11, SETD2, *SOX2*, CDX1, *GLI2* and SIX3), we have tested their activity by widespread expression of reporter constructs (containing the minimal promoter TK, the regulatory element and GFP) in the normal embryo. In all cases, the correspondence between the pattern of expression of the reporter construct and that of the associated target gene in the prospective neural plate of the embryo is remarkably close, as is the time of onset of this activity relative to the position of the gene / element in the network. This provides strong validation of this set of predictions;

(b) we show the normal patterns of expression of nearly all the genes in the network, by in situ hybridisation. All of them are expressed in the normal future neural plate (despite having been selected as responses to a graft of the organiser to a remote region), which provides a strong indication that the node graft assay recapitulates many, if not all, the steps of normal neural plate development in the embryo;

(c) we have complemented the in situ hybridisation-generated expression patterns by single cell RNA-seq analysis from several relevant stages of normal neural plate development. We find that components of the GRN are co-expressed in the same cells, and their timing of expression closely follows their hierarchy in the GRN. This provides strong validation of a transcriptional network that operates at the level of single cells, in normal neural plate development, not just in the whole tissue and as a result of an artificial experimental situation;

(d) we have now added new analysis (and a new figure, Figure 6) predicting the “network hubs” that may represent particularly critical regulatory genes in the network. This includes a few previously known key regulators but also several new ones. This illustrates an aspect in which the GRN can be particularly useful. Finally,

(e) we have now tested whether one of those predicted “network hubs”, MYCN, is required to regulate the expression of its predicted target ZNF423, using a morpholino knockdown. Indeed we find that it does, which provides validation of this prediction. However we would like to comment that in the case of other enhancers such as the N1 and N2 elements of *SOX2*, it appears that no single TF that binds to the element is individually essential – rather, several TFs need to bind but the precise combination may not be so proscriptive. We feel that this could be a general feature of such developmentally regulated elements, therefore testing whether a factor is “essential” can give a misleading view of gene regulation in that context. We have added a brief discussion of this point (page 13).

The reviewers suggested that we should use the GRN to test other aspects like the importance of particular “signals”, or “competence”. Any one of these is a massive undertaking, far beyond the network itself, and certainly beyond the concept of providing a “resource”. In fact, we have been engaged in doing just this in the laboratory and two other large studies are in preparation, one investigating the individual activities of many (more than 100!) candidate signalling molecules expressed in the node, and the other comparing three non-competent regions of epiblast with competent epiblast to understand the factors that may regulate competence. To make it clearer that the present study is a resource focused on the hierarchy of regulatory interactions between transcription factors in the responding cells, we tried to change the title by adding the subtitle: “: Dynamics of transcriptional responses.” – however we were told by the editorial office that two-part titles and punctuation are not allowed by the journal so we had to revert back to the original title. In any case we hope that it will be clear to most readers that “gene regulatory network” does refer to transcriptional regulators and a network that depicts the interactions between them, within a cell, and other alterations we have made to the text should also make this explicit.

3) Improvement in transcriptomic and epigenomic data analysis along the lines suggested by reviewer 3 are essential to consolidate and strengthen the study;

We have followed all of these suggestions. More details below in response to Reviewer 3.

4) Deposition of the in situ hybridation data in a public repository;

Although *eLife* is fully open access and all the ISH data is included in this study (main Figure 7 and its supplements, 1-8), we will also submit all ISH images to GEISHA directly upon acceptance of the paper for publication – this will facilitate searching for these expression patterns by the community.

5) Please discuss how node graft modulates the AO program (reviewer 4, point 1).

We have already discussed this at some length in our previous paper (Trevers et al., PNAS 2018) where we proposed that an early step in the responses to the node graft is to “rewind” the AO cells to a pluripotent state similar to that of ES cells. However, we have now added brief discussion to make it more explicit (page 14, second paragraph and page 15).

6) Please address the comments on pioneering factors, poised/bivalent promoters (reviewers 3 and 4).

Done – see Discussion, pages 12-13 (especially the first paragraph of the new section entitled “The network – interpreting enhancer states”).

7) Take home messages from both the synthetic network analysis and the chosen examples should be clarified.

We have gone through the entire manuscript and, wherever possible, have added clear introductions and conclusions to each of the sections of the Results, to clarify the purpose and the take-home messages. This includes the section on BRD8, which had been chosen as an example for dissection of multiple elements controlling the expression of a gene. We have also expanded the Discussion to explain more fully the biological context of the work as well as potential useful applications and future directions using the resource.

Reviewer #1 (Recommendations for the authors):In this manuscript, Trevers and colleagues undergo a genome-wide exploration of the molecular actors of neural induction in chick embryos. Specifically they describe the gene regulatory network (GRN) governing the patterning of extra-embryonic ectoderm into neural ectoderm, upon the graft of an early Hensen's node ectopically, an assay extensively used by these authors in the past to understand how neural induction and neural commitment were activated by node grafts: They have notably previously identified early responsive genes such as ERNI, mid-term responsive genes (e.g. Sox2), and late specification markers such as Sox1. It still remained unclear to which extent this model faithfully mimics the mechanisms of endogenous neural tissue induction at the midline of the embryo, a question explored here.Here, the authors examine the states of the induced extra-embryonic ectoderm at three different time points after grafting (5h, 9h, 12h), using RNAseq, ATACseq and ChIPseq for histone active/repressive marks. They further select about 200 transcriptional regulators (transcription factors and chromatin modifiers) to conduct a Nanostring analysis at 6 time points and build a large GRN for neural induction, with its temporal dynamics, based on the putative binding sites for these regulators located around target genes.This study generates an extensive set of data, integrated to a genome browser, allowing one to explore the refined details of the molecular events triggered by the graft of a node into the ectoderm. Moreover using the expression patterns for 174 transcriptional regulators in vivo, and single cell transcriptomes, the authors compare their GRN to the dynamics of gene (co)-expression in the neural plate and conclude that the temporal dynamics of the ectopic neural induction assay matches the sequence of gene expression during neural induction at the midline.This study nicely confirms the previous model of neural induction in chick embryos. The new data could be used to predict and assay novel details of neural induction or neural commitment as suggested by the introduction. More than 5600 interactions were predicted but they remain to be validated: for example, the direct regulation of a few new GRN components by their predicted upstream modulators could have been tested. At this point, it is unclear how this study expands our knowledge of neural induction. Nonetheless, this dataset will constitute an important resource for future exploration of neural induction mechanisms.

We thank the reviewer for these comments. Indeed, we do provide some important functional validation of several of the predicted enhancers, using constructs containing the putative regulatory region, a minimal promoter and a fluorescent reporter, co-transfected with a ubiquitous reporter to show the transfected domain. This was done for 6 separate regulatory regions (associated with SOX11, SETD2, *SOX2*, CDX1, *GLI2* and SIX3). In addition, we have now added a prediction of “network hubs” (or “core genes”) that may be particularly important. Further functional data includes knock-down of one of the core genes (NMYC) to test the consequence on its predicted target ZNF423. We therefore feel that the study includes significant attempts at validation of the network, by 5 different methods: comparison of time courses after grafting a node with hierarchy of gene expression in the normal neural plate by in situ hybridisation, single cell RNAseq of normal neural plate at several stages, reporter assays in vivo for six of the predicted enhancers, prediction of putative “core genes” in the network and validation of the regulation of a target gene by one of these using morpholino knockdowns. These findings increase confidence in the predictions generated by this method, as well as strong indication that neural induction elicited by a node graft closely recapitulates the hierarchy of events that accompany development of the normal neural plate.

The direct demonstration of the role of one or two new regulators would add significant validation to the predictions of this study. Among the 174 regulators studied, some probably present a particularly interesting expression patterns, or position in the GRN, suggesting a critical yet unknown function. Demonstrating the role of such a factor on the known elements of the GRN (ERNI, Sox2, Sox1 etc) and validating its regulation by the predicted upstream regulators in vivo, during neural induction at the midline, would demonstrate the discovery power of this new GRN.

Thank you for this suggestion. We have now conducted knockdown experiments on one of the new predicted regulators, NMYC, and confirm some of the predictions for the endogenous domain of expression of the target genes, as suggested by the reviewer. Together with the single cell RNAseq data of the endogenous neural plate, the in situ hybridisation patterns of the factors in the same region, and the validation experiments using reporters, this suggests that the GRN generated for responses to a graft of Hensen’s node does appear to replicate events at the normal neural plate during development in un-manipulated embryos.

Reviewer #2 (Recommendations for the authors):The manuscript by Trevers, Lu et al. Is an impressive and useful dataset on the emergence of neural, placode, neural crest and non-neural ectoderm at the levels of RNA expression, and chromatin marks. The data have been assembled into hypothetical Gene regulatory networks based on timecourses of expression at closely spaced intervals. The most impressive part of the work is this fine scale temporal analysis of gene expression and regulation, which occurs over several hours in the chick epiblast, where the embryological assays that define competence and commitment can be correlated with gene expression, and with changes in chromatin marks. The work primarily emphasises the genes that change in response to node/organizer grafts, but the paper also studies the emergence of these transcripts in normal neural development, and shows that the node inductions closely mirror the changes in the normal ectoderm.While the work is potentially very useful, the current manuscript does not digest the data to generate biological insights of particular interest to the reader. The Venn diagrams summarize the number of changes, but don't give much sense of how they fit into our current understanding of neural induction, and the Biotapestry diagrams are difficult to analyze by eye when they include so much detail. The ppt file provides more annotated versions of the Biotapestry, but again, it is difficult to extract biological interpretations from them.

We thank the reviewer for these comments. Indeed as the reviewer appreciates, what distinguishes this study from all others in this field is that we provide very detailed temporal resolution of the responses of cells to signals from the organizer, but also that we analyse these changes specifically in the responding tissue rather than whole (or large parts of) embryos. These results are then compared to what happens during normal development including some functional assays of the activity of the predicted regulatory elements in the endogenous, normal neural plate. To avoid the risk of being accused of being too speculative, we chose not to generalise too much about the mechanisms of neural induction, which would be required to generalise these findings into “insights” or “biological interpretations”, but rather to present as objective a picture as possible of what happens after cells receive signals from the organizer, in fine time course, and concentrating on transcription factors, their binding sites and changes in their expression and the chromatin marks associated with their chromosomal loci. We have now clarified this further by changes in the Introduction and especially a significantly extended Discussion, which addresses some key biological questions. This resource can now serve as a tool to query mechanisms of neural induction such as the contribution of particular signals to different aspects of the GRN, what changes as the competence of the responding tissue changes, the mechanisms of anterior-posterior patterning, etc. However, our analysis also highlights how difficult it is to define concepts like “commitment” (at what point do the changes experienced by cells become difficult to reverse) and we believe that this is in itself a valuable contribution, which we briefly discuss. But to answer this question definitively and refine what the concept implies from a mechanistic point of view would require a huge amount of additional work, perhaps in multiple developmental events and multiple model systems and this is likely to take many years and the work of many groups. We hope that the GRN presented here will be an invaluable resource to guide such future work.

Biologically, the paper does present some arguments that suppression of non-neural gene expression (by TFAP2C) and other data supports the contention that suppression of non-neural and neural border fates is important in neural induction, but there is little discussion of neural induction itself. On the positive side, there are several already known and some new cis-regulatory modules that have been identified, and some tested for function by electroporation, but this just makes the reader want to know what motifs or logic lie in these domains. For example, the ability of the node, but not BMP antagonists to induce neural tissue in the area opaca is consistent with the early expression domain of Sox3 as a neural competence defining gene, but it would also be useful to couch some of the discussion in terms of what other genes might lead to increased competence to respond to FGFs or absence of BMP signaling over the timecourse of analysis. Recent work for the Briscoe lab has suggested that FGF signaling is important in conditioning posterior fates prior to neural specification in mammalian stem cells, supporting data in *Xenopus* and fish that FGF signaling has a large role in defining a competence region for posterior neural induction. It would be interesting to relate the current work to these kinds of findings. What are the genes that predispose to anterior responses or posterior ones, and how separable are they in time? Also, what can be learned from the large dataset on commitment in neural induction and to what extent does it correlate with expression of rostrocaudal or mediolateral regional markers in the dataset?

We share the reviewer’s desire for a greater understanding of the entire process of neural induction. However, our analyses have started to reveal that there is huge complexity accompanying this process and that it is not always easy to assign a specific classical embryological concept (like “competence”, etc) to a specific change in gene expression (including Sox3) or other property. To ask these broad questions requires better tools, sophisticated biological assays for these concepts, and a more comprehensive dataset that offers greater spatial and temporal resolution. For this reason, we have chosen to focus this paper on transcriptional regulation only, i.e. the responses of cells to receiving signals from the organizer. Even to do just this required more than 15 years’ work from many people in the laboratory (and we took advantage of several technological advances made during this time). We tried to change the title by adding the subtitle: “: Dynamics of transcriptional responses.” – however we were told by the editorial office that two-part titles and punctuation are not allowed by the journal so we had to revert back to the original title. In any case we hope that it will be clear to most readers that “gene regulatory network” does refer to transcriptional regulators and a network that depicts the interactions between them, within a cell, and other alterations we have made to the text should also make this explicit. Indeed, this is a platform to ask questions like those mentioned by the reviewer, such as the contribution of different signals (like BMP, FGF and others), the regional (anterior/posterior) character of induction, competence, and others. Indeed, two follow-up studies are in preparation, one specifically addressing the individual activities of more than 100 signalling molecules expressed by the organizer, and the other exploring the reasons why some tissues are competent to induction by the organizer and some are not. Each of these is also a very large study, taking advantage of the GRN presented here. Likewise, it should be possible to extend this work to head-tail patterning of the CNS and how it relates to signals from the organizer and other tissues, timing and other parameters. We feel that the GRN we present offers a series of very sensitive assays to explore these aspects, as well as to compare between species. The newly expanded Discussion touches on some of these issues more explicitly.

So overall, this is a high priority paper for publication as a resource, but the biological insights that come from the work are surprisingly few, making the manuscript disappointing to read as a regular article. I am not going to follow the prescriptive requirements in the eLife instructions, but I would hope that the authors include more analysis and comparisons to the paper to bring out the strengths of the datasets.

We have now decided to publish the paper as a “Resource”. We are disappointed however that the reviewer considers that revealing the degree of complexity of the transcriptional and chromatin responses following a few hours’ exposure to a graft of the organizer, in fine time course in the responding cells, is not a biological insight in itself. Previous publications from our group hinted at such complexity based on just a few genes that had been isolated in screens and their expression patterns, but this study brings this to a new level of understanding and temporal and spatial resolution. It also more clearly brings forward questions about what is the biological and molecular basis of concepts like “specification”, “commitment”, “competence” and others. Much of the recent literature using single cell RNAseq tend to pick on one or a few chosen genes to consider as “cell fate specifiers” or as markers of these transitions, but looking at our GRN opens the question as to what this means. A study that focuses on a single gene using traditional gain- and loss-of-function approaches can easily be made to tell a story about something critical about that gene in a process, but such studies overlook the underlying complexity of the biology. Other studies of gene regulation (including for example those of the Kondoh group, but also many others in other model systems) also emphasize that it is probably inappropriate to expect all of development to be explainable through a list of “necessary” and “sufficient” components. Our GRN more closely resembles the Waddington "epigenetic" landscapes, where cells make decisions by a combination of previous steps/history, each enhancing the probability of adopting an identity. It also appears that the same state may be reachable by different routes – this is consistent for example with findings in *C. elegans*, where the same cell type can be generated by cells from entirely different lineages, presumably under the influence of many different signals and pathways.

Therefore, the GRN shows a complex dynamic transition between cell states as defined by the combination of TFs expressed by cells with time of exposure to the organiser, rather than dramatic catastrophic changes that could easily be associated with the classical concepts. We hope that our new Discussion (and aspects of the Introduction) raise some of these points in a way that should attract others to explore these questions in greater depth. But we are a little saddened by the reviewer’s statement that this contribution is “disappointing to read”, simply because it does not have a single headline message (or “spin”) – in our view we are opening a window on what the Biology itself is telling us.

Specific comments:The clustering of single cell data is interesting, but it is hard to understand why the clusters were not labelled using the data (illegible) in figure S4, indeed why is S4 not in the main paper instead of the less informative data in Figure 2 or 3B. Why was BRD8 chosen for detailed presentation? Is it a neural specific gene? I note that it is not in Geisha, and its expression is not described here. The biotapestry representation falls down with this many linkages, and it would be good to have included a simplified diagram for the different sites/stages.I had hoped for some representations of specific genes that are central to the likely steps in neural induction, (is BRD8 one of them?) along with a discussion of what feeds into them, and goes beyond the previous characterization of the Sox2 regulatory motifs.

Thank you for these suggestions. We chose BRD8 for several reasons, including that the coding sequence is located in a locus that is relatively gene sparse (therefore making it more likely that the regulatory elements really do control this gene), it contains several discernible regions, and its temporal expression pattern correlates particularly clearly with changes in expression of the transcription factors whose binding sites are included in each of the putative enhancer elements. We have now included a short explanation for this choice. Indeed, its expression was not included in GEISHA or in the literature in other species, so we decided to explore this directly. We now provide new in situ hybridisation data (which we are also depositing in GEISHA) for BRD8 and we are particularly pleased to see that the normal expression of this hitherto undescribed gene in the context of neural plate development matches our predictions precisely in time and in space. This is a nice confirmation that the GRN works as intended.

As we also describe in the text, the GRN independently identified several of the regulatory elements associated with *Sox2* (from the lab of Hisato Kondoh) as well as some putative novel ones for that locus.

To address the request for “genes that are central to the likely steps in neural induction”, we have now added a new figure (Figure 6), using a new approach to predict genes that may represent “hubs” in the network and thus may occupy more central positions. One of them is MYCN, whose importance we have now explored directly by knockdown experiments, which confirm that does indeed regulate its target ZNF423 in the predicted manner. However as also outlined in response to the previous point by this reviewer, we query whether the concept of a gene representing a “central” step in neural induction is meaningful. Although some genes may be required for the whole process (ie. knockdown will cause the whole of neural induction to fail), this does not mean that only those genes are worthy of investigation. In fact we are presenting the view that it is the whole collection/panel of gene expression that establishes “states” through which cells progress along development. Hisato Kondoh’s work, among others, has already shown that some enhancers can be regulated by different combination of several transcriptional regulators whose binding sites are represented in those enhancers. Maybe the view of one-gene-one-function has been over-emphasized by genetic approaches. We think that GRNs like this offer an entry point to explore this in a different way. We have now commented on some of these ideas in the revised Discussion.

Reviewer #3 (Recommendations for the authors):In this work, Trevers et al. combine classic Hensen node transplantation experiments and transcriptional and epigenetic profiling to characterize the gene regulatory network at work during anterior neural commitment. Besides, they combine the analysis of chromatin marks for active/ repressed enhancers together with transcription factor binding sites (TFBS) prediction to define the cis-regulatory code controlling the expression of the genes involved in this process. These next-generation-sequencing based approaches are further supported by extensive analysis of gene expression patterns by whole-mount in situ hybridization and enhancer transgenesis assays. The authors also use scRNAseq analysis to prove that the gene transcriptional dynamics characterized in their node graft model closely recapitulate those of normal anterior neuroectoderm induction. The most notable outcomes of this overall nicely conceived and experimentally well-performed study are the definition of a GRN composed of 175 transcriptional regulators and the characterization of 79 genes not previously associated with neural induction being differentially expressed during this process and will be of interest for the neurodevelopmental biology community. Although the transcriptional profiles associated with the neural induction process have been in part characterized in bulk or sc cell transcriptomic profiling of both mouse and chicken embryos, including recent work from the same authors [1]-[4], the resources generated in this study are intended to further dig into the molecular event occurring during anterior neuroectoderm induction. The main limitation of this study is that it remains largely descriptive without fully addressing the logic of the GRN governing the neural induction process, thus limiting the impact of the work. This aspect could perhaps benefit from a more detailed analysis of the data generated in this study.The main experimental concerns about the study of Trevers et al. are related to the design and analysis of the transcriptional profiles of induced vs non induced ectoderm.Suggestions/comments about the analysis of the GRN and gene cis-regulatory code are also presented below:– In the Material and Methods section, the authors describe having analyzed the RNAseq data against different chicken genome assembly versions, ranging from galGal2.1 to galGal5. The source data file provided by the authors provides the differential expression analysis performed in the galGal3 and galGal4 assemblies. Thus, it is difficult to understand which analysis was used for the elaboration of the figures and the description of the results. Provided that no specific conclusions arise from the analysis of different genome assembly versions (in which case it should be clearly stated), I find the description of the different analyses unnecessary and even confusing. In my opinion, authors should describe only the one used for the elaboration of the figures and data interpretation in the manuscript, preferentially using the latest genome assembly and gene annotation versions.

We thank the reviewer for bringing this to our attention. This work encompasses analysis done over a period of many years, during which the sequence quality and annotations of the chick genome evolved considerably. In Figure 1 we show the criteria used to choose the genes that were going to be put on the NanoString, using the latest version at the time (galGal3 and galGal4). All the rest of the analysis, including all aspects of network construction, uses a single version, galGal5. We have now made this explicit in the revised Methods and Supplementary Methods sections.

– The RNAseq analysis presented by the authors has been performed using individual pooled samples of induced /non induced dissected ectoderm, which imposes identifying differentially expressed genes on the basis of p-value rather than on FDR. While I understand that the complexity of the experiment limits the possibility of performing a high number of biological replicates, the robustness of the data would greatly benefit from having at least a second biological replicate, particularly because this work aims at characterizing the gene regulatory network operating during neural induction based on the analysis of differential gene expression across the process. Besides, the authors report having used different criteria for genes in their galGal3/galGal4 -based differentially expressed analysis and to produce the galGal5 FPKM table of differentially expressed genes (log2(FC)>1.2 in galGal3/galGal4 based analysis and FC>1.5 in the latter case). The log2(FC) results in the source data table for the galGal3 and galGal4 assemblies also seem to be presented in a different manner which does not help to correlate the results. Authors should uniformize their analysis criteria or clearly state the rationale for the use of different parameters.

Thank you for highlighting these issues. As explained above, the initial RNAseq analysis of pooled samples was mainly used for selection of differentially expressed genes to represent on the NanoString for more detailed study of the timing of expression. All NanoString experiments (including the same time points as done by RNAseq) are performed as biological (independent) triplicates (including the same time points as sampled by RNAseq to allow for more quantitative comparisons across time points). All epigenomics experiments were also performed as independent biological triplicates. We have added an explanation in the Methods section clarifying this.

– While the authors report a roughly similar number of activated and repressed genes in induced vs non-induced ectoderm samples (based on RNAseq data) the number of activated and repressed regulatory elements strongly diverge between the two samples, with a large majority of elements activated (particularly at 9h and 12h post grafting) rather than repressed. This difference could be due to "technical" reasons (e.g: the presence of a fraction of cells in the samples where certain regulatory elements are repressed only in a subpopulation of cells may not allow to distinguish these H3K27me3+ elements from those that not at all active, resulting in the identification of only those region that have a nearly all vs nothing response). Alternatively, it could be due to the fact that gene activation relies on combinatorial enhancer activity while its repression may require the action of fewer regulatory elements. Could the authors comment on this?

Figure 2 and its supplements 1-3, offering a global view of activation/repression marks, both show that at early time points, activation marks seem to dominate, but by 9-12 hours there are many more repressive marks. Looking at transcription, we see that at the very start (1-3 hours), many genes are repressed and a few are activated: this probably corresponds to the establishment of what we have called the “common state” (Trevers et al., PNAS 2018). Thereafter, an increasing number of genes is activated – these would include those involved in specification and differentiation of the forming neural plate. Therefore, the changes are quite dynamic over time.

Further complexity is revealed by looking at Figure 5 as an example of several regulatory elements contributing to the regulation of a gene (BRD8). The GRN suggests that its expression is regulated by a combination of five enhancers and one repressive element – we think that repression can also act in combination with activators, especially to define precise spatial territories and/or fine dynamics of expression. We have included a brief discussion of some of this (pages 5-6).

– Authors define the regulatory elements associated with a specific gene based on the correlation in their activity and the gene transcriptional state and on the distribution of CTCF sites. Although it is known that CTCF sites contribute to defining the limits of topologically associated domains (TADs, in which the gene regulatory landscape should be located), they are also found within TADs. Furthermore, considering the orientation of CTCF sites is also an important indicator of whether two CTCF bound regions (or clusters thereof) may contribute to define a gene regulatory domain. The authors could try to improve the definition of the gene regulatory domains by using available chicken HiC data, in combination with CTCF coverage and site orientation. Since TADs are in large part independent from gene transcriptional activity different HiC datasets (eg [5]-[7)] could be combined.

Thank you for highlighting these important issues. Indeed, spatial and temporal aspects of gene expression (and cell/tissue specificity) are controlled by different elements including enhancers, silencer elements and TAD structures. We considered using published HiC data (and other ChIP-seq datasets) to enrich our own. However, all previously published data that have been generated for the chick and other higher vertebrates are based on quite different tissue samples to those used here, therefore it would be counterproductive to use this information to annotate or curate tissue and stage-specific data generated directly from the cells for which we are trying to define the GRN. However, since TADs are believed to be demarcated by “constitutive” (i.e. non-changing) CTCF-binding, cohesin-containing sites, we decided to use published data to identify these sites, and thus provide limits for regions within which we looked for candidate regulatory elements. This information has been amalgamated into the browser tracks we have generated.

– Combining the analysis of epigenetic states and transcriptional profiles of induced/ non-induced ectoderm with binding site prediction of the differentially expressed transcriptional regulators is an interesting approach for addressing the gene regulatory network at work during the neural induction process. However, in my opinion, the analysis presented in the manuscript and figure 3 falls shorts compared to valuable resources generated by the authors (although this is not my field of expertise). For example, would it be possible for the authors to describe better the structure of the network in terms of the distribution of connections (predicted regulatory interactions) per node (genes), perform an unbiased identification of kermel genes and overrepresented network motifs (e.g. feedback and feedforward loops), possibly highlighting some corresponding to known interaction as well as new gene-gene connections?

Thank you for these suggestions. We have now included new analysis (using calculations of Centrality) to identify core genes in the network. The new Figure 6 and the relevant description in the Results (page 6) and Methods section summarise the main findings. After exploring how we could illustrate feedback and feed-forward predictions, however, we decided that this would increase the complexity of the network so much as to render it incomprehensible and more difficult to use. Moreover, we are concerned that including all of these predicted interactions takes the model to a higher level of further speculation, which we would prefer to avoid so that the network remains as robust as possible.

– The role of pioneer factors in the control of cell fate acquisition/ reprogramming is of transversal interest in the developmental biology and chromatin and transcription fields (eg [8]-[10]). The neural induction experimental paradigm used in this work and the datasets generated could point out transcription factor pioneering activities relevant for the neural induction process and provide interesting insights for the understanding of the logic of the neural induction GRN, for example, by analyzing TFBS overrepresented in regulatory elements that are active long before the transcriptional onset of their (putative) target genes.

Thank you for this comment. Please see our previous point. In addition, we feel that overrepresentation of particular binding motifs is not necessarily an indication of the importance of the binding transcription factor(s) – in some regulatory sites it is possible that just a single binding site is critically important, but it is not possible to predict this without testing each individual binding site and combinations thereof by constructing mutations for each regulatory element.

– Despite having generated an important amount of ATACseq data, the authors present these results very briefly, not indicating the statistics of elements displaying differential accessibility across samples and explaining very concisely how these data were used to aid in the selection enhancers for the transgenesis assay. Chromatin accessibility profiles could also help in the analysis of differentially active/repressed/ poised elements presented in Figure 2 and Figure S2-S3 and in the identification of elements targeted by pioneering TFs. Could the authors expand on this analysis (provided that the quality of the data allows genome-wide analysis of changes in chromatin accessibility)?

Thank you for the suggestion. We have found that chromatin accessibility (as marked by ATACseq) changes in a subtle way at these early stages of development. We also see more variation between samples than for ChIPseq datasets, therefore we opted not to draw too strong conclusions from ATACseq – hence we mainly used this for confirmation of putative enhancers identified with the other methods. However, we have now added a heatmap for ATACseq (Figure 2 supplement 2) illustrating the changes taking place. We also changed the organisation of the text to reflect the contributions of the different methods to the construction of the GRN.

– Authors define poised enhancers as those displaying either both H3K27ac/H3K27me3 marks in the same sample or lacking both of them. This definition remains delicate both because of the potential heterogeneous cell population in the dissected tissue (as authors discuss) and by the lack of H3K4me1 profiles. In this sense, crossing the H3K27ac/ H3K4me3 and ATACseq data could improve defining these poised elements (H3K27ac-, H3K27me3+ and high chromatin accessibility [11]).

Please see above comment that ATACseq data were too subtle to be informative. We also tested other antibodies for ChIPseq and found that only H3K27ac and H3K27me3 were robust enough at these stages, with limiting amounts of appropriate tissue, to use for constructing the network. This could also indicate that regulation of the early steps of gene expression may depend more on TF binding than chromatin accessibility and conformation, the latter becoming more important as cell fate decisions become stabilised. We have added some new discussion about “poised”/”bivalent” enhancers including the possibility that there may be underlying cell heterogeneity, or indeed different marks on the two alleles within the same cell (pages 12-13).

– Authors claim having identified the identification of 79 transcriptional regulators not previously associated with neural induction and differentially expressed in the node graft model and during normal chicken neural induction. This finding would deserve further exploration/ discussion, for example, by analyzing the expression of these genes in mouse scRNAseq datasets or from RNAseq profiles in cell culture models of ES/IPS cell-Neural differentiation.

In the manuscript, we confirmed the normal expression of 84 transcriptional regulators that are upregulated in response to a node graft, 79 of which had not previously been associated with neural induction. Using the iTranscriptome database of regional RNAseq expression, we can confirm that 76 of these transcriptional regulators are expressed in neural territories of the mouse embryo at E7.0, during gastrulation (Figure 7 Source Data 1). We have added brief discussion of this in the general conclusions of the study (page 16).

– Based on the description of the authors of the ChIPseq data analysis, authors base their analysis of active/ repressed elements in induced/non induced ectoderm samples on qualitative criteria; i.e, regions called by MACS2 peaks in one sample but not in the other. However, using a more quantitative approach (for example, MAnorm analysis of coverage levels) may also highlight elements that increase or decrease their activation/repression state in induced vs uninduced ectoderm samples as well as across time, which could contribute to answering the question raised in the public review. Besides, and related to that point, if the analysis is restrained to gene promoters, does the proportion of elements differentially enriched in H3K27ac/H3K27me3 reflect the transcript data? Are H3K27ac /H3K27me3 elements located at a different average distance from gene TSS?

We thank the reviewer for this suggestion. MANorm is another method that can be used to normalise counts so one can run differential expression analysis, comparing between ChIPseq samples. In fact, MACS2 can do the same thing but we didn’t use the differential expression analysis function – this analysis would be useful to identify regions that are enriched with one signal (one histone mark) in one sample but not the others. However, we would not be able to identify conditions undergoing “no change” or a “poised” status.

We used MACS2 output to quantify and select regions highly enriched with either H3K27ac or H3K27me3 by comparing to the inputs with cut-off p value < 10^-5^, FC>1.2 and q value > 3. We have now added a more detailed description in the Methods section, as well as a new figure (Figure 2 supplement 3) and associated text in the results (page 4). H3K27me3 is depleted at upstream flanking regions of upregulated genes in comparison to downregulated genes.

– The new advances in RNAseq library preparation and column assisted RNA extraction from low cell number samples allow to produce reliable RNAseq data from considerably lower amounts of tissue, thus eventually reducing the technical challenge of increasing the number of replicates for the transcriptional profile analysis.

Indeed, we agree with this reviewer. Had we started this work now, we would probably have chosen to use RNAseq to examine changes in gene expression with fine dynamics (done here with NanoString). When we started the study more than 15 years ago, we were limited by the sensitivity of the methodology available at the time – we chose NanoString for quantitation because of its reproducibility and great sensitivity as well as allowing the expression of hundreds of genes to be quantified simultaneously in the same sample, yet using relatively small samples.

Reviewer #4 (Recommendations for the authors):This study represents a milestone in the field of neural development, uncovering thousands of predicted regulatory interactions underpinning the process of neural induction and more generally, underlines the complexity of the process of cell fate acquisition. The work constitutes an important resource, presenting and validating useful data sets that capture transcriptomic changes (bulk RNA seq, scRNAseq, Nanostring and in situ hybridisation) leading up to expression of mature neural plate markers. Moreover, by combining ChIPSeq for key chromatin modifications and ATAC seq data the authors identify putative enhancer regions mediating neural induction and carefully validate six of these using mis-expression in the early chick embryo. Overall, the work opens a new frontier in the analysis of molecular basis of neural induction – the logic (beyond the hierarchies identified here) of the thousands of transcription regulating events will require extensive further experimentation.Overall, the experiments are well designed, the data are clearly explained and for the most part well presented in the figures. The paper would be strengthened by inclusion of examples of conservation in mammals, of the key GRN loci identified in the chicken. This reviewer does not have the expertise to comment on detail of data analyses packages and approaches presented.

We thank the reviewer for these positive comments. Concerning conservation in other species such as mammals, we have included DREiVe analysis in the paper to predict equivalent conserved regulatory sites in a number of vertebrates (including mouse and human and other mammals). The correspondence is presented in specific tracks on the browser we are providing with this paper, allowing direct alignment/comparison between species. In addition, we have now checked the expression of the upregulated genes in mouse (iTranscriptome – Patrick Tam’s spatially annotated RNAseq results) which shows strong conservation of key patterns of expression. The results are shown in Figure 7 Source Data 1 along with more discussion of evolutionary conservation in the conclusions (page 15).

1) "Of these, 4130 were upregulated (enriched in "induced" tissue) and 4543 were downregulated (depleted in "induced" tissue) relative to the "uninduced" counterpart."It is an important finding that Organiser induced gene expression changes involve down regulation as well as up regulation of genes in the AO epiblast – can the authors provide an indication of how much down regulation is due to initial starting cell state and may not reflect the endogenous neural induction process?

This is an interesting and important question. Induction implies changing over from one state/identity to another, so it’s expected to some extent that the starting state should be turned off as the new state is acquired. However, we do compare our findings to the progression of events in the normal neural plate at different stages (from scRNAseq data) and show that there is a strong correlation, suggesting that the early stages seen in the AO assay do mirror events in the normal neural plate during its development, including the earliest stages. We previously suggested that the responses to different inducing signals (to the node for neural induction, to lateral head mesoderm for placodal induction, etc.) all seem to share similar initial steps (which we named the “common state” Trevers et al., 2018), which then diverge depending on the signals to which they are exposed. In the normal embryo this expression profile is seen just prior to gastrulation, implying that the neural induction process elicited by a grafted organiser begins by “rewinding” the clock to a more pluripotent epiblast state before redirecting cells to a neural plate identity. Figure 7 and its supplements 1-8 include some of the key data, and we have added new discussion of this in the paper (pages 14 and 15), addressing the questions suggested by the reviewer.

Related to this point (page7 para 2), it is important to have established the relationship between AO epiblast response to the node and sequence of gene expression patterns during normal/ endogenous neural plate formation. The authors note the similarities, but were there also differences that would allow better discrimination of regulatory steps that reflect the AO epiblast cell state? These points should also be considered/discussed in the text.

The unbiased PCA plots (Figure 7B) show a very strong correspondence between the starting state of the area opaca and the central embryonic epiblast of the normal embryo. Looking at the differences we felt that the only differences observed are minor ones (for example differences based on the assumed timing of the endogenous events) and many of them can be ascribed to ISH being less sensitive than RNAseq or NanoString for detecting low levels of transcript.

2) H3K27Me3 is not always a marker of transcriptionally silenced genes. While I note the authors comment that " this is often the case" more caution in subsequent interpretation may be required, along with discussion of further chromatin modifications which could corroborate gene silencing.

Thank you for this point. K27Me3 has been strongly associated with silencing – we have now added several references (Tiwari et al., 2008, Heintzman et al., 2009, Creyghton et al., 2010, Kharchenko et al., 2011, Rada-Iglesias et al., 2011, Tolhuis et al., 2011, Zentner et al., 2011, Bonn et al., 2012) to support this statement (page 4). We did explore other modifications with ChIP, but in our hands only K27Me3 and K27Ac worked well. However, please note that in most cases we have been very cautious by using not only the K27Me3 mark, but also real, associated changes in relation to the previous time point in both mRNA expression and changes in chromatin marks from K27Me3 and K27Ac.

3) Can the authors explain their choice to use the term poised as opposed to bivalent, when referring to gene loci associated with both H3K27Me3 and H3K27ac modifications. Bivalency is usually inferred when H3K27Me3 and H3K4Me3 chromatin modifications are present, was any such analysis also carried out to confirm the "poised" state.

Thank you for this interesting point. We have added some new Discussion on this issue including different possible interpretations of “poised” and “bivalent” (pages 12-13). We did explore other modifications (including H3K4Me3) by ChIPseq but only K27Ac and K27Me3 worked well. We use “poised” to indicate some degree of apparent contradiction between K27Ac and K27Me3, as markers of active and repressed sites respectively. We also discuss the alternative possibilities that “poised” may reflect either that the cell(s) are in transition or in a “neutral” or “intermediate” state, as most of the literature has generally interpreted, or alternatively that the tissue might contain a mixture of cells some of which have the locus activated and others have it repressed, or the third option that the two alleles may differ in their marks even within the same cells. We are unable to distinguish between these three possibilities by currently available methods, but wanted to discuss that they are all possible. Because of this, we also feel that the term “bivalent” may be even more misleading, if it turns out that the apparently contradictory marks are either in different cells or even between the two alleles within a cell. These issues are now addressed explicitly in the revised Discussion (pages 12-13).

4) "In situ hybridization was performed for genes encoding 174 transcriptional regulators (including 123 that are represented in the GRN) at 4 stages: pre-streak (EGKXII-XIII), primitive streak (HH3-4), head process/early neural plate (HH5-6) and neural fold/tube (HH7-9)".Are these ISH data deposited or accessible somewhere? An appropriate place for this resource that is familiar and accessible to the scientific community is Giesha http://geisha.arizona.edu/geisha/ This should be included in the data availability section.

Indeed we are depositing all the expression patterns in Geisha as soon as the paper is accepted for publication, so they should be available and fully searchable by the community. As requested, this has been stated in the “Data Availability” section.

5) Page 7 para 3 (Figure 6)The "ectoderm" dissected and analysed here should be more accurately defined. Figure 6A schematic shows that the authors excluded the ventral midline at HH4 and HH6, but were perhaps unable to do this at later stages? This sets in context their finding that "Cells collected from the two earliest developmental stages (HH4 and HH6) each form a distinct group, whereas cells from later stages (HH8 and HH9+) are clustered primarily according to cell type (Figure 6C)". The authors should comment on inclusion of the ventral midline in these data sets. It is also curious that cluster 12 includes a node marker ADMP, as explanted tissues did not include the node in Figure 6A.

Indeed, the midline was excluded from these explants. One reason for this is that numerous previous studies have shown that the midline of the neural plate territory is molecularly and cellularly distinct from the rest of the neural plate (in fact it has even been suggested, notably by the group of Le Douarin) that this region may share a lineage with the node itself and its descendant, the notochord/head process. We assume that some of the small explants may have contained the edge of the node/notochord territory (adjacent to the midline), and this is consistent with the signature of “cluster 12”.

6) Page 9, the first sentence of the Discussion"This study unfolds the full complexity of the responses to signals from the "organizer" in very fine time course up to the time of appearance of mature neural plate markers, such as SOX1."By what criteria do the authors consider that their study uncovers the "full complexity" of responses to the signals from the organiser. I think they have uncovered the complexity.

Yes we agree – we have toned down these statements as suggested.

7) page 10 para 3Analysis of signalling pathway targets likely to be informative – should make clear that the GRN here only includes transcription factors (TF) and that changing TF expression is also very likely to be regulated by exposure to changing signalling as development proceeds – presented by neighbouring non neural tissues – rather than due simply to a hierarchal cascade of TF regulation from early stages.

In this study we concentrate only on the GRN (interactions between TFs and cross-regulatory gene interactions, trying to identify those that are “direct”) that is deployed in epiblast cells in response to their exposure to signals from the organizer. As this reviewer states, the study opens the door to numerous future experiments one of which is indeed a study of the signalling inputs and their consequences in terms of changes in this GRN. Such a study (on a very large scale) is currently in progress in our laboratory, aiming to identify which parts of the GRN may be regulated by which signals from secreted proteins expressed in the organizer.

8) The last sentence of the Discussion is almost identical to the last sentence of the Results section.

Thanks for pointing this out. We have now completely re-written the Discussion and removed this repetition.

9) Somehow at the end of the paper we are still left wanting an overview of ways forward with these data sets – in particular are there any take home messages / obvious next experiments which would elucidate the regulatory logic of the transcriptional dynamics /hierarchies uncovered here? Is it clear, for example, that many of the TFs characteristic of specific time points are targets of the same signalling pathway? Can they include examples of conservation in mammals (using the DREiVe computational tool) of key GRN loci identified in the chick?

Following the journal guidelines, we now suggest several aspects of the neural induction process that can become more tractable by using this GRN as the key resource. They include the different activities of different signalling molecules from the organizer, the changing competence of the responding tissue in time and space, and the regional nature of the responses (relation between induction and patterning). As mentioned above, we have also commented on conservation of the GRN in mammals and several other vertebrates, including associated browser tracks from DREiVe analysis as well as comparison to the expression of genes in the mouse neural plate based on published data (Figure 7 source data 1). This comparison will enable further studies to assess the degree to which the interactions are similar or divergent in different vertebrates. We have addressed this in several places of the newly expanded Discussion, including the final concluding section.